# Stochastic Momentum Methods for Non-smooth Non-Convex Finite-Sum Coupled Compositional Optimization

**Xingyu Chen**[*]
tamucxy2023@tamu.edu

**Bokun Wang**[*]
bokun-wang@tamu.edu

**Ming Yang**[*]
myang@tamu.edu

**Qihang Lin**[†]
qihang-lin@uiowa.edu

**Tianbao Yang**[*‡]
tianbao-yang@tamu.edu.

## Abstract

Finite-sum Coupled Compositional Optimization (FCCO), characterized by its coupled compositional objective structure, emerges as an important optimization paradigm for addressing a wide range of machine learning problems. In this paper, we focus on a challenging class of non-convex non-smooth FCCO, where the outer functions are non-smooth weakly convex or convex and the inner functions are smooth or weakly convex. Existing state-of-the-art result face two key limitations: (1) a high iteration complexity of $O(1/\epsilon^6)$ under the assumption that the stochastic inner functions are Lipschitz continuous in expectation; (2) reliance on vanilla SGD-type updates, which are not suitable for deep learning applications. Our main contributions are two fold: (i) We propose stochastic momentum methods tailored for non-smooth FCCO that come with provable convergence guarantees; (ii) We establish a **new state-of-the-art** iteration complexity of $O(1/\epsilon^5)$. Moreover, we apply our algorithms to multiple inequality constrained non-convex optimization problems involving smooth or weakly convex functional inequality constraints. By optimizing a smoothed hinge penalty based formulation, we achieve a **new state-of-the-art** complexity of $O(1/\epsilon^5)$ for finding an (nearly) $\epsilon$-level KKT solution. Experiments on three tasks demonstrate the effectiveness of the proposed algorithms.

## 1 Introduction

In this paper, we consider the *finite-sum coupled compositional optimization* (FCCO) problem

$$\min_{\mathbf{w} \in \mathbb{R}^d} F(\mathbf{w}) := \frac{1}{n} \sum_{i=1}^{n} f_i(g_i(\mathbf{w})), \tag{1}$$

where $f_i : \mathbb{R}^{d_1} \to \mathbb{R}$ is continuous, $g_i : \mathbb{R}^d \to \mathbb{R}^{d_1}$ is continuous and satisfies $g_i(\mathbf{w}) = \mathbb{E}g_i(\mathbf{w}, \xi_i)$, and the expectation is taken over the random variable $\xi_i$ for $i = 1, \ldots, n$.

FCCO has been effectively applied to optimizing a wide range of risk functions known as X-risks [1, 2, 3, 4], and group distributionally robust optimization (GDRO) [5, 6]. Recently, it was also applied to solving non-convex inequality constrained optimization problems [7, 8]. Several optimization algorithms have been developed for non-convex FCCO (1) under different conditions.

---

[*]Department of CSE, Texas A&M University, College Station, USA

[†]Tippie College of Business, The University of Iowa, Iowa City, USA.

[‡]Correspondence to: tianbao-yang@tamu.edu.

39th Conference on Neural Information Processing Systems (NeurIPS 2025).

Table 1: Comparison between our algorithms and prior works for the **non-smooth non-convex FCCO** problem. The complexity of SONX is for finding a nearly $\epsilon$-stationary solution (Definition 3.1), and that of SONEX and ALEXR2 for smooth inner functions are for finding an approximate $\epsilon$-stationary solution (Definition 4.1), which implies a nearly $\epsilon$-stationary solution under a verifiable condition of the inner functions. The complexity of ALEXR2 for weakly convex inner functions is for finding a nearly $\epsilon$-stationary solution to the outer smoothed objective. In this table, "WC" means weakly convex, "C" means convex, "$\nearrow$" means monotonically non-decreasing, "SM" means smooth, "PMS" means that the proximal mapping can be easily computed, "MLC0" means that the function is mean Lipschitz continuous of zero-order (Assumption 4.6).

| Algorithm | $f_i$ | $g_i$ | Complexity | Loop | Update type |
|---|---|---|---|---|---|
| SONX [6] | WC, $\nearrow$ | WC, MLC0 | $O(\epsilon^{-6})$ | Single | SGD |
| SONEX (Ours) | WC, PMS | SM, MLC0 | $O(\epsilon^{-5})$ | Single | Momentum or Adam |
| ALEXR2 (Ours) | C, PMS | SM | $O(\epsilon^{-5})$ | Double | Momentum or Adam |
| ALEXR2 (Ours) | C, PMS, $\nearrow$ | WC | $O(\epsilon^{-5})$ | Double | Momentum or Adam |

Table 2: Comparison between our algorithms and prior works for solving **non-convex inequality constrained optimization** problem. $g_0$ is the objective and $g_i$ are constraint functions.

| Algorithm | $g_0$ | $g_i$ | Constraint Sampling | Complexity | Loop | Update type |
|---|---|---|---|---|---|---|
| OSS [9] | WC | WC | No | $O(\epsilon^{-6})$ | Double | SGD |
| ICPPAC [10] | WC | WC | No | $O(\epsilon^{-6})$ | Double | SGD |
| SSG [11] | WC | C | No | $O(\epsilon^{-8})$ | Single | SGD |
| Li et al. [7] | SM | SM | Yes | $O(\epsilon^{-7})$ | Single | Momentum or Adam |
| Yang et al. [8] | WC | WC, MLC0 | Yes | $O(\epsilon^{-6})$ | Single | SGD |
| Liu et al. [12] | WC | WC, MLC0 | No | $O(\epsilon^{-6})$ | Single | SGD |
| SONEX (Ours) | SM | SM, MLC0 | Yes | $O(\epsilon^{-5})$ | Single | Momentum or Adam |
| ALEXR2 (Ours) | WC | WC | Yes | $O(\epsilon^{-5})$ | Double | Momentum or Adam |

Most of them require the smoothness of $f_i$ and $g_i$. Hu et al. [6] has initiated the study of non-smooth non-convex FCCO where both the inner and outer functions could be non-smooth. However, their results face two key limitations: (1) a high iteration complexity of $O(1/\epsilon^6)$ under the assumption that the stochastic inner functions are Lipschitz continuous in expectation; (2) reliance on vanilla SGD-type updates, which are not suitable for deep learning applications.

**Novelty.** This paper addresses these limitations by proposing stochastic momentum methods for non-smooth FCCO, where $f_i$ is non-smooth and $g_i$ could be smooth or weakly convex, and improving the convergence rate to $O(1/\epsilon^5)$. To the best of our knowledge, this is the first work to propose stochastic momentum methods for solving non-smooth FCCO problems. Unlike [6] that directly solves the original problem, the key to our methods is to smooth the outer non-smooth functions based on the Moreau envelope smoothing or equivalently the Nesterov smoothing when they are convex, which is referred to as *outer smoothing*. When the inner functions are non-smooth weakly convex, we further smooth the transformed objective using another layer of Moreau envelope smoothing, which is referred to as *nested smoothing*. Then we propose stochastic momentum algorithms to optimize these smoothed objectives.

**Contributions and Significance.** We establish two main results regarding non-smooth FCCO. In the first result, we consider non-smooth FCCO with non-smooth outer functions and smooth inner functions. With outer smoothing, we propose a single-loop stochastic momentum method named SONEX to solve the resulting smoothed objective. Our main contribution here lies at a theory that guarantees a convergence rate of $O(1/\epsilon^5)$ under a meaningful convergence measure. Specially, we show that when the outer smoothing parameter is small enough, the proposed algorithm is guaranteed to find a novel notion of approximate $\epsilon$-stationary solution to the original problem, which implies the standard nearly $\epsilon$-stationary solution to the original problem under a verifiable condition of inner functions. In the second result, we consider non-smooth FCCO with non-smooth convex outer functions and non-smooth weakly inner functions. With nested smoothing, we propose a novel double-loop stochastic momentum method named ALEXR2 for solving the resulting smoothed objective and establish a convergence rate of $O(1/\epsilon^5)$ for finding an $\epsilon$-stationary solution of the smoothed objective. Table 1 compares our methods with those of the existing works for non-smooth FCCO problems from different aspects.

Then we consider novel applications of these two algorithms to **non-convex inequality constrained optimization**. By optimizing a *smoothed hinge penalty function* with the proposed stothastic momentum methods, we derive a new state-of-the-art rate of $O(1/\epsilon^5)$ for finding an $\epsilon$-KKT solution when objective and constraint functions are smooth, and for finding a nearly $\epsilon$-KKT solution when objective and the constraint functions are weakly convex. Table 2 summarizes the complexity of our methods and existing works for solving non-convex inequality constrained optimization problems.

## 2 Related work

**FCCO**. FCCO is a special case of stochastic composite optimization (SCO) [13, 14] and conditional stochastic optimization (CSO) [15] when the outer expectation is a finite sum. FCCO was first introduced in [4] as a model for optimizing the average precision. Later, [2] proposed the SOX algorithm and improved the convergence rate by adding gradient momentum and further being improved in [16] by replacing exponential moving average (EMA) with MSVR and STORM estimators for inner function and overall gradient, respectively. All these techniques have gain success in optimizing Deep X-Risks [1] such as smooth surrogate losses of AUC and contrastive loss [17, 18]. For convex FCCO problem, [2] reformulate FCCO as a saddle point problem and use restarting technique to boost convergence rate. For non-smooth weakly-convex FCCO (NSWC FCCO), [6] leverages MSVR technique and requires $O(\epsilon^{-6})$ iterations to achieve a nearly $\epsilon$-stationary point.

**Smoothing techniques**: An effective approach for solving non-smooth optimization is to approximate the original problem by a smoothed one using different smoothing techniques, including Nesterov's smoothing [19], randomized smoothing [20], and Moreau envelope [21, 22]. Our method is related to the one based on Moreau envelope, which is a prevalent technique for weakly convex non-smooth problems. This method approximates the objective function by its smooth Moreau envelope and then computes an $\epsilon$-stationary point of the Moreau envelope, which is also a nearly $\epsilon$-stationary solution of the original problem [22, 23]. The similar technique has been extended to min-max problems [24, 25] and constrained problems [9, 10, 11, 26]. Our methods are different from these works because we consider the Moreau envelope of each outer function and design stochastic momentum methods. Similar outer smoothing techniques have been developed in [27] for compositional objectives. However, different from our work (1) their algorithm cannot handle multiple outer functions of a finite sum structure as (1); (2) their algorithm is not momentum method; (3) their convergence theory requires mean Lipchitz continuity of the gradient of the inner functions for finding a primal-dual $\epsilon$-stationary (KKT) solution of the min-max formulation of the original problem.

**Non-convex constrained optimization**: Most existing works for stochastic non-convex constrained optimization focus on the smooth problems. For example, [28] proposes a double-loop inexact augmented Lagrangian method for stochastic smooth nonconvex equality constrained optimization under a regularity assumption on the gradients of the constraints. A quadratic penalty method is developed by [29] under similar assumptions but uses only a single loop. Both [29] and [28] consider only smooth problems while we also consider non-smooth problems. For non-smooth but weakly convex problems, the existing methods can compute a nearly $\epsilon$-KKT point based on the Moreau envelopes of the original problem with iteration complexity $O(\epsilon^{-6})$ [9, 10, 11, 26, 8]. The penalty-based method by [12] reduces the number of evaluations of the subgradients to $O(\epsilon^{-4})$ but still needs to evaluate $O(\epsilon^{-6})$ function values. In contrast, our method only needs complexity $O(\epsilon^{-5})$ to compute a nearly $\epsilon$-KKT point. For smooth constrained optimization with non-convex equality constraints, variance-reduced stochastic gradient methods have been developed based on quadratic penalty [29, 30]. One can convert inequality constraints into equality constraints by adding slack variables $s^2$ or $s$ (requiring $s \geq 0$) to each inequality constraint [30]. However, adding $s^2$ will introduce spurious stationary points [31, 32], while adding $s \geq 0$ will make the domain or the constraint functions unbounded which violate the assumptions in [29, 30]. More importantly, we consider non-smooth constraint functions while [29, 30] only consider smooth cases.

## 3 Preliminaries

Let $\| \cdot \|$ be the Euclidean norm. For $g(\cdot) : \mathbb{R}^d \rightarrow \mathbb{R}^{d_1}$, let $\nabla g(\cdot) \in \mathbb{R}^{d \times d_1}$ be its Jacobian and let $\|\nabla g(\cdot)\| = \max_{\mathbf{u} \in \mathbb{R}^d, \|\mathbf{u}\|=1} \|\nabla g(\cdot)\mathbf{u}\|$ be the operator norm. Let $g_i(\mathbf{w}, \mathcal{B}) = \frac{1}{|\mathcal{B}|} \sum_{\xi \in \mathcal{B}} g_i(\mathbf{w}, \xi), i = 1, 2, \cdots, n$ be an estimator of $g_i$ based on samples from a minibatch $\mathcal{B}$. A function $f(\cdot)$ is monotonically non-decreasing if for any $\mathbf{x}_1, \mathbf{x}_2 \in \mathbb{R}^{d_1}$ satisfying that

$\mathbf{x}_{1,k} \leq \mathbf{x}_{2,k}, k \in [d_1]$ where $\mathbf{x}_k$ denote the $k$-th element of $\mathbf{x}$, it holds that $f(\mathbf{x}_1) \leq f(\mathbf{x}_2)$. A function $f(\cdot)$ is $\rho$-weakly convex for $\rho \geq 0$ if $f(\cdot) + \frac{\rho}{2}\|\cdot\|^2$ is convex. A function $g(\cdot)$ is $L$-smooth for $L \geq 0$ if $g$ is differentiable and $\nabla g(\cdot)$ is $L$-Lipschitz continuous. A vector-valued mapping $g(\cdot)$ is $L$-smooth for $L \geq 0$ if $g$ is differentiable at each component and $\nabla g(\cdot)$ is $L$-Lipschitz continuous w.r.t to the operator norm. Given a $\rho$-weakly convex function $f(\cdot)$, we denote its regular subdifferential as $\partial f(\cdot)$. For a $\rho$-weakly convex function $f(\cdot)$ and a constant $\lambda \in (0, \rho^{-1})$, the proximal mapping and the Moreau envelope of $f(\cdot)$ with parameter $\lambda$ are defined as follows, respectively,

$$\mathrm{prox}_{\lambda f}(\mathbf{u}) := \arg\min_{\mathbf{v}} f(\mathbf{v}) + \frac{1}{2\lambda}\|\mathbf{u}-\mathbf{v}\|^2 \ \text{ and } \ f_\lambda(\mathbf{u}) := \min_{\mathbf{v}} f(\mathbf{v}) + \frac{1}{2\lambda}\|\mathbf{u}-\mathbf{v}\|^2.$$

It is known that, for $\lambda \in (0, \rho^{-1})$, $f_\lambda(\cdot)$ is $L_f := \max\{\frac{1}{\lambda}, \frac{\rho}{1-\lambda\rho}\}$-smooth [33] and

$$\nabla f_\lambda(\mathbf{u}) = \frac{\mathbf{u} - \mathrm{prox}_{\lambda f}(\mathbf{u})}{\lambda} \in \partial f(\mathrm{prox}_{\lambda f}(\mathbf{u})). \tag{2}$$

For a non-smooth weakly convex objective, we follow existing works [21] [6] and aim to find a nearly $\epsilon$-stationary solution defined below.

**Definition 3.1.** A solution $\mathbf{w}$ is an $\epsilon$-stationary solution of (1) if $\mathrm{dist}(0, \partial F(\mathbf{w})) \leq \epsilon$. A solution $\mathbf{w}$ is a nearly $\epsilon$-stationary solution of (1) if there exists $\mathbf{w}'$ such that $\|\mathbf{w}-\mathbf{w}'\| \leq \epsilon$ and $\mathrm{dist}(0, \partial F(\mathbf{w}')) \leq \epsilon$. We make the following assumption on (1) throughout the paper.

**Assumption 3.2.** We assume that $f_i$ is $C_f$-Lipschitz continuous, $g_i$ is $C_g$-Lipschitz continuous for $i = 1, \ldots, n$, $\mathrm{prox}_{\lambda f_i}(\mathbf{w})$ and its subgradient are easily computable, and one of the following conditions hold:

- A1. $f_i$ is $\rho_f$-weakly convex, and $g_i(\mathbf{w})$ is $L_g$-smooth for all $i$.
- A2. $f_i$ is convex, and $g_i(\mathbf{w})$ is $L_g$-smooth for all $i$.
- A3. $f_i$ is convex and monotonically non-decreasing, and $g_i(\mathbf{w})$ is $\rho_g$-weakly convex for all $i$.

**Remark**: The Lipschitz continuity of $f$ and $g$ in Assumption 3.2 are fairly standard for FCCO problems. Since we target at non-smooth $f$ and $g$, Lipchitz continuity is a minimal assumption. In the considered applications of GDRO and learning with fairness constraints in section 6, $f$ is hinge and hence Lipschitz continuous.

## 4 Smoothing of Non-smooth FCCO

We first describe the main idea of our methods and then present detailed algorithms and their convergence. Under the Assumption 3.2, we can show $F$ is also weakly convex. To improve the convergence rates and accommodate deep learning applications, we develop stochastic momentum methods, which can be easily modified to incorporate with adaptive step sizes (cf. Appendix E). The challenges of developing provable stochastic momentum methods for solving (1) lie at the non-smoothness of $f_i$ and potentially $g_i$.

Let us first consider smooth $g_i$ and defer the discussion for weakly convex $g_i$ to subsection 4.2. The key idea is to use a smoothed version of $f_i$, denoted by $f_{i,\lambda}$, and to approximate (1) by

$$\min_{\mathbf{w} \in \mathbb{R}^d} F_\lambda(\mathbf{w}) := \frac{1}{n} \sum_{i=1}^n f_{i,\lambda}(g_i(\mathbf{w})). \tag{3}$$

We can easily show that when $g_i$ is $L_g$-smooth, then $F_\lambda(\mathbf{w})$ is $L_F$-smooth with $L_F = C_g^2 \max\{\frac{1}{\lambda}, \frac{\rho_f}{1-\lambda\rho_f}\} + C_f L_g$ (cf. Lemma A.1). As a result, the problem becomes a smooth FCCO where both inner functions and outer functions are smooth, which makes it possible to employ existing techniques to develop stochastic momentum methods to find an $\epsilon$-stationary point of (3), which is a point $\mathbf{w}$ satisfying $\mathbb{E}[\|\nabla F_\lambda(\mathbf{w})\|] \leq \epsilon$. However, what this implies for solving the original problem and what constitutes a good choice of $\lambda$ remain unclear. Below, we present a theory to address these questions. We introduce the following notion of stationarity of the original problem.

**Definition 4.1.** A solution $\mathbf{w}$ is an approximate $\epsilon$-stationary solution of (1) if there exist $\mathbf{t}_1, \ldots, \mathbf{t}_m$ and $\mathbf{y}_i \in \partial f_i(\mathbf{t}_i)$ for $i = 1, \ldots, n$ such that $\|\mathbf{t}_i - g_i(\mathbf{w})\| \leq \epsilon$, $i = 1, \ldots, n$, and $\left\|\frac{1}{n}\sum_{i=1}^n \nabla g_i(\mathbf{w})\mathbf{y}_i\right\| \leq \epsilon$.

This definition implies that when $\epsilon \to 0$, the solution converges to a stationary solution of $F(\cdot)$. Given this definition, our first theorem is stated below whose proof is given in Section A.3.

**Algorithm 1** SONEX for solving (3)

1: **Input:** $T, \lambda, \mathbf{w}_0 = \mathbf{w}_{-1}, \mathbf{v}_0, \eta, \gamma, \gamma', \beta, \forall t.$
2: Sample a batch of data $\mathcal{B}_{i,2}$ from the distribution of $\xi_i$ for $i = 1, \ldots, n.$
3: Set $u_{i,0} = g_i(\mathbf{w}_0, \mathcal{B}_{i,2})$ for $i = 1, \ldots, n.$
4: **for** $t = 0, 1, \cdots, T - 1$ **do**
5:     Sample $\mathcal{B}_1^t \subset \{1, \ldots, n\}$ and a batch of data $\mathcal{B}_{i,2}^t$ from the distribution of $\xi_i$ for $i \in \mathcal{B}_1^t$.
6:     **for** $i = 1, \ldots, n$ **do**
7:         Update

$$u_{i,t+1} = \begin{cases} (1 - \gamma)u_{i,t} + \gamma g_i(\mathbf{w}_t, \mathcal{B}_{i,2}^t) + \gamma'(g_i(\mathbf{w}_t, \mathcal{B}_{i,2}^t) - g_i(\mathbf{w}_{t-1}, \mathcal{B}_{i,2}^t)) & \text{if } i \in \mathcal{B}_1^t \\ u_{i,t} & \text{if } i \notin \mathcal{B}_1^t \end{cases}$$

8:     **end for**
9:     Compute $G_t = \frac{1}{|\mathcal{B}_1^t|} \sum_{i \in \mathcal{B}_1^t} \nabla f_{i,\lambda}(u_{t,i}) \nabla g_i(\mathbf{w}_t, \mathcal{B}_{i,2}^t)$
10:    Update $\mathbf{v}_{t+1} = (1 - \beta)\mathbf{v}_t + \beta G_t$
11:    Update $\mathbf{w}_{t+1} = \mathbf{w}_t - \eta \mathbf{v}_{t+1}$ or Adam-type update
12: **end for**
13: **Output:** $\mathbf{w}_\tau$ with $\tau$ randomly sampled from $\{1, 2, \cdots, T\}$

**Theorem 4.2.** *If $\mathbf{w}$ is an $\epsilon$-stationary solution to $F_\lambda(\cdot)$ with $\lambda = \epsilon/C_f$ such that $\|\nabla F_\lambda(\mathbf{w})\| \leq \epsilon$, then $\mathbf{w}$ is an approximate $\epsilon$-stationary solution to the original objective $F(\cdot)$.*

A remaining question is whether an approximate $\epsilon$-stationary point is also a nearly $\epsilon$-stationary solution of the original problem (1). We present a theorem below under the following assumption.

**Assumption 4.3** (EVLB assumption). There exist $c > 0$ and $\delta > 0$ such that, if $\mathbf{w}$ is an $\epsilon$-stationary point of $F_\lambda$ with $\lambda = \epsilon/C_f$ and $\epsilon \leq c$, it holds that $\lambda_{\min}\left(\nabla \mathbf{g}(\mathbf{w})\nabla \mathbf{g}(\mathbf{w})^\top\right) \geq \delta$, where $\nabla \mathbf{g}(\mathbf{w}) = [\nabla g_1(\mathbf{w})^\top, \ldots, \nabla g_n(\mathbf{w})^\top]^\top$

**Remark:** In Assumption 4.3, once $\epsilon$ is smaller than $c$, the lower bound $\delta$ does not depend on $\epsilon$. The empirical justification for this assumption is provided in subsection F.5.

**Theorem 4.4.** *Suppose Assumptions 3.2 (A1) and 4.3 hold. If $\mathbf{w}$ is an approximate $\epsilon$-stationary solution of (1) with $\epsilon \leq \min\left\{c, \frac{\delta^2}{16nC_g^2 L_g}\right\}$, $\mathbf{w}$ is also a nearly $\epsilon$-stationary solution of (1).*

**Remark:** The proof is given in Appendix A.4. It is notable that the Assumption 4.3 is easily verifiable (cf. Appendix F.5 for empirical evidence). In addition, when we consider the applications in non-convex constrained optimization, this condition is commonly used in existing analyses [7, 8].

### 4.1 Single-loop Methods for Smooth Inner Functions

Next, we present a single-loop algorithm for finding an $\epsilon$-stationary point of (3). The key ingredients of the algorithm include two parts: (1) maintaining and updating $n$ sequences $u_{i,t}$ for tracking each $g_i(\mathbf{w}_t), i = 1, \ldots, n$, which are updated in a coordinate-wise manner; (2) a stochastic momentum update. We present the detailed updates in Algorithm 1, which is referred to as SONEX. We note that the algorithm is inspired by existing stochastic momentum methods for smooth FCCO. First, Step 7 for updating $u_{i,t+1}$ is from the MSVR algorithm [16]. Different from MSVR, we directly utilize the stochastic momentum update in Step 10, 11, which are similar to SOX [2]. In contrast, MSVR also leverages a variance reduction technique (STROM) to compute an estimate of the stochastic gradient, which requires a stronger assumption that $\nabla g_i(\mathbf{w}; \xi)$ is Lipschitz continuous in expectation. Our analysis shows that this is not helpful for improving the convergence, as the complexity will be dominated by a term related to the smoothness of $f_{i,\lambda}$, which is in the order of $O(1/\epsilon)$.

We assume the following conditions of the stochastic estimators of $g_i$ and their gradients.

**Assumption 4.5.** There exist constants $\sigma_0 \geq 0$ and $\sigma_1 \geq 0$ such that the following statements hold for $g_i(\mathbf{w})$ and $g_i(\mathbf{w}, \xi_i)$: for $i = 1, \ldots, n$ and any $\mathbf{w} \in \mathbb{R}^d$

$$\mathbb{E}\|g_i(\mathbf{w}, \xi_i) - g_i(\mathbf{w})\|^2 \leq \sigma_0^2, \mathbb{E}\|\partial g_i(\mathbf{w}, \xi_i) - \partial g_i(\mathbf{w})\|^2 \leq \sigma_1^2$$

**Assumption 4.6.** There exist a constant $C_g$ such that $\mathbb{E}_\xi\|g_i(\mathbf{w}, \xi) - g_i(\mathbf{w}', \xi)\|_2 \leq C_g\|\mathbf{w} - \mathbf{w}'\|_2.$

It is notable that these assumptions have been made in [6] which is important for analyzing variance reduction technique such as MSVR. We refer to Assumption 4.6 as mean Lipchitz continuity of zero-order (MLC0). Let $B_1$ and $B_2$ denote outer and inner batch sizes.

**Theorem 4.7.** *Under Assumption 3.2 (A1), 4.5 and 4.6, by setting* $\lambda = \Theta(\epsilon), \beta = \Theta(\min\{B_1, B_2\}\epsilon^2), \gamma = \Theta(B_2\epsilon^4), \eta = \Theta(\frac{B_1\sqrt{B_2}}{n}\epsilon^3)$, *SONEX with* $\gamma' = 1 - \gamma + \frac{n-B_1}{B_1(1-\gamma)}$ *and* $\gamma \leq \frac{1}{2}$ *converges to an approximate* $\epsilon$*-stationary solution of* (1) *within* $T = O(\frac{n}{B_1\sqrt{B_2}}\epsilon^{-5})$ *iterations.*

Combining the above theorem with Theorem 4.4, we obtain the following guarantee:

**Corollary 4.8.** *Under Assumption 3.2(A1), 4.3, 4.5 and 4.6, with the same setting as in Theorem 4.7, SONEX converges to a nearly* $\epsilon$*-stationary solution of* (1) *within* $T = O(\frac{n}{B_1\sqrt{B_2}}\epsilon^{-5})$ *iterations.*

We compare our proposed SONEX with SONX for solving (1). Under assumption 4.6, the rate of SONX is $O(1/\epsilon^6)$ which is worse than our result of $O(1/\epsilon^5)$ in Corollary 4.8.

### 4.2 A Double-loop Algorithm for Smooth or Weakly-Convex Inner Functions

In this subsection, we further improve our results to achieve the same convergence rate of $O(1/\epsilon^5)$ by (1) removing the MCL0 assumption instead assuming that $f_i$ are convex; (2) designing a stochastic momentum method with a convergence guarantee for weakly convex $g_i$. It is worth mentioning that the convexity of $f_i$ holds for a broad range of real applications such as group DRO and the application in non-convex constrained optimization as discussed in next section. Before introducing our new algorithm, we need to first reformulate the problem.

Let us first consider the scenario that satisfies Assumption 3.2 (A3). For simplicity of notation, we denote $f_{i,\lambda}$ by $\bar{f}_i$. When $f_i$ is convex, we cast the problem (3) into a minimax formulation:

$$\min_{\mathbf{w}\in\mathbb{R}^d} F_\lambda(\mathbf{w}) := \frac{1}{n}\sum_{i=1}^n \max_{\mathbf{y}_i\in\mathbb{R}^{d_1}} \left\{\mathbf{y}_i^\top g_i(\mathbf{w}) - \bar{f}_i^*(\mathbf{y}_i)\right\}, \tag{4}$$

where $\bar{f}_i^*$ is the conjugate function of $\bar{f}_i$. In Appendix A.2, we show that this is also equivalent to the classical Nesterov's smoothing [19]. Under Assumption 3.2 (A3), $\bar{f}_i$ is also $C_f$-Lipschitz continuous, convex, non-decreasing because of (2). As a consequence, $\text{dom}(\bar{f}_i^*) \subset \{\mathbf{y}_i \in \mathbb{R}_+^{d_1} | \|\mathbf{y}_i\| \leq C_f\}$. This implies that, for any $\mathbf{y}_i \in \text{dom}(\bar{f}_i^*)$, function $\mathbf{y}_i^\top g_i(\mathbf{w})$ is $\|\mathbf{y}_i\|_1 \rho_g$-weakly convex in $\mathbf{w}$ and also $\sqrt{d_1}C_f\rho_g$-weakly convex because $\|\mathbf{y}_i\|_1 \leq \sqrt{d_1}\|\mathbf{y}_i\| \leq \sqrt{d_1}C_f$. This further implies that $F_\lambda(\cdot)$ is $\rho_{F_\lambda} := \sqrt{d_1}C_f\rho_g$-weakly convex.

Different from last subsection, there is another challenge we need to deal with in order to develop a stochastic momentum method, which comes from that $g_i$ are non-smooth. To tackle this challenge, we use another Moreau envelope smoothing of $F_\lambda(\cdot)$ and solve the following problem:

$$\min_{\mathbf{w}\in\mathbb{R}^d} \left\{F_{\lambda,\nu}(\mathbf{w}) := \min_{\mathbf{z}\in\mathbb{R}^d} \left\{F_\lambda(\mathbf{z}) + \frac{1}{2\nu}\|\mathbf{z} - \mathbf{w}\|^2\right\}\right\} \tag{5}$$

$$= \min_{\mathbf{w}\in\mathbb{R}^d} \left\{\min_{\mathbf{z}\in\mathbb{R}^d} \max_{\mathbf{y}\in\mathbb{R}^{nd_1}} \frac{1}{n}\sum_{i=1}^n \mathbf{y}_i^\top g_i(\mathbf{z}) - \bar{f}_i^*(\mathbf{y}_i) + \frac{1}{2\nu}\|\mathbf{z} - \mathbf{w}\|^2\right\}, \tag{6}$$

where $\nu \in (0, \rho_{F_\lambda}^{-1})$ and $\mathbf{y} = [\mathbf{y}_1^\top, \ldots, \mathbf{y}_n^\top]^\top$. The benefit of doing this is that (i) the resulting objective $F_{\lambda,\nu}(\mathbf{w})$ is smooth (cf. Lemma C.2), which allows us to employ stochastic momentum method; (ii) the inner $\min_\mathbf{z}\max_\mathbf{y}$ becomes a strongly-convex and strongly-concave problem due to that $\bar{f}_i$ is smooth and its conjugate is strongly convex, and $\nu \in (0, \rho_{F_\lambda}^{-1})$ (cf. Lemma C.3).

Recall that $\nabla F_{\lambda,\nu}(\mathbf{w}) = \frac{1}{\nu}(\mathbf{w} - \text{prox}_{\nu F_\lambda}(\mathbf{w}))$. Given an approximate solution of $\text{prox}_{\nu F_\lambda}(\mathbf{w})$, denoted by $\hat{\mathbf{z}}$, of the inner minimization problem in (5), we can use $\frac{1}{\nu}(\mathbf{w} - \hat{\mathbf{z}})$ as an inexact gradient $F_{\lambda,\nu}(\mathbf{w})$ to update $\mathbf{w}$ with a momentum method in order to solve (5). To obtain an estimate $\hat{\mathbf{z}}_t$ at the $t$-th iteration, we employ a recent algorithm ALEXR [5] for solving the inner minimax problem. With an estimate $\hat{\mathbf{z}}_t$, we will update $\mathbf{w}_{t+1}$ by a momentum update. We present the detailed steps of our double-loop algorithm named ALEXR2 in Algorithm 2, where $U_{\bar{f}_i^*}(\mathbf{y}_i, \mathbf{y}_i')$ is the Bregman divergence induced by $\bar{f}_i^*$, namely, $U_{\bar{f}_i^*}(\mathbf{y}_i, \mathbf{y}_i') := \bar{f}_i^*(\mathbf{y}_i) - \bar{f}_i^*(\mathbf{y}_i') - \langle\partial\bar{f}_i^*(\mathbf{y}_i'), \mathbf{y}_i - \mathbf{y}_i'\rangle$. For the sake of convergence analyses we need $U_{\bar{f}_i^*}$ to be bounded, as stated in the assumption below.

**Algorithm 2** ALEXR2 for solving (6)

---

1: **Input:** $T$, $\mathbf{w}_0 \in \mathbb{R}^d$, $\mathbf{v}_0 = 0$, $K_t$, $\alpha, \beta, \nu, \eta, \theta, \gamma > 0$
2: **for** $t = 0, 1, \ldots, T - 1$ **do**
3:      Set $\mathbf{z}_{t,0} = \mathbf{z}_{t,-1} = \mathbf{w}_t$ and initialize $\mathbf{y}_{t,0} = [\mathbf{y}_{t,0}^{(1)\top}, \ldots, \mathbf{y}_{t,0}^{(n)\top}]^\top \in \mathbb{R}^{nd_1}$.
4:      **for** $k = 0, 1, \ldots, K_t - 1$ **do**
5:          Sample $\mathcal{B}_1^{t,k} \subset \{1, \ldots, n\}$ and two independent batches of data $\mathcal{B}_{i,2}^{t,k}$ and $\tilde{\mathcal{B}}_{i,2}^{t,k}$ from the distribution of $\xi_i$ for $i \in \mathcal{B}_1^{t,k}$.
6:          Compute $\tilde{g}_{t,k}^{(i)} = g_i(\mathbf{z}_{t,k}, \mathcal{B}_{i,2}^{t,k}) + \theta(g_i(\mathbf{z}_{t,k}, \mathcal{B}_{i,2}^{t,k}) - g_i(\mathbf{z}_{t,k-1}, \mathcal{B}_{i,2}^{t,k}))$ for $i \in \mathcal{B}_1^{t,k}$.
7:          **for** $i = 1, \ldots, n$ **do**
8:
$$\mathbf{y}_{t,k+1}^{(i)} = \begin{cases} \arg\max_{\mathbf{y}^{(i)}} \left\{ \mathbf{y}^{(i)\top} \tilde{g}_{t,k}^{(i)} - \bar{f}_i^*(\mathbf{y}^{(i)}) - \frac{1}{\gamma} U_{\bar{f}_i^*}(\mathbf{y}^{(i)}, \mathbf{y}_{t,k}^{(i)}) \right\} & \text{if } i \in \mathcal{B}_1^{t,k} \\ \mathbf{y}_{t,k}^{(i)} & \text{if } i \notin \mathcal{B}_1^{t,k} \end{cases}$$

9:          **end for**
10:          Compute $G_{t,k} = \frac{1}{|\mathcal{B}_1^{t,k}|} \sum_{i \in \mathcal{B}_1^{t,k}} [\partial g_i(\mathbf{z}_{t,k}, \tilde{\mathcal{B}}_{i,2}^{t,k})]^\top \mathbf{y}_{t,k+1}^{(i)}$
11:          Update $\mathbf{z}_{t,k+1} = \arg\min_{\mathbf{z} \in \mathbb{R}^d} \left\{ \langle G_{t,k}, \mathbf{z} \rangle + \frac{1}{2\nu} \|\mathbf{z} - \mathbf{w}_t\|^2 + \frac{1}{2\eta} \|\mathbf{z} - \mathbf{z}_{t,k}\|^2 \right\}$
12:      **end for**
13:      Let $G_t = \frac{\beta}{\nu}(\mathbf{w}_t - \mathbf{z}_{t,K_t})$
14:      Update $\mathbf{v}_{t+1} = (1 - \beta)\mathbf{v}_t + \beta G_t$
15:      Update $\mathbf{w}_{t+1} = \mathbf{w}_t - \alpha \mathbf{v}_{t+1}$ or use Adam-type update
16: **end for**
17: **Output:** $\mathbf{w}_\tau$ with $\tau$ randomly sampled from $\{1, 2, \ldots T\}$

---

Besides, we would like to point out that the sequences of $y_i$ can be updated similar to $u_i$ sequences in SONEX similar to [5], i.e., $y_{t,k}^i = \nabla f_{i,\lambda}(u_{t,k}^i)$, where $u_{t,k+1}^i = (1 - \hat{\gamma})u_{t,k}^i + \hat{\gamma}g_i(\mathbf{z}_{t,k}; \mathcal{B}_{i,2}^{t,k}) + \hat{\gamma}\theta(g_i(\mathbf{z}_{t,k}; \mathcal{B}_{i,2}^{t,k}) - g_i(\mathbf{z}_{t,k-1}; \mathcal{B}_{i,2}^{t,k}))$, where $\hat{\gamma} = \gamma/(1 + \gamma)$.

**Assumption 4.9.** We assume that $f_i$ is a function s.t. $U_{\bar{f}^*}(y_1, y_2)$ is bounded for any $y_1, y_2 \in dom(\bar{f}_i^*) \subset \mathbb{R}^{d_1}$.

**Remark**: We point out that this condition is not strong under our setting where $\bar{f}_i$ is lipschitz continuous and $dom(\bar{f}_i^*)$ is consequently bounded and it is satisfied by many practical convex lipschitz-continuous functions such as hinge function, smoothed hinge function(i.e. $f_{i,\lambda}$ or $\bar{f}_i$ in this paper), etc.

The following theorem states the convergence of ALEXR2 with proof given in Section C.

**Theorem 4.10.** *Suppose assumption 3.2 (A3), 4.5 and 4.9 hold and $\lambda = \epsilon/C_f$ in (4). For any $\epsilon > 0$, there exists $\theta \in (0, 1)$ with $1 - \theta = O(\epsilon^2)$ such that, by setting $\eta = \frac{1-\theta}{\theta}L_f$ and $\gamma = \frac{(1-\theta)n}{B_1}$, $\beta \leq \frac{1}{2}$, $\alpha = \frac{\beta}{2L_{F_{\lambda,\nu}}}$ and $K_t = \tilde{O}\left(\frac{n}{B_1 B_2 \epsilon^3} + \frac{1}{\epsilon^3}\right)$, ALEXR2 returns $\mathbf{w}_\tau$ as a nearly $\epsilon$-stationary solution of (4) in expectation within $T = O(\epsilon^{-2})$ and a total iteration complexity of $\tilde{O}\left(\frac{n}{B_1 B_2 \epsilon^5} + \frac{1}{\epsilon^5}\right)$.*

**Remark:** We can easily extend the above result to Assumption 3.2 (A2). When $g_i$ are smooth, we do not need the monotonicity of $f_i$ as the minimax problem $\min_{\mathbf{z}} \max_{\mathbf{y}}$ is still guaranteed to be strongly-convex and strongly-concave when $\nu \in (0, \rho_{F_\lambda}^{-1})$, where $\rho_{F_\lambda} := \sqrt{d_1} C_f L_g$ is the weak convexity parameter of $F_\lambda$.

Finally, we show that when $g_i$ are smooth, we can also obtain a nearly $\epsilon$-stationary solution to the original problem (1). Different from the result in Theorem 4.7, the above theorem only guarantees a nearly $\epsilon$-stationary solution to $F_\lambda$. To address this gap, we can recover a nearly $\epsilon$-stationary solution to (1) from $\mathbf{w}_\tau$ by running ALEXR starting with $\mathbf{w}_\tau$ with $K = \tilde{O}(\epsilon^{-5})$ iterations. This result is stated as the following corollary with the proof given in Section C.

**Corollary 4.11.** *Suppose assumption 3.2 (A2), 4.3, 4.5, 4.9 hold. For any $\epsilon \leq \min\left\{c, \frac{\delta^2}{16nC_g^2 L_g}\right\}$, let $\hat{\mathbf{w}}_\tau =$ ALEXR($\mathbf{w}_\tau, K$) with $K = \tilde{O}\left(\frac{n}{B_1 B_2 \epsilon^5} + \frac{1}{\epsilon^5}\right)$, $\eta = \frac{1-\theta}{\theta}L_f$ and $\gamma = \frac{(1-\theta)n}{B_1}$ for $\theta \in (0, 1)$*

*with $1 - \theta = O(\epsilon^4)$. $\hat{\mathbf{w}}_\tau$ is a nearly $\epsilon$-stationary solution of* (1) *and it is found with a total iteration complexity of* $\tilde{O}(\frac{n}{B_1 B_2 \epsilon^5} + \frac{1}{\epsilon^5})$.

# 5 Smoothed Hinge Penalty Method for Constrained Optimization

In this section, we consider a constrained optimization problem with $m > 1$ inequality constraints:

$$\min_{\mathbf{w}} g_0(\mathbf{w}) \text{ s.t. } g_i(\mathbf{w}) \leq 0, i = 1, \cdots, m \tag{7}$$

where $g_i : \mathbb{R}^d \to \mathbb{R}$ is continuous and satisfies $g_i(\mathbf{w}) = \mathbb{E} g_i(\mathbf{w}, \xi_i)$ and the expectation is taken over the random variable $\xi_i$ for $i = 0, 1, \ldots, m$. Following [8], we consider an exact penalty method for (7) by solving the following unconstrained problem:

$$\min_{\mathbf{w} \in \mathbb{R}^d} \Phi(\mathbf{w}) := g_0(\mathbf{w}) + \frac{\rho}{m} \sum_{i=1}^m [g_i(\mathbf{w})]_+ \tag{8}$$

where $\rho$ is a sufficiently large number. Let $f(\cdot) := \rho[\cdot]_+$ so the penalty term $\frac{\rho}{m} \sum_{i=1}^m [g_i(\mathbf{w})]_+$ has the same structure as $F(\mathbf{w})$ in (1). Yang et al. [8] have employed SONX for solving the above problem and established a complexity of $O(1/\epsilon^6)$ for finding a nearly $\epsilon$-KKT solution. Below, we apply the smoothing idea and ALEXR2 or SONEX for improving the convergence rate. Our key idea is to optimize an outer smoothed problem:

$$\min_{\mathbf{w} \in \mathbb{R}^d} \Phi_\lambda(\mathbf{w}) := g_0(\mathbf{w}) + \frac{1}{m} \sum_{i=1}^m f_\lambda(g_i(\mathbf{w})) = g_0(\mathbf{w}) + \frac{1}{m} \sum_{i=1}^m \max_{y_i \in [0, \rho]} \left\{ y_i g_i(\mathbf{w}) - \frac{\lambda}{2} y_i^2 \right\}, \tag{9}$$

where $f_\lambda(z) := \min_{z' \in \mathbb{R}} f(z') + \frac{1}{2\lambda}(z - z')^2$. Except for the term $g_0$, the objective function of (9) has the same structure as (3) and (4) with $\bar{f}_i^*(y_i) = \frac{\lambda}{2} y_i^2 + \mathbf{1}_{[0,\rho]}(y_i)$.

**Assumption 5.1.** There exists a constant $\delta > 0$ such that $\sigma_{min}(\mathbf{J}(\mathbf{w})) \geq \delta$ for any matrix $\mathbf{J}(\mathbf{w}) = [\mathbf{h}_1(\mathbf{w}), \ldots, \mathbf{h}_m(\mathbf{w})] \in \mathbb{R}^{d \times m}$ with $\mathbf{h}_i(\mathbf{w}) \in \partial g_i(\mathbf{w})$ and any $\mathbf{w}$ satisfying $\max_{i=1,\ldots,m} g_i(\mathbf{w}) > 0$.

Under this assumption, we have the following proposition whose proof is given in Section D.1.

**Proposition 5.2.** *Suppose Assumption 5.1 hold. If $\rho > \frac{m(C_g + 1)}{\delta}$ and $\lambda = \frac{\epsilon}{\rho}$, a nearly $\epsilon$-stationary solution $\mathbf{w}$ to* (9) *is also a nearly $O(\epsilon)$-KKT solution to the original problem* (7) *in the sense that there exist $\hat{\mathbf{w}}$ and $\nu_i \geq 0$ for $i = 1, \ldots, m$ such that $\|\mathbf{w} - \hat{\mathbf{w}}\| \leq \epsilon$, $dist(0, \partial g_0(\hat{\mathbf{w}}) + \sum_{i=1}^m \partial g_i(\hat{\mathbf{w}})\nu_i)) \leq O(\epsilon)$, $\max_{i=1,\ldots,m} g_i(\hat{\mathbf{w}}) \leq O(\epsilon)$ and $|g_i(\hat{\mathbf{w}})\nu_i| \leq O(\epsilon), \forall i = 1, 2, \cdots, m$.*

To find a nearly $\epsilon$-stationary solution to (9), we can adapt either SONEX if $g_i$ are smooth or ALEXR2 if $g_i$ are weakly convex with a minor change by including the stochastic gradient of $g_0$ in the calculation of $G_t$. The complexity of this approach is presented below with proofs given in Section D.2.

**Theorem 5.3.** *Suppose assumption 4.5 and 5.1 hold, and the stochastic (sub)gradient of $g_0$ has bounded variance. Let $\rho > m(C_g + 1)/\delta$ and $\lambda = \epsilon/\rho$.*

- *if $\{g_i\}_{i=0}^m$ are weakly convex, then with $\tilde{\mathcal{O}}\left(\frac{m}{B_1 B_2 \delta^4 \epsilon^5}\right)$ iterations ALEXR2 find a nearly $\epsilon$-KKT solution $\mathbf{w}$ for (7) satisfying that there exist $\hat{\mathbf{w}}$ and $\nu_i \geq 0$ for $i = 1, \ldots, m$ such that $\mathbb{E} \|\mathbf{w} - \hat{\mathbf{w}}\| \leq \epsilon$ and $\mathbb{E} dist(0, \partial g_0(\hat{\mathbf{w}}) + \sum_{i=1}^m \partial g_i(\hat{\mathbf{w}})\nu_i)) \leq O(\epsilon)$ and it holds with probability[4] $1 - O(\epsilon)$ that $\max_{i=1,\ldots,m} g_i(\hat{\mathbf{w}}) \leq O(\epsilon)$ and $|g_i(\hat{\mathbf{w}})\nu_i| \leq O(\epsilon), \forall i = 1, 2, \cdots, m$.*
- *if $\{g_i\}_{i=0}^m$ are smooth, then with $\tilde{\mathcal{O}}\left(\frac{m}{B_1 \sqrt{B_2} \delta^4 \epsilon^5}\right)$ iterations SONEX can find an $\epsilon$-KKT solution $\mathbf{w}$ for (7) similar as above except for $\hat{\mathbf{w}} = \mathbf{w}$.*

Notably, our method not only improve the rate but also enjoy an improved dependence on $\delta$, compared to $O(\epsilon^{-6} \delta^{-6})$ in prior work [8].

---

[4]If $|g_i(\mathbf{w})| < +\infty$ for any $i$ and $\mathbf{w}$, this high probability result can be replaced by $\mathbb{E}[\max_{i=1,\ldots,m} g_i(\hat{\mathbf{w}})] \leq O(\epsilon)$ and $\mathbb{E}[|g_i(\hat{\mathbf{w}})\nu_i|] \leq O(\epsilon), \forall i = 1, 2, \cdots, m$.

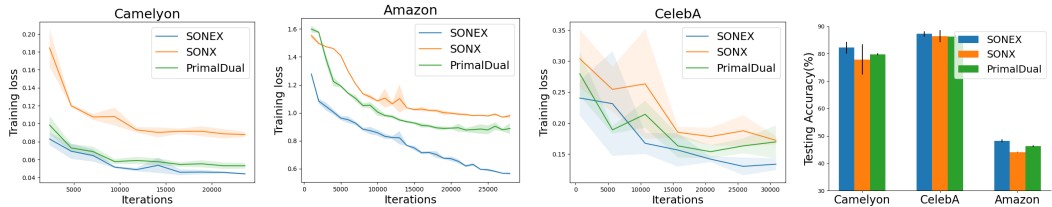

Figure 1: Training loss curves (left three) and testing accuracy (right one) of different methods for Group DRO with CVaR ratio $r = 0.15$ on different datasets.

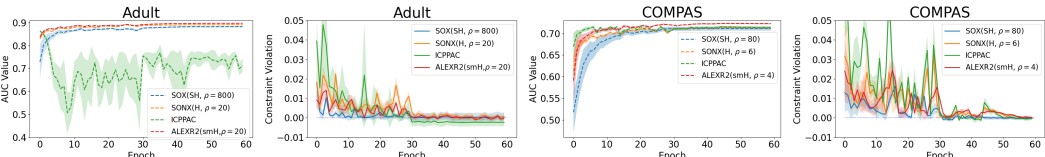

Figure 2: Training curves of AUC values (fig 1,3) and constraint violation (fig 2,4) of different methods. The format of legend is "Algorithm(penalty function, $\rho$)", and SH, H, smH mean square hinge, hinge and smoothed hinge, respectively.

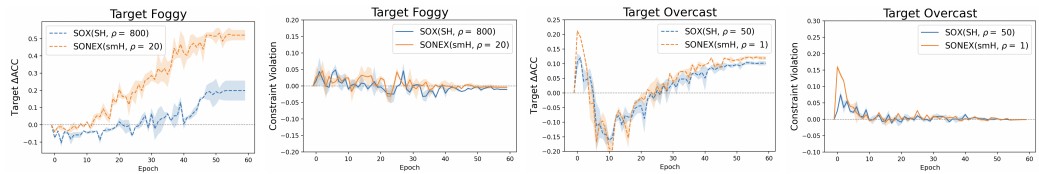

Figure 3: Training curves of Target $\Delta$ACC values (fig 1,3) and constraint violation (fig 2,4) of different methods. The format of label is "Algorithm(penalty function, $\rho$)", and SH, smH mean square hinge and smoothed hinge, respectively.

## 6 Numerical Experiments

We conduct experiments to verify the effectiveness of the proposed algorithms. For non-smooth non-convex FCCO, we consider GDRO with CVaR divergence [5]. For non-convex constrained optimization, we follow [8] and consider two tasks, namely AUC maximization with ROC fairness constraints and continual learning with non-forgetting constraints.

**GDRO with CVaR divergence.** We consider the following GDRO with CVaR divergence, which minimizes top-$k$ worst groups' losses [5]:

$$\min_{\theta} \min_{s} s + \frac{1}{n} \sum_{g=1}^{n} \frac{1}{r} [\mathbb{E}_{(\mathbf{x},y) \sim \mathcal{D}_g} \ell(\theta; (\mathbf{x}, y)) - s]_+ \tag{10}$$

where $n$ is the number of groups, $r = k/n$, and $\mathcal{D}_g$ denotes the data of the $g$-th group.

We use 3 datasets: Camelyon17, Amazon [34], and CelebA [35]. The first two datasets are from WILDS Benchmark for evaluating methods tackling the distributional shift [36], where Camelyon17 has 30 groups and Amazon has 1252 groups. CelebA is a large-scale facial attribute dataset containing over 200,000 celebrity images annotated with 40 binary attributes. We select 4 binary attributes 'Attractive', 'Mouth_Slightly_Open', 'Male' and 'Blonde_Hair' and construct 16 groups, where 'Blonde_Hair' also serves as the label for classification. We compare SONEX with SONX and Primal-Dual, where PrimalDual is a stochastic primal-dual algorithm [5]. We use pretrained Densenet121 [38], Distilbert [39] and ResNet50 [40] for Camelyon17, Amazon and CelebA, respectively. We perform Adam-type update for SONEX on Amazon dataset and momentum-type update on Camelyon17 and CelebA datasets. We perform Adam-type update for PrimalDual on Amazon dataset and SGD-type update on Camelyon17 and CelebA. We run each experiment for 3 random seeds and report their average performance. The hyperparameter tuning is discussed in Appendix F.1.

---

[5]Named 'OOA' in [5], it's derived by extending the algorithm in [37] to GDRO with CVaR setting

**Results.** We report our result under setting of $r = 0.15$ in Figure 1, where the first 3 figures are the loss curves on the training dataset while the last one is the test accuracy. Our experiment results shows that SONEX performs better than SONX and PrimalDual on GDRO tasks.

**AUC maximization with ROC fairness constraints.** The formulation is given in Appendix F.2, where the objective is AUC loss and constraints specify the tolerance of the gap between false positive rates (true positive rates) of two sensitive groups at different classification thresholds. For this experiment, we follow almost the same setting as in [8]. Two datasets are used, namely Adult [41] and COMPAS [42], which contain male/female, Caucasian/non-Caucasian groups, respectively. We set thresholds at $\Gamma = \{-3, -2, -1, \cdots, 3\}$ and the tolerance $\kappa = 0.005$, which gives us 14 constraints. We learn a simple neural network with 2 hidden layers. We compare ALEXR2 with the method in [7] that optimizes a squared-hinge penalty function with the SOX algorithm [2], the method in [8] that optimizes a hinge penalty function with the SONX algorithm [6], the double-loop method ICPPAC [10]. We perform Adam-type update for ALEXR2. We run each method for totally 60 epoches with a batch size of 128, repeat five times with different seeds and then report average of the AUC scores and constraint values. Hyperparameter tuning is presented in Appendix F.2.

**Results.** We compare the training curves of objective AUC values and the constraint violation measured by the worst constraint function value at each epoch for different methods in Figure 2 on different datasets. These results demonstrate that ALEXR2 optimizing smoothed hinge penalty function can better maximize AUC value on both datasets than the baseline methods while still having similar constraint satisfaction.

**Continual Learning with non-forgetting constraints.** We follow a similar experimental setup to [7] for fine-tuning a CLIP model for autonomous driving on the BDD100K dataset [43], a large-scale, multi-task driving image dataset. The formulation is given in Appendix F.3, where the objective is the contrastive loss on a target task (e.g., classifying foggy and overcast condition), and the constraints specify the logistic loss of the new model should not be larger than that of the old model for other different classes (e.g., clear, snowy, rainy, partly cloudy). Since the objective function is another FCCO, we modify SONEX such that the gradient estimator of the objective is computed similar as the smoothed square penalty function. We compare with SOX that optimizes the squared hinge loss use the Adam-type update for both methods. We do not compare with SONX that optimizes the hinge penalty function with SGD-type update as it fails for learning the Transformer network used in this experiment. Hyperparameter tuning is presented in Appendix F.3.

**Results.** We present training curves of accuracy improvement and constraint violation on different target tasks in Figure 3. It shows that our method can achieve higher accuracy on target tasks than the baseline method of using squared hinge function while retaining similar constraint satisfaction.

## 7  Conclusions and Discussion

In this paper, we have considered non-smooth non-convex finite-sum coupled compositional optimization, where the out functions are non-smooth and inner functions are smooth or weakly convex. We proposed stochastic momentum methods to improve the convergence rate and accelerate the training for deep neural networks. We also considered the applications of the proposed stochastic momentum methods to solving non-convex inequality constrained optimization problems and derived state-of-the-art results for finding an $\epsilon$-KKT solution. Our experiments demonstrate the effectiveness of our methods. One limitation of our results is that our single-loop method SONEX still has a worse complexity than the double-loop method ALEXR2 in terms of dependence on the inner batch size.

## Acknowledgments and Disclosure of Funding

We thank Quanqi Hu for the assistance in developing the codebase for group DRO. X. Chen, B. Wang, M. Yang and T. Yang were partially supported by the National Science Foundation Career Award 2246753, the National Science Foundation Award 2246757 and 2306572. Q. Lin was partially supported by National Science Foundation Award 2147253.

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

# A Proofs of Technical Lemmas

## A.1 Proof of Lemma A.1

**Lemma A.1.** *Under Assumption 3.2(A1), $F(\cdot)$ in (1) is $\rho_F$-weakly convex with $\rho_F = \sqrt{d_1}L_g C_f + \rho_f C_g^2$. If $g_i$ is $L_g$ smooth, then $F_\lambda(\mathbf{w})$ is $L_F$-smooth with $L_F = C_g^2 \max\{\frac{1}{\lambda}, \frac{\rho_f}{1-\lambda\rho_f}\} + C_f L_g$.*

*Proof.* Consider any $\mathbf{w}$ and $\mathbf{w}'$ in $\mathbb{R}^d$, any $i \in \{1, \ldots, n\}$ and any $\mathbf{v}_i \in \partial f_i(\mathbf{t}_i)$ at $\mathbf{t}_i = g_i(\mathbf{w})$. By Assumption 3.2(A1), $\mathbf{v}_i^\top g_i(\mathbf{w})$ is $\|\mathbf{v}_i\|_1 L_g$-smooth in $\mathbf{w}$ and $f_i$ is $\rho_f$-weakly convex, we have

$$f_i(g_i(\mathbf{w}')) \geq f_i(g_i(\mathbf{w})) + \mathbf{v}_i^\top (g_i(\mathbf{w}') - g_i(\mathbf{w})) - \frac{\rho_f}{2}\|g_i(\mathbf{w}') - g_i(\mathbf{w})\|^2$$

$$\geq f_i(g_i(\mathbf{w})) + \mathbf{v}_i^\top (g_i(\mathbf{w}') - g_i(\mathbf{w})) - \frac{\rho_f C_g^2}{2}\|\mathbf{w}' - \mathbf{w}\|^2$$

$$\geq f_i(g_i(\mathbf{w})) + \mathbf{v}_i^\top \nabla g_i(\mathbf{w}')(\mathbf{w}' - \mathbf{w}) - \frac{\|\mathbf{v}_i\|_1 L_g}{2}\|\mathbf{w}' - \mathbf{w}\|^2 - \frac{\rho_f C_g^2}{2}\|\mathbf{w}' - \mathbf{w}\|^2$$

$$\geq f_i(g_i(\mathbf{w})) + \mathbf{v}_i^\top \nabla g_i(\mathbf{w}')(\mathbf{w}' - \mathbf{w}) - \frac{\sqrt{d_1}C_f L_g + \rho_f C_g^2}{2}\|\mathbf{w}' - \mathbf{w}\|^2,$$

where the second and the last inequalities hold because of lipschitzness of $f_i$ and $g_i$ from Assumption 3.2, the third because $\mathbf{v}_i^\top g_i(\mathbf{w})$ is $\|\mathbf{v}_i\|_1 L_g$-smooth in $\mathbf{w}$.

By the $\rho_f$-weak convexity of $f_i$ and lemma 13 in [33], we know that $f_{i,\lambda}$ is $L_{f_\lambda} := \max\{\frac{1}{\lambda}, \frac{\rho_f}{1-\lambda\rho_f}\}$-smooth. Since $\nabla f_{i,\lambda}(\mathbf{w}) = (\mathbf{w} - \text{prox}_{\lambda f_i}(\mathbf{w}))/\lambda \in \partial f_i(\text{prox}_{\lambda f_i}(\mathbf{w}))$, we have $\|\nabla f_{i,\lambda}(\mathbf{w})\| \leq C_f$ by Assumption 3.2. Therefore, by Assumption 3.2 again, we can easily show that $\nabla f_{i,\lambda}(g_i(\mathbf{w}))\nabla g_i(\mathbf{w})$ is $(C_g^2 L_{f_\lambda} + C_f L_g)$-Lipschitz continuous (see Lemma 4.2 in [44]), so is $\nabla F_\lambda(\mathbf{w}) = \frac{1}{n}\sum_{i=1}^n \nabla f_{i,\lambda}(g_i(\mathbf{w}))\nabla g_i(\mathbf{w})$. $\qquad\square$

## A.2 Proof of Proposition A.2

**Proposition A.2.** *When $f_i$ is convex, the smoothed problem (3) is equivalent to the Nesterov's smoothing, i.e.,*

$$\min_{\mathbf{w}} \max_{\mathbf{y} \in \Omega^n} \frac{1}{n}\sum_{i=1}^n y_i^\top g_i(\mathbf{w}) - f_i^*(y_i) - \frac{\lambda}{2}\|y_i\|_2^2 \tag{11}$$

*where $\Omega = dom(f_i^*)$, $f_i^*(y_i) + \frac{\lambda}{2}\|y_i\|_2^2 = f_{i,\lambda}^*(y_i)$. In addition, $f_{i,\lambda}(g) \leq f_i(g) \leq f_{i,\lambda}(g) + \frac{\lambda C_f^2}{2}$.*

*Proof.* We prove by deriving (11) from (3).

$$F_\lambda(\mathbf{w}) = \frac{1}{n}\sum_{i=1}^n f_{i,\lambda}(g_i(\mathbf{w}))$$

$$= \min_{\mathbf{y} \in \Omega^n} \frac{1}{n}\sum_{i=1}^n f_i(\mathbf{y}_i) + \frac{1}{2\lambda}\|\mathbf{y}_i - g_i(\mathbf{w})\|^2$$

$$= \min_{\mathbf{y} \in \Omega^n} \max_{\mathbf{z} \in \mathcal{Z}^n} \frac{1}{n}\sum_{i=1}^n \mathbf{y}_i^\top \mathbf{z}_i - f_i^*(\mathbf{z}_i) + \frac{1}{2\lambda}\|\mathbf{y}_i - g_i(\mathbf{w})\|^2$$

$$= \max_{\mathbf{z} \in \mathcal{Z}^n} \min_{\mathbf{y} \in \Omega^n} \frac{1}{n}\sum_{i=1}^n \mathbf{y}_i^\top \mathbf{z}_i - f_i^*(\mathbf{z}_i) + \frac{1}{2\lambda}\|\mathbf{y}_i - g_i(\mathbf{w})\|^2$$

where the dual domian $\mathcal{Z} = \{\mathbf{z} : \mathbf{z} \in \partial f_i(\mathbf{t}), t \in dom f_i\} \subseteq \{\mathbf{z} : \|\mathbf{z}\| \leq C_f\}$ is bounded; the second equality is from definition of $f_{i,\lambda}$, the third equality holds from convexity of $f_i$ and the fourth equality holds from Sion's minimax theorem [45] and the convexity of $f_i^*$.

Note that the the inner-level minimization problem can be solved exactly as $\mathbf{y}_i^* = g_i(\mathbf{w}) - \lambda \mathbf{z}_i$ since it's a quadratic function, we plug $\mathbf{y}_i^*$ into the RHS, then we have:

$$F_\lambda(\mathbf{w}) = \max_{\mathbf{z} \in \mathcal{Z}^n} \frac{1}{n} \sum_{i=1}^n \mathbf{z}_i^\top g_i(\mathbf{w}) - f_i^*(\mathbf{z}_i) - \frac{\lambda}{2} \|\mathbf{z}_i\|^2 \tag{12}$$

Besides, note that $f^+(\mathbf{x}, \mathbf{u}) = f_i(\mathbf{u}) + \frac{1}{2\lambda} \|\mathbf{u} - \mathbf{x}\|^2$ is jointly convex, $f_{i,\lambda}(\mathbf{x}) = \min_{\mathbf{u}} f_i(\mathbf{u}) + \frac{1}{2\lambda} \|\mathbf{u} - \mathbf{x}\|^2$ is also convex. Then we have

$$\frac{1}{n} \sum_{i=1}^n f_{i,\lambda}(g_i(\mathbf{w})) = \frac{1}{n} \sum_{i=1}^n \mathbf{z}_i^\top g_i(\mathbf{w}) - f_{i,\lambda}^*(\mathbf{z}_i)$$

Compare this with 12 we have

$$f_{i,\lambda}^*(\mathbf{z}_i) = f_i^*(\mathbf{z}_i) + \frac{\lambda}{2} \|\mathbf{z}_i\|^2$$

Now we already have that

$$f_{i,\lambda}(g) = \max_{\mathbf{z}_i \in \mathcal{Z}} \mathbf{z}_i^\top g - f_i^*(\mathbf{z}_i) - \frac{\lambda}{2} \|\mathbf{z}_i\|^2 \text{ and } f_i(g) = \max_{\mathbf{z}_i \in \mathcal{Z}} \mathbf{z}_i^\top g - f_i^*(\mathbf{z}_i)$$

then $f_{i,\lambda}(g) \leq f_i(g)$ naturally holds. Besides,

$$f_i(g) = \max_{\mathbf{z}_i \in \mathcal{Z}} \mathbf{z}_i^\top g - f_i^*(\mathbf{z}_i) \leq \max_{\mathbf{z}_i \in \mathcal{Z}} \mathbf{z}_i^\top g - f_i^*(\mathbf{z}_i) - \frac{\lambda}{2} \|\mathbf{z}_i\|^2 + \max_{\mathbf{z}_i \in \mathcal{Z}} \frac{\lambda}{2} \|\mathbf{z}_i\|^2 \leq f_{i,\lambda}(g) + \frac{\lambda}{2} C_f^2$$

$\square$

### A.3 Proof of Theorem 4.2

*Proof of Theorem 4.2.* Suppose $\mathbf{w}$ is an $\epsilon$-stationary solution to $F_\lambda(\cdot)$ with $\lambda = \epsilon/C_f$. It holds that

$$\|\nabla F_\lambda(\mathbf{w})\| = \left\| \frac{1}{n} \sum_{i=1}^n \nabla f_{i,\lambda}(g_i(\mathbf{w})) \nabla g_i(\mathbf{w}) \right\| = \left\| \frac{1}{n\lambda} \sum_{i=1}^n (g_i(\mathbf{w}) - \mathrm{prox}_{\lambda f_i}(g_i(\mathbf{w}))) \nabla g_i(\mathbf{w}) \right\| \leq \epsilon.$$

By (2) and $C_f$-Lipschitz continuity of $f_i$, we have $\|g_i(\mathbf{w}) - \mathrm{prox}_{\lambda f_i}(g_i(\mathbf{w}))\| \leq \lambda C_f \leq \epsilon$. Let $\mathbf{t}_i = \mathrm{prox}_{\lambda f_i}(g_i(\mathbf{w}))$ and $\mathbf{y}_i = (g_i(\mathbf{w}) - \mathrm{prox}_{\lambda f_i}(g_i(\mathbf{w})))/\lambda \in \partial f_i(\mathrm{prox}_{\lambda f_i}(g_i(\mathbf{w}))) = \partial f_i(\mathbf{t}_i)$. It holds that $\|\mathbf{t}_i - g_i(\mathbf{w})\| \leq \epsilon$, $i = 1, \ldots, n$, and $\left\| \frac{1}{n} \sum_{i=1}^n \nabla g_i(\mathbf{w}) \mathbf{y}_i \right\| \leq \epsilon$, so $\mathbf{w}$ is an approximate $\epsilon$-stationary solution to the original objective $F(\cdot)$ by definition. $\square$

### A.4 Proof of Theorem 4.4

We need the following lemma.

**Lemma A.3.** *Suppose Assumption 3.2 (A1) or (A2) holds. If $\lambda_{\min}\left(\nabla g_i(\mathbf{w})\nabla g_i(\mathbf{w})^\top\right) \geq \delta$, it holds that $\lambda_{\min}\left(\nabla g_i(\mathbf{w}')\nabla g_i(\mathbf{w}')^\top\right) \geq \frac{\delta}{2}$ for any $\mathbf{w}'$ satisfying $\|\mathbf{w}' - \mathbf{w}\| \leq \frac{\delta}{4C_g L_g}$.*

*Proof.* Consider any $\mathbf{w}'$ satisfying $\|\mathbf{w}' - \mathbf{w}\| \leq \frac{\delta}{4C_g L_g}$. Note that

$$\lambda_{\min}\left(\nabla g_i(\mathbf{w}')\nabla g_i(\mathbf{w}')^\top\right) = \min_{\mathbf{u} \in \mathbb{R}^{d_1}, \|\mathbf{u}\|=1} \mathbf{u}^\top \nabla g_i(\mathbf{w}')\nabla g_i(\mathbf{w}')^\top \mathbf{u}$$

Let

$$\mathbf{u}_{\mathbf{w}'} \in \underset{\mathbf{u} \in \mathbb{R}^{d_1}, \|\mathbf{u}\|=1}{\arg\min} \mathbf{u}^\top \nabla g_i(\mathbf{w}')\nabla g_i(\mathbf{w}')^\top \mathbf{u}.$$

It holds that

$$\begin{aligned}
&\lambda_{\min}\left(\nabla g_i(\mathbf{w})\nabla g_i(\mathbf{w})^\top\right) - \lambda_{\min}\left(\nabla g_i(\mathbf{w}')\nabla g_i(\mathbf{w}')^\top\right) \\
\leq\, & \mathbf{u}_{\mathbf{w}'}^\top \nabla g_i(\mathbf{w})\nabla g_i(\mathbf{w})^\top \mathbf{u}_{\mathbf{w}'} - \mathbf{u}_{\mathbf{w}'}^\top \nabla g_i(\mathbf{w}')\nabla g_i(\mathbf{w}')^\top \mathbf{u}_{\mathbf{w}'} \\
=\, & \mathbf{u}_{\mathbf{w}'}^\top \left(\nabla g_i(\mathbf{w})\nabla g_i(\mathbf{w})^\top - \nabla g_i(\mathbf{w}')\nabla g_i(\mathbf{w}')^\top\right) \mathbf{u}_{\mathbf{w}'} \\
\leq\, & \left\|\nabla g_i(\mathbf{w})\nabla g_i(\mathbf{w})^\top - \nabla g_i(\mathbf{w}')\nabla g_i(\mathbf{w}')^\top\right\| \\
\leq\, & 2C_g L_g \|\mathbf{w} - \mathbf{w}'\|.
\end{aligned}$$

This implies

$$\lambda_{\min}\left(\nabla g_i(\mathbf{w}')\nabla g_i(\mathbf{w}')^\top\right) \geq \lambda_{\min}\left(\nabla g_i(\mathbf{w}')\nabla g_i(\mathbf{w}')^\top\right) - 2C_g L_g \|\mathbf{w} - \mathbf{w}'\| \geq \delta - \frac{\delta}{2} = \frac{\delta}{2}.$$

$\square$

*Proof of Theorem 4.4.* Suppose $\mathbf{w}$ is an approximate $\epsilon$-stationary solution of (1) with $\epsilon \leq \min\left\{c, \frac{\delta^2}{16nC_g^2 L_g}\right\}$. There exist $\mathbf{t}_1, \ldots, \mathbf{t}_m$ and $\mathbf{y}_i \in \partial f_i(\mathbf{t}_i)$ for $i = 1, \ldots, n$ such that $\|\mathbf{t}_i - g_i(\mathbf{w})\| \leq \epsilon$, $i = 1, \ldots, n$, and $\left\|\frac{1}{n}\sum_{i=1}^n \nabla g_i(\mathbf{w})\mathbf{y}_i\right\| \leq \epsilon$.

Consider the following optimization problem

$$\min_{\mathbf{v}} \left\{ q(\mathbf{v}) := \frac{1}{2n}\sum_{i=1}^n \|\mathbf{t}_i - g_i(\mathbf{v})\|^2 \right\}. \tag{13}$$

We want to show that the optimal objective value of (13) is zero and there exists an optimal solution $\mathbf{w}'$ such that $\|\mathbf{w}' - \mathbf{w}\| \leq \frac{4C_g n\epsilon}{\delta}$.

By the condition on $\epsilon$, it holds that $\frac{4C_g n\epsilon}{\delta} \leq \frac{\delta}{4C_g L_g}$. Hence, by Lemma A.3, if $\|\mathbf{v} - \mathbf{w}\| \leq \frac{4C_g n\epsilon}{\delta}$, it holds that

$$\|\nabla q(\mathbf{v})\|^2 = \frac{1}{n^2}\left\|\sum_{i=1}^n \nabla g_i(\mathbf{v})^\top [g_i(\mathbf{v}) - \mathbf{t}_i]\right\|^2 \geq \frac{\delta}{2n^2}\sum_{i=1}^n \|g_i(\mathbf{v}) - \mathbf{t}_i\|^2 = \frac{\delta}{n}q(\mathbf{v}).$$

Moreover, it always holds that

$$\|\nabla q(\mathbf{v})\|^2 \leq \frac{C_g^2}{n}\sum_{i=1}^n \|g_i(\mathbf{v}) - \mathbf{t}_i\|^2 = 2C_g^2 q(\mathbf{v}).$$

Let $L_q$ be the smoothness parameter of $q(\mathbf{v})$ on the compact set $\{\mathbf{v} \in \mathbb{R}^d | \|\mathbf{v} - \mathbf{w}\| \leq \frac{4C_g n\epsilon}{\delta}\}$. Let $\mathbf{v}_t$ for $t = 0, 1, \ldots$ be generated by the gradient descent method using $\mathbf{v}_0 = \mathbf{w}$ and a step size of $\eta = \frac{1}{L_q}$, i.e., $\mathbf{v}_{t+1} = \mathbf{v}_t - \eta\nabla q(\mathbf{v}_t)$. Suppose $\|\mathbf{v}_s - \mathbf{w}\| \leq \frac{4C_g n\epsilon}{\delta}$ for $s = 0, 1, \ldots, t$ (which is true at least when $t = 0$). By the standard convergence analysis of the gradient descent method for a $L_q$-smooth objective function, we have

$$q(\mathbf{v}_{t+1}) \leq q(\mathbf{v}_t) + \langle \nabla q(\mathbf{v}_t), \mathbf{v}_{t+1} - \mathbf{v}_t \rangle + \frac{L_q}{2}\|\mathbf{v}_{t+1} - \mathbf{v}_t\|^2$$

$$\leq q(\mathbf{v}_t) - \frac{1}{2L_q}\|\nabla q(\mathbf{v}_t)\|^2 \leq q(\mathbf{v}_t) - \frac{\delta}{2L_q n}q(\mathbf{v}_t).$$

Applying this inequality recursively gives

$$\frac{1}{2C_g^2}\|\nabla q(\mathbf{v}_t)\|^2 \leq q(\mathbf{v}_t) \leq \left(1 - \frac{\delta}{2L_q n}\right)^t q(\mathbf{v}_0).$$

Recall that $\|\mathbf{t}_i - g_i(\mathbf{w})\| \leq \epsilon$ such that $q(\mathbf{v}_0) = \frac{1}{2n}\sum_{i=1}^n \|\mathbf{t}_i - g_i(\mathbf{w})\|^2 \leq \frac{\epsilon^2}{2}$. By the triangular inequality, we have

$$\|\mathbf{v}_{t+1} - \mathbf{w}\| \leq \sum_{s=0}^t \eta\|\nabla q(\mathbf{v}_s)\| \leq \sum_{s=0}^t \eta C_g \left(1 - \frac{\delta}{2L_q n}\right)^{\frac{s}{2}}\epsilon \leq \frac{4\eta C_g L_q n}{\delta}\epsilon = \frac{4C_g n\epsilon}{\delta}.$$

By induction, we have $\|\mathbf{v}_t - \mathbf{w}\| \leq \frac{4C_g n\epsilon}{\delta}$ for $t = 0, 1, \ldots$.

Let $\mathbf{w}'$ be any limiting point of $\{\mathbf{v}_t\}_{t\geq 0}$. It holds that $\|\mathbf{w}' - \mathbf{w}\| \leq \frac{4C_g n\epsilon}{\delta}$ and

$$q(\mathbf{w}') \leq \lim_{t\to\infty}\left(1 - \frac{\delta}{2L_q}\right)^t q(\mathbf{v}_0) = 0,$$

meaning that $\mathbf{w}'$ is the optimal solution to (13) with the objective value equal to zero, which implies $\mathbf{t}_i = g_i(\mathbf{w}')$ for $i = 1, \ldots, n$.

Since $\mathbf{w}$ is an approximate $\epsilon$-stationary point of (1) and $\mathbf{y}_i \in \partial f_i(\mathbf{t}_i)$, we have

$$
\left\| \frac{1}{n} \sum_i \partial f_i(g_i(\mathbf{w}')) \nabla g_i(\mathbf{w}') \right\|
$$

$$
= \left\| \frac{1}{n} \sum_i \partial f_i(\mathbf{t}_i) \nabla g_i(\mathbf{w}') \right\|
$$

$$
\leq \left\| \frac{1}{n} \sum_i \partial f_i(\mathbf{t}_i) \nabla g_i(\mathbf{w}) \right\| + \left\| \frac{1}{n} \sum_i \partial f_i(\mathbf{t}_i) \left( \nabla g_i(\mathbf{w}') - \nabla g_i(\mathbf{w}) \right) \right\|
$$

$$
= \left\| \frac{1}{n} \sum_i \mathbf{y}_i \nabla g_i(\mathbf{w}) \right\| + \left\| \frac{1}{n} \sum_i \partial f_i(\mathbf{t}_i) \left( \nabla g_i(\mathbf{w}') - \nabla g_i(\mathbf{w}) \right) \right\|
$$

$$
\leq \epsilon + C_f L_g \| \mathbf{w} - \mathbf{w}' \| \leq (1 + \frac{4 C_g C_f L_g n}{\delta}) \epsilon,
$$

where the last inequality is by Assumptions 3.2. This means $\mathbf{w}'$ is an $O(\epsilon)$-stationary solution of (1) and thus $\mathbf{w}$ is a nearly $O(\epsilon)$-stationary solution of (1). □

## B  Convergence Analysis of SONEX

In this section, we present the proofs for Theorems 4.7. Let $\mathbb{E}_t$ be the conditional expectation conditioning on $\mathcal{B}_1^s$ and $\mathcal{B}_{i,2}^s$ for $i = 1, \ldots, n$ and $s = 0, 1, \ldots, t-1$. To facilitate the proofs, the following quantities are defined based on the notations in Algorithm 1 for $t = 0, \ldots, T-1$,

$$
V_{t+1} := \| \mathbf{v}_{t+1} - \nabla F_\lambda(\mathbf{w}_t) \|^2 \tag{14}
$$

$$
U_{t+1} := \frac{1}{n} \| \mathbf{u}_{t+1} - \mathbf{g}(\mathbf{w}_t) \|^2 \tag{15}
$$

$$
\mathbf{u}_t := [u_{1,t}, \ldots, u_{n,t}]^\top \tag{16}
$$

$$
\mathbf{g}(\mathbf{w}_t) := [g_1(\mathbf{w}_t), \ldots, g_n(\mathbf{w}_t)]^\top. \tag{17}
$$

### B.1  Proof of Theorem 4.7

**Lemma B.1** (Lemma 2 in [46]). *Suppose $\eta \leq \frac{1}{2 L_F}$ in Algorithm 1 with $L_F$ defined in lemma A.1. It holds that*

$$
F_\lambda(\mathbf{w}_{t+1}) \leq F_\lambda(\mathbf{w}_t) + \frac{\eta}{2} V_{t+1} - \frac{\eta}{2} \| \nabla F_\lambda(\mathbf{w}_t) \|^2 - \frac{\eta}{4} \| \mathbf{v}_{t+1} \|^2, \tag{18}
$$

*where $V_{t+1}$ is defined as in* (14).

A recursion for $V_{t+1}$ is provided below.

**Lemma B.2** (Lemma 9 in [2]). *Let $V_{t+1}$ be defined as in* (14). *Suppose $\beta \leq \frac{2}{7}$ in Algorithm 1. It holds that*

$$
\mathbb{E}[V_{t+2}] \leq (1 - \beta) \mathbb{E}[V_{t+1}] + \frac{2 L_F^2 \eta^2 \mathbb{E}\left[ \| \mathbf{v}_{t+1} \|^2 \right]}{\beta} + \frac{3 L_f^2 C_1^2}{n} \mathbb{E}\left[ \| \mathbf{u}_{t+2} - \mathbf{u}_{t+1} \|^2 \right]
$$

$$
+ \frac{2 \beta^2 C_f^2 (C_g^2 + \sigma_1^2)}{\min\{B_1, B_2\}} + 5 \beta L_f^2 C_1^2 \mathbb{E}[U_{t+2}] \tag{19}
$$

*where $C_1^2 := C_g^2 + \frac{\sigma_1^2}{B_2}$, and $U_{t+1}$, $\mathbf{u}_t$ and $\mathbf{g}(\mathbf{w}_t)$ are defined as in* (15), (16) *and* (17), *respectively.*

The following lemma provides the recursion for the estimation error $U_{t+1}$ of the MSVR estimator of $\mathbf{g}(\mathbf{w})$.

**Lemma B.3** (Lemma 1 in [16]). *Suppose $\gamma' = \frac{n - B_1}{B_1(1-\gamma)} + (1 - \gamma)$ and $\gamma \leq \frac{1}{2}$ and the MSVR update is used in Algorithm 1. It holds that*

$$\mathbb{E}\left[U_{t+2}\right] \leq (1 - \frac{\gamma B_1}{n})\mathbb{E}[U_{t+1}] + \frac{8nC_g^2}{B_1}\mathbb{E}\left\|\mathbf{w}_{t+1} - \mathbf{w}_t\right\|^2 + \frac{2B_1\gamma^2\sigma_1^2}{nB_2}. \tag{20}$$

The following lemma handles the term $\|\mathbf{u}_{t+1} - \mathbf{u}_t\|^2$.

**Lemma B.4** (Lemma 6 in [16]). *Under the same assumptions as Lemma B.3, it holds that*

$$\mathbb{E}[\|\mathbf{u}_{t+2} - \mathbf{u}_{t+1}\|^2] \leq \frac{4B_1\gamma^2}{n}\mathbb{E}[U_{t+1}] + \frac{9n^2C_g^2}{B_1}\mathbb{E}\|\mathbf{w}_{t+1} - \mathbf{w}_t\|^2 + \frac{2B_1\gamma^2\sigma_0^2}{B_2}. \tag{21}$$

Now we are ready to proof Theorem 4.7

*Proof of Theorem 4.7.* Note that $\|\mathbf{w}_{t+1} - \mathbf{w}_t\|^2 = \eta^2\|\mathbf{v}_{t+1}\|^2$. Let $P$ be a non-negative constant to be determined later. Taking the weighted summation of both sides (18), (19), (20), and (21) as specified in the following formula

$$\frac{1}{\eta} \times (18) + \frac{1}{\beta} \times (19) + P \times (20) + \frac{3L_f^2C_1^2}{n\beta} \times (21),$$

we have

$$\frac{1}{\eta}\mathbb{E}F_\lambda(\mathbf{w}_{t+1}) + \frac{1}{\beta}\mathbb{E}V_{t+2} + \left(P - 5L_f^2C_1^2\right)\mathbb{E}U_{t+2}$$

$$\leq \frac{1}{\eta}\mathbb{E}F_\lambda(\mathbf{w}_t) + \left(\frac{1}{\beta} - \frac{1}{2}\right)\mathbb{E}V_{t+1} + \left(P\left(1 - \frac{\gamma B_1}{n}\right) + \frac{12L_f^2C_1^2\gamma^2B_1}{\beta n^2}\right)\mathbb{E}U_{t+1}$$

$$- \left(\frac{1}{4} - \frac{2L_F^2\eta^2}{\beta^2} - \frac{8nC_g^2P\eta^2}{B_1} - \frac{27nL_f^2C_1^2C_g^2\eta^2}{B_1\beta}\right)\mathbb{E}\left\|\mathbf{v}_{t+1}\right\|^2$$

$$+ \frac{2\beta C_f^2(C_g^2 + \sigma_1^2)}{\min\{B_1, B_2\}} + \frac{2B_1\gamma^2\sigma_1^2P}{nB_2} + \frac{6L_f^2C_1^2\gamma^2\sigma_0^2B_1}{\beta nB_2} - \frac{1}{2}\mathbb{E}\left\|\nabla F_\lambda(\mathbf{w}_t)\right\|^2$$

We choose $\eta$ such that

$$\frac{1}{4} - \frac{2L_F^2\eta^2}{\beta^2} - \frac{8nC_g^2P\eta^2}{B_1} - \frac{27nL_f^2C_1^2C_g^2\eta^2}{B_1\beta} \geq 0. \tag{22}$$

Moreover, we set $P$ to be the solution of the following linear equation

$$P\left(1 - \frac{\gamma B_1}{n}\right) + \frac{12L_f^2C_1^2\gamma^2B_1}{\beta n^2} = P - 5L_f^2C_1^2,$$

which means

$$P = \frac{n}{\gamma B_1}\left(5L_f^2C_1^2 + \frac{12L_f^2C_1^2\gamma^2B_1}{\beta n^2}\right).$$

Combining the results above gives

$$\frac{1}{\eta}\mathbb{E}F_\lambda(\mathbf{w}_{t+1}) + \frac{1}{\beta}\mathbb{E}V_{t+2} + \left(P - 5L_f^2C_1^2\right)\mathbb{E}U_{t+2}$$

$$\leq \frac{1}{\eta}\mathbb{E}F_\lambda(\mathbf{w}_t) + \frac{1}{\beta}\mathbb{E}V_{t+1} + \left(P - 5L_f^2C_1^2\right)\mathbb{E}U_{t+1}$$

$$+ \frac{2\beta C_f^2(C_g^2 + \sigma_1^2)}{\min\{B_1, B_2\}} + \frac{2B_1\gamma^2\sigma_1^2P}{nB_2} + \frac{6L_f^2C_1^2\gamma^2\sigma_0^2B_1}{\beta nB_2} - \frac{1}{2}\mathbb{E}\left\|\nabla F_\lambda(\mathbf{w}_t)\right\|^2.$$

Summing both sides of the inequality above over $t = 0, 1, \ldots, T - 1$ and organizing term lead to

$$\frac{1}{T}\sum_{t=0}^{T-1}\|\nabla F_\lambda(\mathbf{w}_t)\|^2 \leq \frac{2}{T}\left(\frac{1}{\eta}(F_\lambda(\mathbf{w}_0) - F_\lambda(\mathbf{w}_T)) + \frac{1}{\beta}\mathbb{E}V_1 + \left(P - 5L_f^2C_1^2\right)\mathbb{E}U_1\right)$$

$$+ \frac{4\beta C_f^2(C_g^2 + \sigma_1^2)}{\min\{B_1, B_2\}} + \frac{4B_1\gamma^2\sigma_1^2P}{nB_2} + \frac{12L_f^2C_1^2\gamma^2\sigma_0^2B_1}{\beta nB_2} \tag{23}$$

Note that $L_F = O(\frac{1}{\epsilon})$, $L_f = O(\frac{1}{\epsilon})$ and $P = O(\frac{n}{B_1\gamma\epsilon^2} + \frac{\gamma}{n\beta\epsilon^2})$. We require that all the terms in the right-hand side to be in the order of $O(\epsilon^2)$. To ensure $\frac{4\beta C_f^2(C_g^2+\sigma_1^2)}{\min\{B_1,B_2\}} = O(\epsilon^2)$, we must have $\beta = O(\min\{B_1,B_2\}\epsilon^2)$. To ensure $\frac{4B_1\gamma^2\sigma_1^2 P}{nB_2} = O(\frac{\gamma}{B_2\epsilon^2} + \frac{B_1\gamma^3}{n^2B_2\beta\epsilon^2}) = O(\epsilon^2)$, we must have $\gamma = O(B_2\epsilon^4)$. With these orders of $\beta$ and $\gamma$, we also have $P = O(\frac{n}{B_1B_2\epsilon^6})$ and $\frac{12L_f^2C_1^2\gamma^2\sigma_0^2B_1}{\beta nB_2} = O(\epsilon^4)$. Therefore the last three terms in (23) are $O(\epsilon^2)$.

To satisfy (22), we will need to set

$$\eta = O\left(\min\left\{\frac{\beta}{L_F}, \sqrt{\frac{B_1}{nP}}, \frac{1}{L_f}\sqrt{\frac{B_1\beta}{n}}\right\}\right)$$
$$= O\left(\min\left\{\min\{B_1,B_2\}\epsilon^3, \frac{B_1\sqrt{B_2}}{n}\epsilon^3, \sqrt{\frac{B_1\min\{B_1,B_2\}}{n}}\epsilon^2\right\}\right)$$
$$= O(\frac{B_1\sqrt{B_2}}{n}\epsilon^3).$$

We also set the sizes of $\mathcal{B}_{i,2}$ to be $O(\frac{1}{\epsilon^3})$ so that $\mathbb{E}U_1 = O(\epsilon^3)$. Also, it is easy to see that $F_\lambda(\mathbf{w}_0) - F_\lambda(\mathbf{w}_T) = O(1)$ and $\mathbb{E}V_1 = O(1)$. Then the first term in (23) becomes

$$\frac{1}{T}O\left(\frac{1}{\eta} + \frac{1}{\beta} + P\epsilon^3\right) = \frac{1}{T}O\left(\frac{n}{B_1\sqrt{B_2}\epsilon^3} + \frac{1}{\min\{B_1,B_2\}\epsilon^2} + \frac{n}{B_1B_2\epsilon^3}\right)$$

which will become $O(\epsilon^2)$ when $T = O(\frac{n}{B_1\sqrt{B_2}\epsilon^5})$.

$\square$

## C   Convergence Analysis of ALEXR2

To establish the complexity of Algorithm 2, we first present the convergence result of ALEXR [5] which is the inner loop of ALEXR2, and it shows the complexity of Algorithm 2 to produce $\hat{\mathbf{z}}_t$ such that $\left\|\hat{\mathbf{z}}_t - \text{prox}_{\nu F_\lambda}(\mathbf{w}_t)\right\|_2^2 \leq O(\epsilon')$.

**Theorem C.1** (A Variant of Theorem 1 in [5] when $g_i$ is non-smooth $\rho_g$-weakly convex and $\mu = \frac{1}{\nu} - \sqrt{d_1}C_f\rho_g$). *Suppose Assumptions 3.2 (A3) and 4.5 hold. For any $\epsilon' > 0$, there exists $\theta \in (0,1)$ with $1 - \theta = O(\epsilon\epsilon')$ such that, by setting $\eta = \frac{1-\theta}{(4\theta-1)\sqrt{d_1}C_f\rho_g}$ and $\gamma = (\frac{B_1}{n(1-\theta)} - 1)^{-1}$, the inner loop of Algorithm 2 (i.e., ALEXR) guarantees $\frac{\mu}{2}\mathbb{E}\left\|\hat{\mathbf{z}}_t - prox_{\nu F_\lambda}(\mathbf{w}_t)\right\|_2^2 \leq \epsilon'$ for any $t$ after*

$$K_t = \tilde{O}\left(\frac{n}{B_1} + C_g\sqrt{\frac{nL_f}{B_1\sqrt{d_1}\rho_gC_f}} + \frac{nL_f\sigma_0^2}{B_2B_1\epsilon'} + \frac{C_f\sigma_1^2}{B_2\sqrt{d_1}\rho_g\epsilon'} + \frac{C_fC_g^2}{B_1\sqrt{d_1}\rho_g\epsilon'} + \frac{C_fC_g^2}{\sqrt{d_1}\rho_g\epsilon'}\right) = \tilde{O}(\frac{n\sigma_0^2}{B_2B_1\epsilon\epsilon'})$$

*iterations.*

In this section, we provide the convergence analysis for ALEXR2 in Algorithm 2.

**Lemma C.2.** $F_{\lambda,\nu}(\mathbf{w})$ *is $L_{F_{\lambda,\nu}}$-smooth with $L_{F_{\lambda,\nu}} = \max\{\frac{1}{\nu}, \frac{\rho_{F_\lambda}}{1-\nu\rho_{F_\lambda}}\}$ and $\rho_{F_\lambda} = \sqrt{d_1}C_f\rho_g$.*

**Lemma C.3.** *The inner min-max problem in 6 is a $(\frac{1}{\nu} - \rho_{F_\lambda})$-strongly-convex and $\lambda$-strongly-concave problem when $\nu \in (0, \rho_{F_\lambda}^{-1})$.*

*Proof.* As discussed in Section 4.2, the term $\mathbf{y}_i^\top g_i(\mathbf{w})$ is $\rho_{F_\lambda}$-weakly convex with $\rho_{F_\lambda} = \sqrt{d_1}C_f\rho_g$. Therefore, when $\nu \in (0, \rho_{F_\lambda}^{-1})$, the objective in (6) becomes strongly convex in $\mathbf{z}$ for any fixed $\mathbf{y} \in \mathbb{R}^{nd_1}$.

To establish strong concavity in $\mathbf{y}$, we note that the term $\bar{f}_i^*$ is the Fenchel conjugate of a $\frac{1}{\lambda}$-smooth function $\bar{f}_i$. It is a standard result in convex analysis (see, e.g., Theorem 18.15 in [47]) that the conjugate of an $L$-smooth function is $\frac{1}{L}$-strongly concave. Thus, $\bar{f}_i^*$ is $\lambda$-strongly convex.

The remaining terms in the objective are either independent of $\mathbf{y}$ or linear in $\mathbf{y}$, and hence do not affect strong concavity. Therefore, the objective is strongly convex in $\mathbf{z}$ and strongly concave in $\mathbf{y}$ when $\nu \in (0, \rho_{F_\lambda}^{-1})$. $\square$

## C.1 Proof of Technical Lemma

Note that our proof for theorem C.1 is almost the same as that of theorem 1 in [5], we only show the difference in this and the following subsections.

**Lemma C.4** (Generalization of Lemma 7 in ALEXR [5] to weakly-convex $g_i$). *Suppose that Assumptions 3.2(A2), 4.5 hold. Then, the following holds for Algorithm 2.*

$$\frac{1}{n}\mathbb{E}\sum_{i=1}^{n}\left\langle g_i(\mathbf{w}_{k+1}) - g_i(\mathbf{w}_*),\, \bar{y}_{k+1}^{(i)}\right\rangle - \mathbb{E}\left\langle G_k,\, \mathbf{w}_{k+1} - \mathbf{w}_*\right\rangle$$

$$\leq \frac{\frac{C_f^2\sigma_1^2}{B_2} + \frac{C_f^2 C_g^2}{B_1}}{\frac{1}{\eta} + \frac{1}{\nu}} + \frac{L_g C_f}{2}\left\|\mathbf{w}_{k+1} - \mathbf{w}_k\right\|_2^2 + \frac{\sqrt{d_1}C_f L_g}{2}\left\|\mathbf{w}_k - \mathbf{w}^*\right\|_2^2 \qquad (24)$$

*Proof.* For this lemma we only highlight the difference to the proof of lemma 7 in ALEXR [5]. We define $\Delta_k := \frac{1}{B_1}\sum_{i\in\mathcal{B}_1^k}[\nabla g_i(\mathbf{w}_k; \tilde{\mathcal{B}}_{i,2}^k)]^\top y_{k+1}^{(i)} - \frac{1}{n}\sum_{i=1}^n [\nabla g_i(\mathbf{w}_k)]^\top \bar{y}_{k+1}^{(i)}$.

$$\frac{1}{n}\sum_{i=1}^{n}\left\langle g_i(\mathbf{w}_{k+1}) - g_i(\mathbf{w}_*),\, \bar{y}_{k+1}^{(i)}\right\rangle - \left\langle G_k,\, \mathbf{w}_{k+1} - \mathbf{w}_*\right\rangle$$

$$= \frac{1}{n}\sum_{i=1}^{n}\left\langle g_i(\mathbf{w}_{k+1}) - g_i(\mathbf{w}_k),\, \bar{y}_{k+1}^{(i)}\right\rangle + \frac{1}{n}\sum_{i=1}^{n}\left\langle g_i(\mathbf{w}_k) - g_i(\mathbf{w}_*),\, \bar{y}_{k+1}^{(i)}\right\rangle$$

$$+ \left\langle \frac{1}{n}\sum_{i=1}^{n}[\nabla g_i(\mathbf{w}_k)]^\top \bar{y}_{k+1}^{(i)} + \Delta_k,\, \mathbf{w}_* - \mathbf{w}_{k+1}\right\rangle$$

$$\overset{g_i \text{ smooth}^6}{\leq} \frac{1}{n}\sum_{i=1}^{n}\left\langle g_i(\mathbf{w}_{k+1}) - g_i(\mathbf{w}_k),\, \bar{y}_{k+1}^{(i)}\right\rangle + \left\langle \frac{1}{n}\sum_{i=1}^{n}[\nabla g_i(\mathbf{w}_k)]^\top \bar{y}_{k+1}^{(i)},\, \mathbf{w}_k - \mathbf{w}_*\right\rangle$$

$$+ \frac{L_g}{2}\left\|\mathbf{w}_k - \mathbf{w}^*\right\|^2 * \frac{1}{n}\sum_{i=1}^{n}\left\langle \bar{y}_{k+1}^{(i)},\, \mathbf{1}\right\rangle + \left\langle \frac{1}{n}\sum_{i=1}^{n}[\nabla g_i(\mathbf{w}_k)]^\top \bar{y}_{k+1}^{(i)} + \Delta_k,\, \mathbf{w}_* - \mathbf{w}_{k+1}\right\rangle$$

$$= \frac{1}{n}\sum_{i=1}^{n}\left\langle \bar{y}_{k+1}^{(i)},\, g_i(\mathbf{w}_{k+1}) - g_i(\mathbf{w}_k)\right\rangle + \left\langle \frac{1}{n}\sum_{i=1}^{n}[\nabla g_i(\mathbf{w}_k)]^\top \bar{y}_{k+1}^{(i)},\, \mathbf{w}_k - \mathbf{w}_{k+1}\right\rangle$$

$$+ \left\langle \Delta_k,\, \mathbf{w}_* - \mathbf{w}_{k+1}\right\rangle + \frac{\sqrt{d_1}C_f L_g}{2}\left\|\mathbf{w}_k - \mathbf{w}^*\right\|^2 \qquad (25)$$

Note that we have an extra term in blue in 25 comparing with lemma 7 in [5] due to the weakly convexity of $g_i$. And the remaining part is just the same so we omit it. □

**Lemma C.5** (Generalization of Lemma 8 in ALEXR to weakly-convex $g_i$). *Suppose that assumptions 3.2(A3), 4.5 hold. Then, the following holds for Algorithm 2.*

$$\frac{1}{n}\mathbb{E}\sum_{i=1}^{n}\left\langle g_i(\mathbf{w}_{k+1}) - g_i(\mathbf{w}_*),\, \bar{y}_{k+1}^{(i)}\right\rangle - \mathbb{E}\left\langle G_k,\, \mathbf{w}_{k+1} - \mathbf{w}_*\right\rangle$$

$$\leq \frac{\frac{C_f^2\sigma_1^2}{B_2} + \frac{C_f^2 C_g^2}{B_1} + 4C_f^2 C_g^2}{\frac{1}{\eta} + \frac{1}{\nu}} + \frac{\frac{1}{\eta} + \frac{1}{\nu}}{4}\left\|\mathbf{w}_{k+1} - \mathbf{w}_k\right\|_2^2 + \frac{\sqrt{d_1}C_f\rho_g}{2}\left\|\mathbf{w}_k - \mathbf{w}^*\right\|^2 \qquad (26)$$

*Proof.* Similar to lemma C.4, we have an extra term $\frac{\sqrt{d_1}C_f\rho_g}{2}\left\|\mathbf{w}_k - \mathbf{w}^*\right\|^2$ in the upper bound for weakly convex $g_i$. □

---

[6] this inequality also holds with assumption 3.2(A3) so lemma C.5 can use the same technique to handle weakly convexity of $g_i$

## C.2 Proof of Inner Loop

*Proof of theorem C.1.* Similar to theorem 1 in [5] but use lemma C.5 instead of lemma 8 in ALEXR, we have

$$
\begin{aligned}
&\mathbb{E}[L(\mathbf{w}_{k+1}, y_*) - L(\mathbf{w}_*, \bar{y}_{k+1})] \\
&\leq \frac{\frac{1}{\gamma} + \left(1 - \frac{B_1}{n}\right)}{B_1} \mathbb{E}[U_{\bar{f}_i^*}(y_*, y_k)] - \frac{\frac{1}{\gamma} + 1}{B_1} \mathbb{E}[U_{\bar{f}_i^*}(y_*, y_{k+1})] \\
&\quad + \left(\frac{1}{2\eta} + \frac{\sqrt{d_1} C_f \rho_g}{2}\right) \mathbb{E} \|\mathbf{w}_* - \mathbf{w}_k\|_2^2 - \left(\frac{1}{2\eta} + \frac{1}{2\nu}\right) \mathbb{E} \|\mathbf{w}_* - \mathbf{w}_{k+1}\|_2^2 \\
&\quad - \left(\frac{1}{\gamma n} - \frac{\lambda_2 + \lambda_3 \theta}{\lambda n}\right) \mathbb{E}\left[U_{\bar{f}_i^*}(\bar{y}_{k+1}, y_k)\right] - \left(\frac{1}{2\eta} - \frac{C_g^2}{2\lambda_2} - \frac{\frac{1}{\eta} + \frac{1}{\nu}}{4}\right) \mathbb{E} \|\mathbf{w}_{k+1} - \mathbf{w}_k\|_2^2 + \frac{\theta C_g^2}{2\lambda_3} \mathbb{E} \|\mathbf{w}_k - \mathbf{w}_{k-1}\|_2^2 \\
&\quad + \mathbb{E}[\Gamma_{k+1} - \theta \Gamma_k] + \frac{2(1+2\theta)\sigma_0^2}{B\lambda(1 + \frac{1}{\gamma})} + \frac{\frac{C_f^2 \sigma_1^2}{B_2} + \frac{C_f^2 C_g^2}{B_1} + 4 C_f^2 C_g^2}{\frac{1}{\eta} + \frac{1}{\nu}}. \quad\quad (27)
\end{aligned}
$$

Define $\Upsilon_k^{\mathbf{w}} := \frac{1}{2}\mathbb{E} \|\mathbf{w}_* - \mathbf{w}_k\|_2^2$ and $\Upsilon_k^y = \frac{1}{S}\mathbb{E}U_{\bar{f}_i^*}(y_*, y_k)$. Note that $L(\mathbf{w}_{k+1}, y_*) - L(\mathbf{w}_*, \bar{y}_{k+1}) \geq 0$. Multiply both sides of (27) by $\theta^{-k}$ and do telescoping sum from $k = 0$ to $K_t - 1$. Add $\eta \theta^{-K_t} \Upsilon_{K_t}^{\mathbf{w}}$ to both sides.

$$
\begin{aligned}
\frac{\theta^{-K_t} \Upsilon_{K_t}^{\mathbf{w}}}{\eta} &\leq \sum_{k=0}^{K_t-1} \theta^{-k} \left(\left((\frac{1}{\eta} + \sqrt{d_1} C_f \rho_g)\Upsilon_k^{\mathbf{w}} + \left(\frac{1}{\gamma} + \left(1 - \frac{B_1}{n}\right)\right) \Upsilon_k^y - \theta \mathbb{E}\Gamma_k\right)\right. \\
&\quad \left. - \left((\frac{1}{\eta} + \frac{1}{\nu})\Upsilon_{k+1}^{\mathbf{w}} + (\frac{1}{\gamma} + 1)\Upsilon_{k+1}^y - \mathbb{E}\Gamma_{k+1}\right)\right) \\
&\quad + \frac{\theta^{-K_t} \Upsilon_{K_t}^{\mathbf{w}}}{\eta} + \left(\frac{2(1+2\theta)\sigma_0^2}{\lambda B(1 + \frac{1}{\gamma})} + \frac{\frac{C_f^2 \sigma_1^2}{B_2} + \frac{C_f^2 C_g^2}{B_1} + 4 C_f^2 C_g^2}{\frac{1}{\eta} + \frac{1}{\nu}}\right) \sum_{k=0}^{K_t-1} \theta^{-k} \\
&\quad - \sum_{k=0}^{K_t-1} \theta^{-k} \left(\frac{1}{\gamma n} - \frac{(\lambda_2 + \lambda_3 \theta)}{\lambda n}\right) \mathbb{E}[U_{\bar{f}_i^*}(\bar{y}_{k+1}, y_k)] \\
&\quad - \sum_{k=0}^{K_t-1} \theta^{-k} \left(\frac{1}{2\eta} - \frac{\frac{1}{\eta} + \frac{1}{\nu}}{4} - \frac{C_g^2}{2\lambda_2} - \frac{C_g^2}{2\lambda_3}\right) \mathbb{E} \|\mathbf{w}_{k+1} - \mathbf{w}_k\|_2^2 - \theta^{-K_t+1} \frac{C_g^2}{2\lambda_3} \mathbb{E} \|\mathbf{w}_{K_t} - \mathbf{w}_{K_t-1}\|_2^2
\end{aligned}
$$

$$(28)$$

Let $\eta \geq \frac{1-\theta}{\theta/\nu - \sqrt{d_1} C_f \rho_g}, \theta \geq \frac{1}{2}$ and $1/\nu > 4\sqrt{d_1} C_f \rho_g$ such that $\theta \geq \frac{\frac{1}{\eta} + \sqrt{d_1} C_f \rho_g}{\frac{1}{\eta} + \frac{1}{\nu}}$ and $\frac{1}{\gamma} + 1 \leq \frac{B_1}{n(1-\theta)}$ such that $\theta \geq \frac{\frac{1}{\gamma} + 1 - \frac{B_1}{n}}{\frac{1}{\gamma} + 1}$. Then,

$$
\begin{aligned}
&\sum_{k=0}^{K_t-1} \theta^{-k} \left(\left((\frac{1}{\eta} + \sqrt{d_1} C_f \rho_g)\Upsilon_k^{\mathbf{w}} + \left(\frac{1}{\gamma} + \left(1 - \frac{B_1}{n}\right)\right) \Upsilon_k^y - \theta \mathbb{E}\Gamma_k\right) - ((\frac{1}{\eta} + \frac{1}{\nu})\Upsilon_{k+1}^{\mathbf{w}} + (\frac{1}{\gamma} + 1)\Upsilon_{k+1}^y - \mathbb{E}\Gamma_{k+1})\right) \\
&\leq (\frac{1}{\eta} + \sqrt{d_1} C_f \rho_g)\Upsilon_0^{\mathbf{w}} + \left(\frac{1}{\gamma} + \left(1 - \frac{B_1}{n}\right)\right) \Upsilon_0^y - \theta \mathbb{E}\Gamma_0 - \theta^{-K_t+1} \left((\frac{1}{\eta} + \frac{1}{\nu})\Upsilon_{K_t}^{\mathbf{w}} + (\frac{1}{\gamma} + 1)\Upsilon_{K_t}^y - \mathbb{E}\Gamma_{K_t}\right).
\end{aligned}
$$

By setting $\mathbf{w}_{-1} = \mathbf{w}_0$, we have $\Gamma_0 = 0$. Besides, we have $\Gamma_{K_t} \leq \frac{1}{n}\sum_{i=1}^n \|g_i(\mathbf{w}_{K_t}) - g_i(\mathbf{w}_{K_t-1})\| \|y_*^{(i)} - y_k^{(i)}\| \leq \frac{C_g}{n} \|\mathbf{w}_{K_t} - \mathbf{w}_{K_t-1}\| \|y_* - y_{K_t}\| \leq \frac{C_g^2}{2\lambda_3} \|\mathbf{w}_{K_t} - \mathbf{w}_{K_t-1}\|^2 + \frac{\lambda_3}{2\lambda n^2} U_{\bar{f}_i^*}(y_*, y_{K_t})$. Note that the first term in the RHS here can-

celled out with the last term of RHS in 28. Thus,

$$\frac{\theta^{-K_t}\Upsilon^{\mathbf{w}}_{K_t}}{\eta} \le (\frac{1}{\eta} + \sqrt{d_1}C_f\rho_g)\Upsilon^{\mathbf{w}}_0 + \left(\frac{1}{\gamma} + \left(1 - \frac{B_1}{n}\right)\right)\Upsilon^y_0$$

$$- \theta^{-K_t+1}\left((\frac{1}{\eta} + \frac{1}{\nu})\Upsilon^{\mathbf{w}}_{K_t} + \underbrace{(\frac{1}{\gamma} + 1 - \frac{\lambda_3 S}{2\lambda n^2})}_{\heartsuit}\Upsilon^y_{K_t} - \frac{1}{\eta\theta}\Upsilon^{\mathbf{w}}_{K_t}\right)$$

$$+ \left(\frac{2(1+2\theta)\sigma_0^2}{\lambda B(1+\frac{1}{\gamma})} + \frac{\frac{C_f^2\sigma_1^2}{B_2} + \frac{C_f^2 C_g^2}{B_1} + 4C_f^2 C_g^2}{\frac{1}{\eta} + \frac{1}{\nu}}\right)\sum_{k=0}^{K_t-1}\theta^{-k} - \sum_{k=0}^{K_t-1}\theta^{-k}\underbrace{\left(\frac{1}{\gamma n} - \frac{(\lambda_2 + \lambda_3\theta)}{\lambda n}\right)}_{\heartsuit}\mathbb{E}[U_{\bar{f}_i^*}(\bar{y}_{k+1}, y_k)]$$

$$- \sum_{k=0}^{K_t-1}\theta^{-k}\underbrace{\left(\frac{1}{2\eta} - \frac{\frac{1}{\eta} + \frac{1}{\nu}}{4} - \frac{C_g^2}{2\lambda_2} - \frac{C_g^2}{2\lambda_3}\right)}_{\heartsuit}\mathbb{E}\left\|\mathbf{w}_{k+1} - \mathbf{w}_k\right\|_2^2. \tag{29}$$

To make the $\heartsuit$ terms in (29) be non-negative, we choose $\lambda_2 \asymp \lambda_3 \asymp \frac{C_g\sqrt{B_1\lambda}}{\sqrt{n}\sqrt{d_1}C_f\rho_g}$ while ensuring that

$$\gamma \le O\left(\frac{\lambda n^2}{\lambda_3 B_1} \wedge \frac{\lambda}{\lambda_2 + \lambda_3\theta}\right) = O\left(\frac{\sqrt{n\sqrt{d_1}C_f\rho_g\lambda}}{C_g\sqrt{B_1}}\right), \quad \eta \le O\left(\frac{\sqrt{B_1\lambda}}{C_g\sqrt{n\sqrt{d_1}C_f\rho_g}} \wedge \frac{1}{\rho_g C_f}\right). \tag{30}$$

By selecting $\eta = \frac{1-\theta}{\theta/\nu - \sqrt{d_1}C_f\rho_g}$, $\frac{1}{\gamma} = \frac{B_1}{n(1-\theta)} - 1$ and $\frac{1}{\nu} = 4\sqrt{d_1}C_f\rho_g$ we have that $\frac{1}{\gamma} + \left(1 - \frac{B_1}{n}\right) = \frac{B_1\theta}{n(1-\theta)}$.

$$\mu\Upsilon^{\mathbf{w}}_{K_t} \le \mu(1 + \eta\sqrt{d_1}C_f\rho_g)\theta^{K_t}\Upsilon^{\mathbf{w}}_0 + \frac{\left(\frac{1}{\gamma} + \left(1 - \frac{B_1}{n}\right)\right)\mu(1-\theta)}{\theta/\nu - \sqrt{d_1}C_f\rho_g}\theta^{K_t}\Upsilon^y_0$$

$$+ \left(\frac{2(1+2\theta)\sigma_0^2}{\lambda B(1+\frac{1}{\gamma})} + \frac{\frac{C_f^2\sigma_1^2}{B_2} + \frac{C_f^2 C_g^2}{B_1} + 4C_f^2 C_g^2}{\frac{1}{\eta} + \frac{1}{\nu}}\right)\frac{(\theta^{-K_t} - 1)\theta^{K_t}}{\theta^{-1} - 1}\frac{(1-\theta)\mu}{\theta/\nu - \sqrt{d_1}C_f\rho_g}$$

$$\le \mu(1 + \eta\sqrt{d_1}C_f\rho_g)\theta^{K_t}\Upsilon^{\mathbf{w}}_0 + \frac{\frac{B_1}{n}\theta\mu}{\frac{1}{2}\theta/\nu}\theta^{K_t}\Upsilon^y_0 + 2\mu\nu\left(\frac{2(1+2\theta)\sigma_0^2}{\lambda B_2(1+\frac{1}{\gamma})} + \frac{\frac{C_f^2\sigma_1^2}{B_2} + \frac{C_f^2 C_g^2}{B_1} + 4C_f^2 C_g^2}{\frac{1}{\eta} + \frac{1}{\nu}}\right)$$

$$\le \mu(1 + \eta\sqrt{d_1}C_f\rho_g)\theta^{K_t}\Upsilon^{\mathbf{w}}_0 + 2\theta^{K_t}\Upsilon^y_0 + 2\left(\frac{2(1+2\theta)\sigma_0^2}{\lambda B_2(\frac{1}{\gamma} + 1)} + \eta\left(\frac{C_f^2\sigma_1^2}{B_2} + \frac{C_f^2 C_g^2}{B_1} + 4C_f^2 C_g^2\right)\right).$$

We select

$$1 - \theta = O\left(\frac{1}{2} \wedge \frac{B_1}{n} \wedge \frac{1}{C_g}\sqrt{\frac{B_1\lambda\sqrt{d_1}C_f\rho_g}{n}} \wedge \frac{\lambda B_2 B_1\epsilon'}{\sigma_0^2 n} \wedge \frac{B_2\sqrt{d_1}\rho_g\epsilon'}{C_f\sigma_1^2} \wedge \frac{B_1\sqrt{d_1}\rho_g\epsilon'}{C_f C_g^2} \wedge \frac{\sqrt{d_1}\rho_g\epsilon'}{C_f C_g^2}\right)$$

to make (30) hold and

$$\frac{2(1+2\theta)\sigma_0^2}{\lambda B_2(\frac{1}{\gamma} + 1)} + \eta\left(\frac{C_f^2\sigma_1^2}{B_2} + \frac{C_f^2 C_g^2}{B_1} + 4C_f^2 C_g^2\right) \le \frac{2(1+2\theta)(1-\theta)\sigma_0^2 n}{\lambda B_2 B_1} + \frac{(1-\theta)\left(\frac{C_f^2\sigma_1^2}{B_2} + \frac{C_f^2 C_g^2}{B_1} + 4C_f^2 C_g^2\right)}{\sqrt{d_1}C_f\rho_g} \le \epsilon'.$$

Besides, we show that $\Upsilon^{\mathbf{w}}_0$ can be bounded by constant.(boundedness of $\Upsilon^y_0$ is already guaranteed by assumption 4.9). Note that $\mathbf{w}^*_t = \text{prox}_{\nu F_\lambda}(\mathbf{w}_{t,0})$ we have:

$$\because 0 \in \partial F_\lambda(\mathbf{w}^*) + \frac{1}{\nu}(\mathbf{w}^* - \mathbf{w}_0)$$

$$\therefore \mathbf{w}_0 - \mathbf{w}^* \in \nu\partial F_\lambda(\mathbf{w}^*)$$

$$\therefore \Upsilon^{\mathbf{w}}_0 = \frac{1}{2}\left\|\mathbf{w}_0 - \mathbf{w}^*\right\|^2 \le \frac{1}{2}(\nu C_{F_\lambda})^2 = \frac{1}{2}(\nu C_f C_g)^2 = O(C_f^4 C_g^2 \rho_g^2)$$

Since $L_f := \frac{1}{\lambda} = O(\frac{1}{\epsilon})$, the number of inner loop iterations needed by Algorithm 2 to make $\mu \Upsilon^{\mathbf{w}}_{K_t} \leq \epsilon'$ is

$$K_t = \tilde{O}\left( \frac{n}{B_1} + C_g \sqrt{\frac{nL_f}{B_1\sqrt{d_1}\rho_g C_f}} + \frac{nL_f\sigma_0^2}{B_2 B_1\epsilon'} + \frac{C_f\sigma_1^2}{B_2\sqrt{d_1}\rho_g\epsilon'} + \frac{C_f C_g^2}{B_1\sqrt{d_1}\rho_g\epsilon'} + \frac{C_f C_g^2}{\sqrt{d_1}\rho_g\epsilon'} \right)$$

where $\tilde{O}(\cdot)$ hides the $\text{polylog}(C_f C_g \rho_g/\epsilon')$ factor and the green term is the dominant term so $K_t = \tilde{O}(\frac{n\sigma_0^2}{B_2 B_1\epsilon\epsilon'})$. $\qquad\square$

## C.3 Proof of Outer loop

*Proof of Theorem 4.10.* Let $\mathbf{w}_{-1} = \mathbf{w}_0$. By lemma C.2, $F_{\lambda,\nu}(\mathbf{w}_t)$ is $L_{F_{\lambda,\nu}}$-smooth. Since $\beta \leq \frac{1}{2}$, we have

$$\|\mathbf{v}_{t+1} - \nabla F_{\lambda,\nu}(\mathbf{w}_t)\|^2$$

$$= \left\| (1-\beta)(\mathbf{v}_t - \nabla F_{\lambda,\nu}(\mathbf{w}_{t-1})) + (1-\beta)(\nabla F_{\lambda,\nu}(\mathbf{w}_{t-1}) - \nabla F_{\lambda,\nu}(\mathbf{w}_t)) + \beta\left(\frac{1}{\nu}(\mathbf{w}_t - \hat{\mathbf{z}}_t) - \nabla F_{\lambda,\nu}(\mathbf{w}_t)\right) \right\|^2$$

$$\leq (1+\frac{\beta}{2})(1-\beta)^2\left( (1+\frac{\beta}{2})\|\mathbf{v}_t - \nabla F_{\lambda,\nu}(\mathbf{w}_{t-1})\|^2 + (1+\frac{2}{\beta})\|\nabla F_{\lambda,\nu}(\mathbf{w}_{t-1}) - \nabla F_{\lambda,\nu}(\mathbf{w}_t)\|^2 \right)$$

$$+ (1+\frac{2}{\beta})\beta^2\left\| \frac{1}{\nu}(\mathbf{w}_t - \hat{\mathbf{z}}_t) - \nabla F_{\lambda,\nu}(\mathbf{w}_t) \right\|^2$$

$$\leq (1-\frac{\beta}{2})\|\mathbf{v}_t - \nabla F_{\lambda,\nu}(\mathbf{w}_{t-1})\|^2 + \frac{3L_{F_{\lambda,\nu}}^2}{\beta}\|\mathbf{w}_{t-1} - \mathbf{w}_t\|^2 + 3\beta\left\| \frac{1}{\nu}(\mathbf{w}_t - \hat{\mathbf{z}}_t) - \nabla F_{\lambda,\nu}(\mathbf{w}_t) \right\|^2$$

$$\leq (1-\frac{\beta}{2})\|\mathbf{v}_t - \nabla F_{\lambda,\nu}(\mathbf{w}_{t-1})\|^2 + \frac{3\alpha^2 L_{F_{\lambda,\nu}}^2}{\beta}\|\mathbf{v}_t\|^2 + 3\beta\left\| \frac{1}{\nu}(\mathbf{w}_t - \hat{\mathbf{z}}_t) - \nabla F_{\lambda,\nu}(\mathbf{w}_t) \right\|^2 \qquad (31)$$

Since $\alpha \leq \frac{1}{2L_{F_{\lambda,\nu}}}$, by Lemma 2 in [46], we can obtain a result similar to (18) in Lemma B.1, that is,

$$F_{\lambda,\nu}(\mathbf{w}_{t+1}) \leq F_{\lambda,\nu}(\mathbf{w}_t) + \frac{\alpha}{2}\|\mathbf{v}_{t+1} - \nabla F_{\lambda,\nu}(\mathbf{w}_t)\|^2 - \frac{\alpha}{2}\|\nabla F_{\lambda,\nu}(\mathbf{w}_t)\|^2 - \frac{\alpha}{4}\|\mathbf{v}_{t+1}\|^2 \qquad (32)$$

Multiplying both sides of (31) by $\frac{\alpha}{\beta}$ and adding them to both sides of (32), we have

$$\frac{\alpha}{2}\|\nabla F_{\lambda,\nu}(\mathbf{w}_t)\|^2$$

$$\leq F_{\lambda,\nu}(\mathbf{w}_t) - F_{\lambda,\nu}(\mathbf{w}_{t+1}) + (\frac{\alpha}{\beta} - \frac{\alpha}{2})\left( \|\mathbf{v}_t - \nabla F_{\lambda,\nu}(\mathbf{w}_{t-1})\|^2 - \|\mathbf{v}_{t+1} - \nabla F_{\lambda,\nu}(\mathbf{w}_t)\|^2 \right)$$

$$- \frac{\alpha}{4}\|\mathbf{v}_{t+1}\|^2 + \frac{\alpha^3 L_{F_{\lambda,\nu}}^2}{\beta^2}\|\mathbf{v}_t\|^2 + 3\alpha\left\| \frac{1}{\nu}(\mathbf{w}_t - \hat{\mathbf{z}}_t) - \nabla F_{\lambda,\nu}(\mathbf{w}_t) \right\|^2$$

$$\leq F_{\lambda,\nu}(\mathbf{w}_t) - F_{\lambda,\nu}(\mathbf{w}_{t+1}) + (\frac{\alpha}{\beta} - \frac{\alpha}{2})\left( \|\mathbf{v}_t - \nabla F_{\lambda,\nu}(\mathbf{w}_{t-1})\|^2 - \|\mathbf{v}_{t+1} - \nabla F_{\lambda,\nu}(\mathbf{w}_t)\|^2 \right)$$

$$\frac{\alpha}{4}\left( \|\mathbf{v}_t\|^2 - \|\mathbf{v}_{t+1}\|^2 \right) + \frac{3\alpha}{\nu^2}\left\| \hat{\mathbf{z}}_t - \text{prox}_{\nu F_\lambda}(\mathbf{w}_t) \right\|^2,$$

where the second inequality is because $\nabla F_{\lambda,\nu}(\mathbf{w}_t) = \frac{1}{\nu}(\mathbf{w}_t - \text{prox}_{\nu F_\lambda}(\mathbf{w}_t))$ and $\alpha \leq \frac{\beta}{2L_{F_{\lambda,\nu}}}$ so $\frac{\alpha}{4} \geq \frac{\alpha^3 L_{F_{\lambda,\nu}}^2}{\beta^2}$.

By Theorem C.1, for $\epsilon' = \frac{\mu\nu^2\epsilon^2}{36}$, there exists $\theta \in (0,1)$ with $1-\theta = O(\epsilon^2)$ such that, by setting $\eta = \frac{1-\theta}{\theta\mu}$ and $\gamma = \frac{(1-\theta)n}{B_1}$, inner loop of Algorithm 2(i.e. ALEXR) guarantees $\mathbb{E}\left\| \hat{\mathbf{z}}_t - \text{prox}_{\nu F_\lambda}(\mathbf{w}_t) \right\|_2^2 \leq \frac{\nu^2\epsilon^2}{18}$ for any $t$ after $K_t = \tilde{O}\left( \frac{nL_f}{B_1 B_2\epsilon^2} + \frac{L_f}{\epsilon^2} \right)$ iterations.

Recall that $\mathbf{v}_0 = 0$ and $\mathbf{w}_{-1} = \mathbf{w}_0$. Summing the inequality above over $t = 0, \cdots T - 1$ and dividing both sides by $\frac{\alpha T}{2}$, we have:

$$\mathbb{E}[\text{dist}(0, \partial F_\lambda(\text{prox}_{\nu F_\lambda}(\mathbf{w}_\tau)))^2] = \frac{1}{T} \sum_{t=0}^{T-1} \mathbb{E} \|\nabla F_{\lambda,\nu}(\mathbf{w}_t)\|^2$$

$$\leq \frac{2\mathbb{E}(F_{\lambda,\nu}(\mathbf{w}_0) - F_{\lambda,\nu}(\mathbf{w}_T))}{\alpha T} + \frac{\left(\frac{2}{\beta} - 1\right) \|\nabla F_{\lambda,\nu}(\mathbf{w}_0)\|^2}{T} + \frac{6}{\nu^2 T} \sum_{t=0}^{T-1} \mathbb{E} \|\hat{\mathbf{z}}_t - \text{prox}_{\nu F_\lambda}(\mathbf{w}_t)\|^2$$

$$\leq \frac{2\mathbb{E}(F_{\lambda,\nu}(\mathbf{w}_0) - F_{\lambda,\nu}(\mathbf{w}_T))}{\alpha T} + \frac{\left(\frac{2}{\beta} - 1\right) \|\nabla F_{\lambda,\nu}(\mathbf{w}_0)\|^2}{T} + \frac{\epsilon^2}{3},$$

By setting $T = O(\epsilon^{-2})$, we'll have $\mathbb{E}[\text{dist}(0, \partial F_\lambda(\text{prox}_{\nu F_\lambda}(\mathbf{w}_\tau)))^2] \leq \epsilon^2$, meaning that $\mathbf{w}_\tau$ is a nearly $\epsilon$-stationary point of (4) in expectation. Recall that $L_f = O(\frac{1}{\epsilon})$ so $K_t = O(\epsilon^{-3})$. Hence, the total complexity for finding such a solution is $\sum_{t=0}^{T-1} K_t = \tilde{O}(\epsilon^{-5})$. $\qquad\square$

*Proof of Corollary 4.11.* When $g_i$ is $L_g$-smooth, $F_\lambda(\cdot)$ is $L_F$-smooth where $L_F = O(\frac{1}{\lambda}) = O(\frac{1}{\epsilon})$ is defined in lemma A.1. By Theorem 4.10,

$$\mathbb{E}\|\nabla F_\lambda(\text{prox}_{\nu F_\lambda}(\mathbf{w}_\tau))\| = \frac{1}{\nu} \mathbb{E}\|\text{prox}_{\nu F_\lambda}(\mathbf{w}_\tau) - \mathbf{w}_\tau\| \leq O(\epsilon).$$

According to Theorem C.1, for $\epsilon' = O(\epsilon^4)$, there exists $\theta \in (0, 1)$ with $1 - \theta = O(\epsilon^4)$ such that, by setting $\eta = \frac{1-\theta}{\theta} L_f$ and $\gamma = \frac{(1-\theta)n}{B_1}$ in inner loop of Algorithm 2(i.e. ALEXR), $\hat{\mathbf{w}}_\tau = \text{ALEXR}(\mathbf{w}_\tau, K)$ satisfies $\frac{\mu}{2} \mathbb{E} \|\hat{\mathbf{w}}_\tau - \text{prox}_{\nu F_\lambda}(\mathbf{w}_\tau)\|^2 \leq \epsilon^4$ and thus

$$\mathbb{E} \|\hat{\mathbf{w}}_\tau - \text{prox}_{\nu F_\lambda}(\mathbf{w}_\tau)\| \leq O(\epsilon^2)$$

after $K = \tilde{O}\left(\frac{nL_f}{B_1 B_2 \epsilon^4} + \frac{L_f}{\epsilon^4}\right)$ iterations. Then we have

$$\mathbb{E}\|\nabla F_\lambda(\hat{\mathbf{w}}_\tau)\| \leq \mathbb{E}\|\nabla F_\lambda(\text{prox}_{\nu F_\lambda}(\mathbf{w}_\tau))\| + L_F \mathbb{E} \|\text{prox}_{\nu F_\lambda}(\mathbf{w}_\tau) - \hat{\mathbf{w}}_\tau\| \leq O(\epsilon),$$

which means $\hat{\mathbf{w}}_\tau$ is an $\epsilon$-stationary solution of (3). By Proposition 4.4, $\hat{\mathbf{w}}_\tau$ is a nearly $\epsilon$-stationary solution of (1). By Theorem 4.10, $\mathbf{w}_\tau$ is found within complexity $O(\epsilon^{-5})$ and $\hat{\mathbf{w}}_\tau = \text{ALEXR}(\mathbf{w}_\tau, K)$ has complexity $K = \tilde{O}\left(\frac{nL_f}{B_1 B_2 \epsilon^4} + \frac{L_f}{\epsilon^4}\right)$. The total complexity for computing $\hat{\mathbf{w}}_\tau$ is still $\tilde{O}\left(\frac{n}{B_1 B_2 \epsilon^5} + \frac{1}{\epsilon^5}\right)$. $\qquad\square$

# D  Convergence Analysis of SONEX and ALEXR2 applying to Constraint Optimization Problems

In this section, we present the complexity analysis of Algorithm 2 when it is applied to (8) to solve the constrained optimization problem 7.

## D.1  Proof of Proposition 5.2 and its variant

*Proof of Proposition 5.2.* By the definition of $f_\lambda(\cdot)$, we have

$$\nabla f_\lambda(\cdot) = \frac{1}{\lambda} \min\{[\cdot]_+, \lambda\rho\}.$$

Suppose $\mathbf{w}$ is a nearly $\epsilon$-stationary point of (9), which means there exists $\hat{\mathbf{w}}$ such that $\|\mathbf{w} - \hat{\mathbf{w}}\| \leq \epsilon$ and $\text{dist}(0, \partial \Phi_\lambda(\hat{\mathbf{w}})) \leq \epsilon$. This means there exist $\mathbf{h}_i(\hat{\mathbf{w}}) \in \partial g_i(\hat{\mathbf{w}})$ and

$$\nu_i = \frac{1}{m} \nabla f_\lambda(g_i(\hat{\mathbf{w}})) = \frac{1}{\lambda m} \min\{[g_i(\hat{\mathbf{w}})]_+, \lambda\rho\}$$

for $i = 0, 1, \ldots, m$ such that

$$\|\mathbf{h}_0(\hat{\mathbf{w}}) + \mathbf{J}(\hat{\mathbf{w}})\boldsymbol{\nu}\| \leq \epsilon \tag{33}$$

where $\mathbf{J}(\hat{\mathbf{w}}) = [\mathbf{h}_1(\hat{\mathbf{w}}), \ldots, \mathbf{h}_m(\hat{\mathbf{w}})] \in \mathbb{R}^{d \times m}$ and $\boldsymbol{\nu} = (\nu_1, \cdots, \nu_m)^\top \in \mathbb{R}^m$.

Suppose $\max_{i=1,\ldots,m} g_i(\hat{\mathbf{w}}) > \epsilon$. Then there exists $k$ such that $[g_k(\hat{\mathbf{w}})]_+ > \epsilon$. Recall that $\lambda = \frac{\epsilon}{\rho}$. We have

$$\nu_k = \frac{1}{\lambda m} \min\{[g_k(\hat{\mathbf{w}})]_+, \lambda\rho\} = \frac{1}{\lambda m}\epsilon = \frac{\rho}{m}.$$

By Assumption 5.1, we have

$$\|\mathbf{h}_0(\hat{\mathbf{w}}) + \mathbf{J}(\hat{\mathbf{w}})\boldsymbol{\nu}\| \geq \|\mathbf{J}(\hat{\mathbf{w}})\boldsymbol{\nu}\| - \|\mathbf{h}_0(\hat{\mathbf{w}})\| \geq \sigma_{min}(\mathbf{J}(\hat{\mathbf{w}})) \|\boldsymbol{\nu}\| - \|\mathbf{h}_0(\hat{\mathbf{w}})\| \geq \frac{\delta\rho}{m} - C_g > 1,$$

which contradicts with (33). Therefore, we must have

$$\max_{i=1,\ldots,m} g_i(\hat{\mathbf{w}}) \leq \epsilon. \tag{34}$$

Finally, when $\max_{i=1,\ldots,m} g_i(\hat{\mathbf{w}}) \leq \epsilon$, we have for $\forall i = 1, 2, \cdots, m$

$$|g_i(\hat{\mathbf{w}})\nu_i| \leq \sum_{i=1}^m |g_i(\hat{\mathbf{w}})\nu_i| = \sum_{i=1}^m [g_i(\hat{\mathbf{w}})]_+ \frac{\min\{[g_i(\hat{\mathbf{w}})]_+, \lambda\rho\}}{\lambda m} \leq \frac{\rho}{m} \sum_{i=1}^m [g_i(\hat{\mathbf{w}})]_+ \leq O(\epsilon). \tag{35}$$

With (33), (34) and (35), $\hat{\mathbf{w}}$ is an $\epsilon$-KKT point of (7) so $\mathbf{w}$ is a nearly $\epsilon$-KKT point of (7). $\qquad\square$

Proposition 5.2 is given when $\mathbf{w}$ is a deterministic nearly $\epsilon$-stationary solution to (9). However, the solution $\mathbf{w}$ found by our algorithms is only a nearly $\epsilon$-stationary solution to (9) in expectation. This means there exists $\hat{\mathbf{w}}$ such that $\mathbb{E}\|\mathbf{w} - \hat{\mathbf{w}}\| \leq \epsilon$ and $\mathbb{E}\mathrm{dist}(0, \partial\Phi_\lambda(\hat{\mathbf{w}})) \leq \epsilon$. In this case, we cannot prove the four inequalities in the conclusion of Proposition 5.2 hold deterministically. Instead, we can only show that the first two inequality hold in expectation while the last two hold in high probabilities. We present this variant of Proposition 5.2 below with its proof.

**Proposition D.1.** *Suppose Assumptions 3.2(A3) and 5.1 hold. If $\rho > \frac{m(C_g+1)}{\delta}$ and $\lambda = \frac{\epsilon}{\rho}$, a nearly $\epsilon$-stationary solution $\mathbf{w}$ to (9) in expectation is also a nearly $O(\epsilon)$-KKT solution to the original problem (7) in the sense that there exist $\hat{\mathbf{w}}$ and $\nu_i \geq 0$ for $i = 1, \ldots, m$ such that $\mathbb{E}\|\mathbf{w} - \hat{\mathbf{w}}\| \leq \epsilon$ and $\mathbb{E}\mathrm{dist}(0, \partial g_0(\hat{\mathbf{w}}) + \sum_{i=1}^m \partial g_i(\hat{\mathbf{w}})\nu_i) \leq O(\epsilon)$ and it holds with probability $1 - O(\epsilon)$ that $\max_{i=1,\ldots,m} g_i(\hat{\mathbf{w}}) \leq O(\epsilon)$ and $|g_i(\hat{\mathbf{w}})\nu_i| \leq O(\epsilon), \forall i = 1, 2, \cdots, m$.*

*Proof.* By the definition of $f_\lambda(\cdot)$, we have

$$\nabla f_\lambda(\cdot) = \frac{1}{\lambda} = \min\{[\cdot]_+, \lambda\rho\}.$$

Suppose $\mathbf{w}$ is a nearly $\epsilon$-stationary point of (9) in expecation, which means there exists $\hat{\mathbf{w}}$ such that $\mathbb{E}\|\mathbf{w} - \hat{\mathbf{w}}\| \leq \epsilon$ and $\mathbb{E}\mathrm{dist}(0, \partial\Phi_\lambda(\hat{\mathbf{w}})) \leq \epsilon$. This means there exist $\mathbf{h}_i(\hat{\mathbf{w}}) \in \partial g_i(\hat{\mathbf{w}})$ and

$$\nu_i = \frac{1}{m}\nabla f_\lambda(g_i(\hat{\mathbf{w}})) = \frac{1}{\lambda m} \min\{[g_i(\hat{\mathbf{w}})]_+, \lambda\rho\}$$

for $i = 0, 1, \ldots, m$ such that

$$\mathbb{E}\|\mathbf{h}_0(\hat{\mathbf{w}}) + \mathbf{J}(\hat{\mathbf{w}})\boldsymbol{\nu}\| \leq \epsilon$$

where $\mathbf{J}(\hat{\mathbf{w}}) = [\mathbf{h}_1(\hat{\mathbf{w}}), \ldots, \mathbf{h}_m(\hat{\mathbf{w}})] \in \mathbb{R}^{d \times m}$ and $\boldsymbol{\nu} = (\nu_1, \cdots, \nu_m)^\top \in \mathbb{R}^m$.

Suppose $\max_{i=1,\ldots,m} g_i(\hat{\mathbf{w}}) > \epsilon$. Then there exists $k$ such that $[g_k(\hat{\mathbf{w}})]_+ > \epsilon$. Recall that $\lambda = \frac{\epsilon}{\rho}$. We have

$$\nu_k = \frac{1}{\lambda m} \min\{[g_k(\hat{\mathbf{w}})]_+, \lambda\rho\} = \frac{1}{\lambda m}\epsilon = \frac{\rho}{m}.$$

By Assumption 5.1, we have

$$\|\mathbf{h}_0(\hat{\mathbf{w}}) + \mathbf{J}(\hat{\mathbf{w}})\boldsymbol{\nu}\| \geq \|\mathbf{J}(\hat{\mathbf{w}})\boldsymbol{\nu}\| - \|\nabla g_0(\hat{\mathbf{w}})\| \geq \sigma_{min}(\mathbf{J}(\hat{\mathbf{w}})) \|\boldsymbol{\nu}\| - \|\mathbf{h}_0(\hat{\mathbf{w}})\| \geq \frac{\delta\rho}{m} - C_g > 1$$

Therefore,

$$\epsilon \geq \mathbb{E} \left\| \mathbf{h}_0(\hat{\mathbf{w}}) + \mathbf{J}(\hat{\mathbf{w}})\boldsymbol{\nu} \right\|$$

$$= \mathbb{E} \left[ \left\| \mathbf{h}_0(\hat{\mathbf{w}}) + \mathbf{J}(\hat{\mathbf{w}})\boldsymbol{\nu} \right\| \,\Big|\, \max_{i=1,\dots,m} g_i(\hat{\mathbf{w}}) > \epsilon \right] \mathrm{Prob}(\max_{i=1,\dots,m} g_i(\hat{\mathbf{w}}) > \epsilon)$$

$$+ \mathbb{E} \left[ \left\| \mathbf{h}_0(\hat{\mathbf{w}}) + \mathbf{J}(\hat{\mathbf{w}})\boldsymbol{\nu} \right\| \,\Big|\, \max_{i=1,\dots,m} g_i(\hat{\mathbf{w}}) \leq \epsilon \right] \mathrm{Prob}(\max_{i=1,\dots,m} g_i(\hat{\mathbf{w}}) \leq \epsilon)$$

$$\geq \mathrm{Prob}(\max_{i=1,\dots,m} g_i(\hat{\mathbf{w}}) > \epsilon) \left( \frac{\delta\rho}{m} - C_g \right)$$

As a result,

$$\mathrm{Prob}(\max_{i=1,\dots,m} g_i(\hat{\mathbf{w}}) > \epsilon) \leq \epsilon \left( \frac{\delta\rho}{m} - C_g \right)^{-1} = O(\epsilon). \tag{36}$$

Finally, when $\max_{i=1,\dots,m} g_i(\hat{\mathbf{w}}) \leq \epsilon$, we have for $\forall i = 1, 2, \cdots, m$

$$|g_i(\hat{\mathbf{w}})\nu_i| \leq \sum_{i=1}^{m} |g_i(\hat{\mathbf{w}})\nu_i| = \sum_{i=1}^{m} [g_i(\hat{\mathbf{w}})]_+ \frac{\min\{[g_i(\hat{\mathbf{w}})]_+, \lambda\rho\}}{\lambda m} \leq \frac{\rho}{m} \sum_{i=1}^{m} [g_i(\hat{\mathbf{w}})]_+ \leq O(\epsilon).$$

It then follows from (36) that for $\forall i = 1, 2, \cdots, m$

$$\mathrm{Prob}\left( |g_i(\hat{\mathbf{w}})\nu_i| \geq O(\epsilon) \right) \leq O(\epsilon).$$

$\square$

## D.2 Sketch Proof of Theorem 5.3

Since the proof is almost the same as that of Theorem 4.10, we only highlight the difference in the proof.

We first consider the case where $\{g_i\}_{i=0}^{m}$ are weakly convex. We will slightly modify Algorithm 2 to solve the following problem

$$\min_{\mathbf{w} \in \mathbb{R}^d} \left\{ \Phi_{\lambda,\nu}(\mathbf{w}) := \min_{\mathbf{z} \in \mathbb{R}^d} \left\{ \Phi_\lambda(\mathbf{z}) + \frac{1}{2\nu} \|\mathbf{z} - \mathbf{w}\|^2 \right\} \right\} \tag{37}$$

$$= \min_{\mathbf{w} \in \mathbb{R}^d} \left\{ \min_{\mathbf{z} \in \mathbb{R}^d} \max_{y_i \in [0,\rho]} \left\{ g_0(\mathbf{z}) + \frac{1}{m} \sum_{i=1}^{m} y_i g_i(\mathbf{z}) - \frac{\lambda}{2} y_i^2 + \frac{1}{2\nu} \|\mathbf{z} - \mathbf{w}\|^2 \right\} \right\}, \tag{38}$$

where $\Phi_{\lambda,\nu}(\mathbf{w})$ is defined in (9). According to Proposition D.2 in [6], the second (compositional) term in $\Phi_\lambda(\mathbf{w})$ is $(\rho\rho_g)$-weakly convex. Hence, $\Phi_\lambda(\mathbf{w})$ is $\rho_{\Phi_\lambda}$-weakly convex with $\rho_{\Phi_\lambda} = \rho_g + \rho\rho_g$. If $\nu < \rho_{\Phi_\lambda}^{-1}$, $\Phi_{\lambda,\nu}(\mathbf{w})$ is $(\frac{1}{\nu} - \rho_{\Phi_\lambda})$-strongly convex. As done in Section (4.2), in the $t$th iteration of Algorithm 2, we apply ALEXR to solve the inner min-max problem in (38) with $\mathbf{w} = \mathbf{w}_t$, namely,

$$\min_{\mathbf{z}} \max_{\mathbf{y} \in [0,\rho]^m} \left\{ L_t(\mathbf{z}, \mathbf{y}) := g_0(\mathbf{z}) + \frac{1}{m} \sum_{i=1}^{m} \left( \mathbf{y}_i g_i(\mathbf{w}) - \frac{\lambda}{2} y_i^2 \right) + \frac{1}{2\nu} \|\mathbf{z} - \mathbf{w}_t\|_2^2 \right\}.$$

we only need to replace $G_{t,k}$ in Algorithm 2 to

$$G_{t,k} = G_{t,k}^0 + G_{t,k}^1 = \partial g_0(\mathbf{z}_{t,k}, \mathcal{B}_{0,2}^{t,k}) + \frac{1}{|\mathcal{B}_1^{t,k}|} \sum_{i \in \mathcal{B}_1^{t,k}} [\partial g_i(\mathbf{z}_{t,k}, \tilde{\mathcal{B}}_{i,2}^{t,k})]^\top \mathbf{y}_{t,k+1}^{(i)}, \tag{39}$$

where $\mathcal{B}_{0,2}^{t,k}$ is a batch of data sampled from the distribution of $\xi_0$.

We then consider the case where $\{g_i\}_{i=0}^{m}$ are smooth. In this case, we can directly apply Algorithm 1 to (9), which is different from (3) only in the extra term $g_0$. To handle this difference, we only need to replace $G_t$ in Algorithm 1 to

$$G_t = G_t^0 + G_t^1 = \nabla g_0(\mathbf{w}_t, \mathcal{B}_{0,2}^t) + \frac{1}{|\mathcal{B}_1^t|} \sum_{i \in \mathcal{B}_1^t} \nabla f_{i,\lambda}(u_{t,i}) \nabla g_i(\mathbf{w}_t, \mathcal{B}_{i,2}^t) \tag{40}$$

where $\mathcal{B}_{0,2}^t$ is a batch of data sampled from the distribution of $\xi_0$. With these changes, we can prove Theorem 5.3 as follows.

*Sketch Proof of Theorem 5.3.* **When $\{g_i\}_{i=0}^m$ are weakly convex:**

For simplicity of notation, we drop the dependence on $t$ in the notation of all variables and parameters in the proof below. First, we need to modify Lemma 5 in [5]. We have

$$L_t(\mathbf{z}_{k+1}, \mathbf{y}) - L_t(\mathbf{z}, \bar{\mathbf{y}}_{k+1}) \tag{41}$$

$$\leq \frac{\gamma^{-1}}{m} U_{\bar{f}_i^*}(\mathbf{y}, \mathbf{y}_k) - \frac{\gamma^{-1}+1}{m} U_{\bar{f}_i^*}(\mathbf{y}, \bar{\mathbf{y}}_{k+1}) - \frac{\gamma^{-1}}{m} U_{\bar{f}_i^*}(\bar{\mathbf{y}}_{k+1}, \mathbf{y}_k)$$

$$+ \frac{1}{m} \sum_{i=1}^m \left\langle g_i(\mathbf{z}_{k+1}) - \tilde{g}_k^{(i)}, y_i - \bar{y}_{k+1}^{(i)} \right\rangle + \frac{1}{2\eta} \|\mathbf{z} - \mathbf{z}_k\|_2^2 \tag{42}$$

$$- \left( \frac{1}{2\eta} + \frac{1}{4\nu} \right) \|\mathbf{z} - \mathbf{z}_{k+1}\|_2^2 - \frac{1}{2\eta} \|\mathbf{z}_{k+1} - \mathbf{z}_k\|_2^2$$

$$+ \frac{1}{m} \sum_{i=1}^m \left\langle g_i(\mathbf{z}_{k+1}) - g_i(\mathbf{z}), \bar{y}_{k+1}^{(i)} \right\rangle - \left\langle G_k^1, \mathbf{z}_{k+1} - \mathbf{z} \right\rangle \tag{43}$$

$$\textcolor{blue}{+ g_0(\mathbf{z}_{k+1}) - g_0(\mathbf{z}) - \left\langle G_k^0, \mathbf{z}_{k+1} - \mathbf{z} \right\rangle,}$$

where the blue terms above are included due to the modifications to Algorithm 2. Next, we modify Lemma 8 in [5] to handle the additional terms above. Denote that $\Delta_k^0 := g_0'(\mathbf{z}_k) - G_k^0$. Note that

$$- \left\langle G_k^0, \mathbf{z}_{k+1} - \mathbf{z} \right\rangle = - \left\langle g_0'(\mathbf{z}_k), \mathbf{z}_{k+1} - \mathbf{z} \right\rangle + \left\langle \Delta_k^0, \mathbf{z}_{k+1} - \mathbf{z} \right\rangle,$$

where $g_0'(\mathbf{z}_k) \in \partial g(\mathbf{z}_k)$ is a subgradient at $\mathbf{z}_k$. The term $\left\langle \Delta_k^0, \mathbf{z}_{k+1} - \mathbf{z} \right\rangle$ can be handled in the same way as in (24) in [5]. Due to the weak convexity of $g_0$, we have

$$g_0(\mathbf{z}) - g_0(\mathbf{z}_k) \geq \left\langle g_0'(\mathbf{z}_k), \mathbf{z} - \mathbf{z}_k \right\rangle - \frac{\rho_g}{2} \|\mathbf{z} - \mathbf{z}_k\|_2^2.$$

Then, the remaining blue terms in (41) can be upper bounded as

$$g_0(\mathbf{z}_{k+1}) - g_0(\mathbf{z}) - \left\langle g_0'(\mathbf{z}_k), \mathbf{z}_{k+1} - \mathbf{z} \right\rangle$$

$$= g_0(\mathbf{z}_{k+1}) - g_0(\mathbf{z}_k) + g_0(\mathbf{z}_k) - g_0(\mathbf{z}) - \left\langle g_0'(\mathbf{z}_k), \mathbf{z}_{k+1} - \mathbf{z} \right\rangle$$

$$\leq C_g \|\mathbf{z}_{k+1} - \mathbf{z}_k\|_2 + \left\langle g_0'(\mathbf{z}_k), \mathbf{z}_k - \mathbf{z} \right\rangle - \left\langle g_0'(\mathbf{z}_k), \mathbf{z}_{k+1} - \mathbf{z} \right\rangle + \frac{\rho_g}{2} \|\mathbf{z} - \mathbf{z}_k\|_2^2$$

$$\leq \frac{4C_g^2}{\frac{1}{\eta} + \frac{1}{\nu}} + \frac{\left( \frac{1}{\eta} + \frac{1}{\nu} \right) \|\mathbf{z}_{k+1} - \mathbf{z}_k\|_2^2}{16} + \left\langle g_0'(\mathbf{z}_k), \mathbf{z}_k - \mathbf{z}_{k+1} \right\rangle + \frac{\rho_g}{2} \|\mathbf{z} - \mathbf{z}_k\|_2^2$$

$$\leq \frac{8C_g^2}{\frac{1}{\eta} + \frac{1}{\nu}} + \frac{\left( \frac{1}{\eta} + \frac{1}{\nu} \right) \|\mathbf{z}_{k+1} - \mathbf{z}_k\|_2^2}{8} + \frac{\rho_g}{2} \|\mathbf{z} - \mathbf{z}_k\|_2^2.$$

Let $(\mathbf{z}_*, \mathbf{y}_*)$ be the saddle point of $L_t(\mathbf{z}, \mathbf{y})$. Then, we can get

$$\mathbb{E}[L_t(\mathbf{z}_{k+1}, \mathbf{y}_*) - L_t(\mathbf{z}_*, \bar{\mathbf{y}}_{k+1})]$$

$$\leq \frac{\gamma^{-1} + \left( 1 - \frac{B_1}{m} \right)}{B_1} \mathbb{E}[U_{\bar{f}_i^*}(\mathbf{y}_*, \mathbf{y}_k)] - \frac{\gamma^{-1}+1}{B_1} \mathbb{E}[U_{\bar{f}_i^*}(\mathbf{y}_*, \mathbf{y}_{k+1})]$$

$$+ \left( \frac{1}{2\eta} + \frac{\sqrt{d_1} C_f \rho_g}{2} + \frac{\rho_g}{2} \right) \mathbb{E} \|\mathbf{z}_* - \mathbf{z}_k\|_2^2 - \left( \frac{1}{2\eta} + \frac{1}{4\nu} \right) \mathbb{E} \|\mathbf{z}_* - \mathbf{z}_{k+1}\|_2^2$$

$$- \left( \frac{1}{m\gamma} - \frac{\lambda_2 + \lambda_3 \theta}{\lambda m} \right) \mathbb{E} \left[ U_{\bar{f}_i^*}(\bar{\mathbf{y}}_{k+1}, \mathbf{y}_k) \right] - \left( \frac{1}{2\eta} - \frac{C_g^2}{2\lambda_2} - \frac{3}{8}(\frac{1}{\eta} + \frac{1}{\nu}) \right) \mathbb{E} \|\mathbf{z}_{k+1} - \mathbf{z}_k\|_2^2$$

$$+ \frac{\theta C_g^2}{2\lambda_3} \mathbb{E} \|\mathbf{z}_k - \mathbf{z}_{k-1}\|_2^2 + \mathbb{E}[\Gamma_{k+1} - \theta \Gamma_k] + \frac{2(1+2\theta)\sigma_0^2}{B_2 \lambda(1+\gamma^{-1})} + \frac{\frac{C_f^2 \sigma_1^2}{B_2} + \frac{C_f^2 C_g^2}{B_1} + 8 C_f^2 C_g^2 + 8 C_g^2}{\frac{1}{\eta} + \frac{1}{\nu}}. \tag{44}$$

Following the similar proof as in Theorem C.1, the modified inner loop of ALEXR2 described in Section 5 can guarantee $\frac{\mu}{2} \mathbb{E} \left\| \hat{\mathbf{z}}_t - \text{prox}_{\nu F_\lambda}(\mathbf{w}_t) \right\|_2^2 \leq \epsilon'$ for any $t$ after $K_t = \tilde{O}(\frac{m}{\lambda B_2 B_1 \epsilon'})$. Because

$\lambda = \frac{\epsilon}{\rho}$ and $\rho > \frac{C_g}{\delta}$ as specified in Proposition D.1, we have $K_t = \tilde{O}(\frac{C_g^2 m}{B_2 B_1 \delta^2 \epsilon \epsilon'})$. For outer loop, we directly leverage the result from Appendix C, i.e.

$$T = O(\max\{\rho_g + \rho\rho_g, \rho C_g\}\epsilon^{-2}) = O(\max\{\rho_g + \frac{C_g \rho_g}{\delta}, \frac{C_g^2}{\delta}\}\epsilon^{-2})$$

By setting $\epsilon' = \frac{\epsilon^2}{\rho_g + \rho\rho_g} = \frac{\epsilon^2}{\rho_g + \frac{C_g \rho_g}{\delta}}$, the total iteration will be

$$\tilde{\mathcal{O}}\left(\frac{m\rho_g^2}{B_2 B_1 \delta^4 \epsilon^5}\right)$$

**When $\{g_i\}_{i=0}^m$ are smooth:**

The proof follows almost the same procedure as the proof of Theorem 4.7. In particular, since $\mathbf{v}_t$ and $G_t$ will contain the additional stochastic gradient from $g_0$, we need to replace (14) by

$$V_{t+1} := \|\mathbf{v}_{t+1} - \nabla\Phi_\lambda(\mathbf{w}_t)\|^2 \tag{45}$$

while $U_{t+1}$, $\mathbf{u}_t$ and $\mathbf{g}(\mathbf{w}_t)$ are still defined as in (15), (16) and (17). Like Lemma B.1, it holds that

$$\Phi_\lambda(\mathbf{w}_{t+1}) \le \Phi_\lambda(\mathbf{w}_t) + \frac{\eta}{2}V_{t+1} - \frac{\eta}{2}\|\nabla\Phi_\lambda(\mathbf{w}_t)\|^2 - \frac{\eta}{4}\|\mathbf{v}_{t+1}\|^2, \tag{46}$$

if $\eta \le \frac{1}{2L_F}$. By a proof similar to Lemma 3 in [2], we can still show (19) for $V_{t+1}$ in the new definition in (45). Moreover, (20) and (21) still hold as they are not related to $g_0$. Finally, we can still take the weighted summation of both sides (46), (19), (20), and (21) as in the proof of Theorem 4.7 to show that

$$\frac{1}{T}\sum_{t=0}^{T-1}\|\nabla\Phi_\lambda(\mathbf{w}_t)\|^2 \le \frac{1}{T}O\left(\frac{m}{B_1\sqrt{B_2}\epsilon^3} + \frac{1}{\min\{B_1, B_2\}\epsilon^2} + \frac{m}{B_1 B_2 \epsilon^3}\right).$$

This means Algorithm 1 finds a nearly $\epsilon$-stationary point of $\Phi_\lambda(\mathbf{w})$ in expectation in $T = O(\frac{m}{B_1\sqrt{B_2}\epsilon^5})$ iterations. Then the conclusion follows from Proposition D.1 and the fact that $C_f = O(\frac{1}{\delta})$.

$\square$

# E   SONEX with Adaptive Learning Rates

In this section we will show that our algorithms can also be easily extended to adaptive learning rate while still retaining the same complexity under an additional assumption introduced later. We consider an adaptive step size update such as adam:

$$\mathbf{w}_{t+1} = \mathbf{w}_t - \tilde{\eta} \circ \mathbf{v}_{t+1}, \tilde{\eta} = \frac{\eta}{\sqrt{\mathbf{s}_t} + \varepsilon}, \mathbf{s}_{t+1} = (1 - \beta')\mathbf{s}_t + \beta' G_t \circ G_t$$

where $G_t$ is the overall gradient estimator mentioned in algorithm 1 and $\circ$ denotes Hadamard(element-wise) product. The following assumption has been justified for the adaptive step size by [48].

**Assumption E.1.** We assume that the adaptive learning rates $\tilde{\eta}$ are (element-wise)bounded, i.e.

$$\eta c_l \le \tilde{\eta}_i \le \eta c_u$$

for any element $\tilde{\eta}_i$ of $\tilde{\eta}$.

Below we provide lemmas which are straightforward extensions of lemma B.1$\sim$ B.4 to adaptive learning setting. The proof is similar except that $\|\mathbf{w}_{t+1} - \mathbf{w}_t\|^2 \le \|\tilde{\eta} \circ \mathbf{v}_{t+1}\|^2 \le \|\eta^2 c_u^2 \mathbf{v}_{t+1}\|^2 = \eta^2 c_u^2 \|\mathbf{v}_{t+1}\|^2$.

**Lemma E.2.** *(Lemma 3 in [48]) Under assumption E.1, for $\mathbf{w}_{t+1} = \mathbf{w}_t - \tilde{\eta} \circ \mathbf{v}_{t+1}$, with $\eta c_l \le \tilde{\eta} \le \eta c_u$ and $\tilde{\eta}L_F \le \frac{c_l}{2c_u^2}$, we have:*

$$F(\mathbf{w}_{t+1}) \le F(\mathbf{w}_t) + \frac{\eta c_u}{2}V_{t+1} - \frac{\eta c_l}{2}\|\nabla F(\mathbf{w}_t)\|^2 - \frac{\eta c_l}{4}\|\mathbf{v}_{t+1}\|^2 \tag{47}$$

*where $V_{t+1} = \|\mathbf{v}_{t+1} - \nabla F(\mathbf{w}_t)\|^2$*

**Lemma E.3.** *If $\beta \leq \frac{2}{7}$, the gradient variance $V_{t+1} := \|\mathbf{v}_{t+1} - \nabla F(\mathbf{w}_t)\|^2$ can be bounded as*

$$\mathbb{E}[V_{t+2}] \leq (1-\beta)\mathbb{E}[V_{t+1}] + \frac{2L_F^2\eta^2 c_u^2 \mathbb{E}\left[\|\mathbf{v}_{t+1}\|^2\right]}{\beta} + \frac{3L_f^2 C_1^2}{n}\mathbb{E}\left[\sum_{\mathbf{z}_i \in \mathcal{B}_1^{t+1}} \|u_{t+2,i} - u_{t+1,i}\|^2\right]$$

(48)

$$+ \frac{2\beta^2 C_f^2(\zeta^2 + C_g^2)}{\min\{B_1, B_2\}} + 5\beta L_f^2 C_1^2 \mathbb{E}[U_{t+2}],$$

*where $U_{t+1} = \frac{1}{n}\|\mathbf{u}_{t+1} - \mathbf{g}(\mathbf{w}_t)\|^2$, $\mathbf{u}_t = [u_{t,1}, \ldots, u_{t,n}]^\top$, $\mathbf{g}(\mathbf{w}_t) = [g_1(\mathbf{w}_t), \ldots, g_n(\mathbf{w}_t)]^\top$.*

**Lemma E.4.** *Suppose $\gamma' = \frac{n-B_1}{B_1(1-\gamma)} + (1-\gamma)$ and $\gamma \leq \frac{1}{2}$ and the MSVR update is used in Algorithm 1. It holds that*

$$\mathbb{E}[U_{t+2}] \leq (1 - \frac{\gamma B_1}{n})\mathbb{E}[U_{t+1}] + \frac{8nC_g^2\eta^2 c_u^2}{B_1}\mathbb{E}\|\mathbf{v}_{t+1}\|^2 + \frac{2B_1\gamma^2\sigma_1^2}{nB_2}.$$

(49)

**Lemma E.5.** *Under the same assumptions as Lemma B.3, it holds that*

$$\mathbb{E}[\|\mathbf{u}_{t+2} - \mathbf{u}_{t+1}\|^2] \leq \frac{4B_1\gamma^2}{n}\mathbb{E}[U_{t+1}] + \frac{9n^2C_g^2\eta^2 c_u^2}{B_1}\mathbb{E}\|\mathbf{v}_{t+1}\|^2 + \frac{2B_1\gamma^2\sigma_0^2}{B_2}.$$

(50)

We then give the following theorem with similar proof technique.

**Theorem E.6.** *Under Assumption 3.2 (A1), 4.5 and 4.6, by setting $\lambda = \Theta(\epsilon), \beta = \Theta(\frac{c_l \min\{B_1,B_2\}\epsilon^2}{c_u}), \gamma = \Theta(\frac{c_l B_2\epsilon^4}{c_u}), \eta = \Theta(\frac{c_l^{1.5}B_1\sqrt{B_2}}{c_u^{2.5}n}\epsilon^3)$, SONEX with $\gamma' = 1 - \gamma + \frac{n-B_1}{B_1(1-\gamma)}$ converges to an approximate $\epsilon$-stationary solution of (1) within $T = O(\frac{c_u^{2.5}n}{c_l^{2.5}B_1\sqrt{B_2}}\epsilon^{-5})$ iterations.*

Combining the above theorem with Theorem 4.4, we obtain the following guarantee:

**Corollary E.7.** *Under Assumption 3.2(A1), 4.3, 4.5 and 4.6, with the same setting as in Theorem E.6, SONEX converges to a nearly $\epsilon$-stationary solution of (1) within $T = O(\frac{c_u^{2.5}n}{c_l^{2.5}B_1\sqrt{B_2}}\epsilon^{-5})$ iterations.*

*Proof of theorem E.6 and corollary E.7.* Since the proof is almost the same as the theorem 4.7 and corollary 4.8, we only highlight the difference.
Let $P > 0$ be a positive constant to be decided later. Taking the weighted summation of both sides (47), (48), (49), and (50) as specified in the following formula

$$\frac{1}{\eta c_u} \times (47) + \frac{1}{\beta} \times (48) + P \times (49) + \frac{3L_f^2 C_1^2}{n\beta} \times (50),$$

we have

$$\frac{1}{\eta c_u}\mathbb{E}F_\lambda(\mathbf{w}_{t+1}) + \frac{1}{\beta}\mathbb{E}V_{t+2} + \left(P - 5L_f^2 C_1^2\right)\mathbb{E}U_{t+2}$$

$$\leq \frac{1}{\eta c_u}\mathbb{E}F_\lambda(\mathbf{w}_t) + \left(\frac{1}{\beta} - \frac{1}{2}\right)\mathbb{E}V_{t+1} + \left(P\left(1 - \frac{\gamma B_1}{n}\right) + \frac{12L_f^2 C_1^2\gamma^2 B_1}{\beta n^2}\right)\mathbb{E}U_{t+1}$$

$$- \left(\frac{c_l}{4c_u} - \frac{2L_F^2\eta^2 c_u^2}{\beta^2} - \frac{8nC_g^2 P\eta^2 c_u^2}{B_1} - \frac{27nL_f^2 C_1^2 C_g^2\eta^2 c_u^2}{B_1\beta}\right)\mathbb{E}\|\mathbf{v}_{t+1}\|^2$$

$$+ \frac{2\beta C_f^2(C_g^2 + \sigma_1^2)}{\min\{B_1, B_2\}} + \frac{2B_1\gamma^2\sigma_1^2 P}{nB_2} + \frac{6L_f^2 C_1^2\gamma^2\sigma_0^2 B_1}{\beta nB_2} - \frac{c_l}{2c_u}\mathbb{E}\|\nabla F_\lambda(\mathbf{w}_t)\|^2$$

We choose $\eta$ such that

$$\frac{c_l}{4c_u} - \frac{2L_F^2\eta^2 c_u^2}{\beta^2} - \frac{8nC_g^2 P\eta^2 c_u^2}{B_1} - \frac{27nL_f^2 C_1^2 C_g^2\eta^2 c_u^2}{B_1\beta} \geq 0.$$

(51)

We follow proof of 4.7 to set

$$P = \frac{n}{\gamma B_1} \left( 5L_f^2 C_1^2 + \frac{12L_f^2 C_1^2 \gamma^2 B_1}{\beta n^2} \right).$$

which satisfies the following linear equation

$$P \left( 1 - \frac{\gamma B_1}{n} \right) + \frac{12L_f^2 C_1^2 \gamma^2 B_1}{\beta n^2} = P - 5L_f^2 C_1^2,$$

Combining the results above gives

$$\frac{1}{\eta c_u} \mathbb{E} F_\lambda(\mathbf{w}_{t+1}) + \frac{1}{\beta} \mathbb{E} V_{t+2} + \left( P - 5L_f^2 C_1^2 \right) \mathbb{E} U_{t+2}$$

$$\leq \frac{1}{\eta c_u} \mathbb{E} F_\lambda(\mathbf{w}_t) + \frac{1}{\beta} \mathbb{E} V_{t+1} + \left( P - 5L_f^2 C_1^2 \right) \mathbb{E} U_{t+1}$$

$$+ \frac{2\beta C_f^2 (C_g^2 + \sigma_1^2)}{\min\{B_1, B_2\}} + \frac{2B_1 \gamma^2 \sigma_1^2 P}{nB_2} + \frac{6L_f^2 C_1^2 \gamma^2 \sigma_0^2 B_1}{\beta n B_2} - \frac{c_l}{2c_u} \mathbb{E} \|\nabla F_\lambda(\mathbf{w}_t)\|^2.$$

Summing both sides of the inequality above over $t = 0, 1, \ldots, T-1$ and organizing term lead to

$$\frac{1}{T} \sum_{t=0}^{T-1} \|\nabla F_\lambda(\mathbf{w}_t)\|^2 \leq \frac{2c_u}{c_l T} \left( \frac{1}{\eta c_u} (F_\lambda(\mathbf{w}_0) - F_\lambda(\mathbf{w}_T)) + \frac{1}{\beta} \mathbb{E} V_1 + \left( P - 5L_f^2 C_1^2 \right) \mathbb{E} U_1 \right)$$

$$+ \frac{c_u}{c_l} \left( \frac{4\beta C_f^2 (C_g^2 + \sigma_1^2)}{\min\{B_1, B_2\}} + \frac{4B_1 \gamma^2 \sigma_1^2 P}{nB_2} + \frac{12L_f^2 C_1^2 \gamma^2 \sigma_0^2 B_1}{\beta n B_2} \right) \quad (52)$$

Note that $L_F = O(\frac{1}{\epsilon})$, $L_f = O(\frac{1}{\epsilon})$ and $P = O(\frac{n}{B_1 \gamma \epsilon^2} + \frac{\gamma}{n\beta \epsilon^2})$. We require that all the terms in the right-hand side to be in the order of $O(\epsilon^2)$. To ensure $\frac{4c_u \beta C_f^2 (C_g^2 + \sigma_1^2)}{c_l \min\{B_1, B_2\}} = O(\epsilon^2)$, we must have $\beta = O(\frac{c_l \min\{B_1, B_2\} \epsilon^2}{c_u})$. To ensure $\frac{4c_u B_1 \gamma^2 \sigma_1^2 P}{c_l n B_2} = \frac{c_u}{c_l} O(\frac{\gamma}{B_2 \epsilon^2} + \frac{B_1 \gamma^3}{n^2 B_2 \beta \epsilon^2}) = O(\epsilon^2)$, we must have $\gamma = O(\frac{c_l B_2 \epsilon^4}{c_u})$. With these orders of $\beta$ and $\gamma$, we also have $P = O(\frac{c_u n}{c_l B_1 B_2 \epsilon^6})$ and $\frac{12c_u L_f^2 C_1^2 \gamma^2 \sigma_0^2 B_1}{c_l \beta n B_2} = O(\epsilon^4)$. Therefore the last three terms in (52) are $O(\epsilon^2)$.

To satisfy (51), we will need to set

$$\eta = O \left( \sqrt{\frac{c_l}{c_u^3}} \min \left\{ \frac{\beta}{L_F}, \sqrt{\frac{B_1}{nP}}, \frac{1}{L_f} \sqrt{\frac{B_1 \beta}{n}} \right\} \right)$$

$$= O \left( \sqrt{\frac{c_l}{c_u^3}} \min \left\{ \frac{c_l}{c_u} \min\{B_1, B_2\} \epsilon^3, \frac{B_1 \sqrt{B_2}}{n} \epsilon^3, \sqrt{\frac{c_l B_1 \min\{B_1, B_2\}}{c_u n}} \epsilon^2 \right\} \right)$$

$$= O \left( \frac{c_l^{1.5} B_1 \sqrt{B_2}}{c_u^{2.5} n} \epsilon^3 \right).$$

We also set the sizes of $\mathcal{B}_{i,2}$ to be $O(\frac{1}{\epsilon^3})$ so that $\mathbb{E} U_1 = O(\epsilon^3)$. Also, it is easy to see that $F_\lambda(\mathbf{w}_0) - F_\lambda(\mathbf{w}_T) = O(1)$ and $\mathbb{E} V_1 = O(1)$. Then the first term in (52) becomes

$$\frac{1}{T} O \left( \frac{1}{\eta c_l} + \frac{1}{\beta} + P\epsilon^3 \right) = \frac{1}{T} O \left( \frac{c_u^{2.5} n}{c_l^{2.5} B_1 \sqrt{B_2} \epsilon^3} + \frac{c_u}{c_l \min\{B_1, B_2\} \epsilon^2} + \frac{c_u n}{c_l B_1 B_2 \epsilon^3} \right)$$

which will become $O(\epsilon^2)$ when $T = O(\frac{c_u^{2.5} n}{c_l^{2.5} B_1 \sqrt{B_2} \epsilon^5})$.

Then the remaining part is just the same as the proof of corollary 4.8. $\qquad \square$

## F   More Experiment Details

### F.1   GDRO with CVaR divergence

**Motivation of GDRO** Modern machine learning models are typically trained under the empirical risk minimization (ERM) framework, which treats all samples in the dataset equally. Although ERM

often achieves strong average performance on test sets drawn from distributions similar to the training data, it can perform poorly on rare or underrepresented subpopulations, i.e., it lacks robustness to distributional shifts. The motivation of GDRO is to address the spurious correlation between features and labels, and distributional shift. GDRO partitions the dataset into groups representing different distributions and applies a robust optimization scheme that assigns more weight on the worst groups. Specifically, GDRO considers the following objective:

$$\min_{\theta} \max_{\mathbf{p} \in \Omega} \frac{1}{n} \sum_{g=1}^{n} p_g \mathbb{E}_{(\mathbf{x},y) \sim \mathcal{D}_g} l(\theta; (\mathbf{x}, y))$$

where $\Omega \subseteq \Delta$ is the set of distribution under consideration and $\Delta$ is the simplex. We consider here a popular choice of $\Omega = \{\mathbf{p} \in \Delta : p_i \leq \frac{1}{k}, \forall i \in [n]\}$, which, by solving the inner-level maximization problem, leads to an equivalent reformulation 10 corresponding to the so-called CVaR divergence [5, 49]. Intuitively GDRO with CVaR divergence aims to minimize averaged loss of the top-k worst groups.

**CAMELYON17-WILDS** [34] is part of the WILDS benchmark suite and consists of histopathology whole-slide images from five medical centers, with the goal of detecting metastatic tissue in lymph node biopsies. Following the WILDS setup, we frame this as a binary classification task on image patches, where the primary challenge lies in distribution shift across hospitals (domains). We construct group with attributes 'hospital' and 'slide' which generates 30 groups.

**Amazon-WILDS** [50] is a text classification dataset derived from Amazon product reviews, where the goal is to predict binary sentiment (positive or negative) based on TF-IDF features of review text. The data spans multiple product categories. We construct group with the attribute 'user' which generates 1252 groups.

**CelebA** [35] is a large-scale facial attribute dataset containing over 200,000 celebrity images annotated with 40 binary attributes. We select 4 attributes 'Attractive', 'Mouth_Slightly_Open', 'Male' and 'Blonde_Hair' and construct 16 groups, where 'Blonde_Hair' also serves as the target attribute for us to do classification.

**Hyperparameter tuning.** We tune the same hyperparameters of different methods from the same candidates as follows for fair comparison. For the three tasks we train the models for 10, 4, 15 epochs with batch size and the number of groups within a mini batch of 256(8), 32(8), 64(4), respectively. We set $\alpha = 0.15$ for all the three dataset. We tune learning rate in {1e-5, 2e-5, 5e-5, 1e-4, 2e-4, 5e-4, 1e-3, 2e-3, 5e-3}, $\lambda$ in {1, 0.1, 0.01}, $\gamma$ and $\beta$ in {0.1, 0.2, 0.5, 0.8} and $\gamma'$ in {0.01, 0.02, 0.05, 0.1, 0.2}. We set weight decay to be 0.01, 0.01, 0.02 for the three tasks, respectively. We use step decay (decay by 0.3x for every 3 epochs), linear decay with 1st epoch warmup, step decay (decay by 0.2x for every 3 epochs) for learning rate for the three tasks, respectively.

### F.2 AUC Maximization with ROC Fairness Constraints

In this part we perform experiment on learning a model with ROC fairness constraint [51] following the same experiment setting as [8]. Suppose the data are divided into two demographic groups $\mathcal{D}_p = \{(\mathbf{a}_i^p, b_i^p)\}_{i=1}^{n_p}$ and $\mathcal{D}_u = \{(\mathbf{a}_i^u, b_i^u)\}_{i=1}^{n_u}$, where $\mathbf{a}$ denotes the input data and $b \in \{1, -1\}$ denotes the class label. A ROC fairness is to ensure the ROC curves for classification of the two groups are the same.

Since the ROC curve is constructed with all possible thresholds, we follow [51] by using a set of thresholds $\Gamma = \{\tau_1, \cdots, \tau_m\}$ to define the ROC fairness. For each threshold $\tau$, we impose a constraint that the false positive rate (FPR) and true positive rate (TPR) of the two groups are close, formulated as follow:

$$h_\tau^+(\mathbf{w}) := \left| \frac{1}{n_p^+} \sum_{i=1}^{n_p} \mathbf{I}\{b_i^p = 1\} \sigma(s_\mathbf{w}(\mathbf{a}_i^p) - \tau) - \frac{1}{n_u^+} \sum_{i=1}^{n_u} \mathbf{I}\{b_i^u = 1\} \sigma(s_\mathbf{w}(\mathbf{a}_i^u) - \tau) \right| - \kappa \leq 0$$

$$h_\tau^-(\mathbf{w}) := \left| \frac{1}{n_p^-} \sum_{i=1}^{n_p} \mathbf{I}\{b_i^p = -1\} \sigma(s_\mathbf{w}(\mathbf{a}_i^p) - \tau) - \frac{1}{n_u^-} \sum_{i=1}^{n_u} \mathbf{I}\{b_i^u = -1\} \sigma(s_\mathbf{w}(\mathbf{a}_i^u) - \tau) \right| - \kappa \leq 0,$$

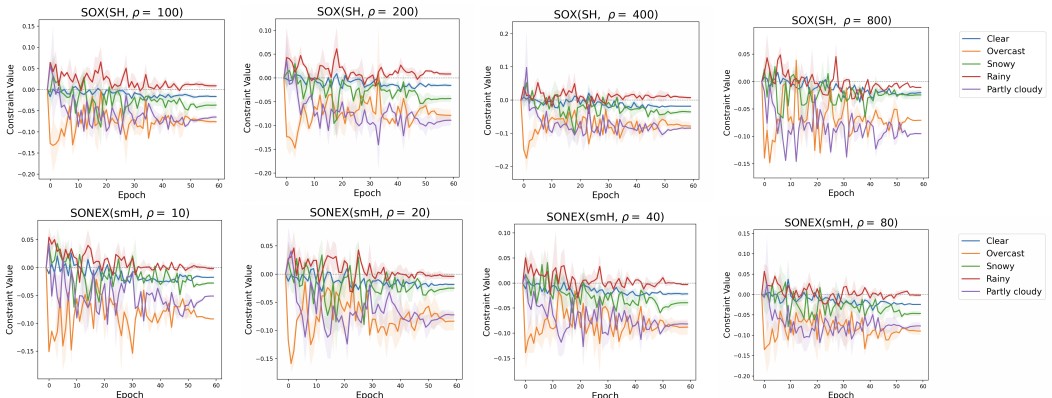

Figure 4: Training curves of 5 constraint values in zero-one loss of different methods for continual learning with non-forgetting constraints when targeting the foggy class. Top: squared-hinge penalty method with different $\rho$; Bottom: smoothed hinge penalty method with different $\rho$.

where $s_{\mathbf{w}}(\cdot)$ denotes a parameterized model, $\sigma(z)$ is the sigmoid function, and $\kappa > 0$ is a tolerance parameter. We use the pairwise AUC loss as the objective function:

$$F(\mathbf{w}) = \frac{-1}{|\mathcal{D}_+||\mathcal{D}_-|} \sum_{\mathbf{x}_i \in \mathcal{D}_+} \sum_{\mathbf{x}_j \in \mathcal{D}_-} \sigma(s(\mathbf{w}, \mathbf{x}_i) - s(\mathbf{w}, \mathbf{x}_j)),$$

where $\mathcal{D}_+$ and $\mathcal{D}_-$ denote the set of positive and negative examples regardless their groups, respectively. We follow [8] to recast original constraint optimization as the following hinge-penalized objective:

$$\min_{\mathbf{w}} F(\mathbf{w}) + \frac{1}{2|\Gamma|} \sum_{\tau \in \Gamma} \beta[h_\tau^+(\mathbf{w})]_+ + \beta[h_\tau^-(\mathbf{w})]_+ \tag{53}$$

It is notable that the penalty terms in the above problem can be formulated as $[h_\tau^+(\mathbf{w})]_+ = f(g(\mathbf{w}; \tau))$, where $f(g) = [|g_1 - g_2| - \kappa]_+$, $g_1(\mathbf{w}; \tau) = \frac{1}{n_p^+} \sum_{i=1}^{n_p} \mathbf{I}\{b_i^p = 1\}\sigma(s_{\mathbf{w}}(\mathbf{a}_i^p) - \tau)$, $g_2(\mathbf{w}; \tau) = \frac{1}{n_u^+} \sum_{i=1}^{n_u} \mathbf{I}\{b_i^u = 1\}\sigma(s_{\mathbf{w}}(\mathbf{a}_i^u) - \tau)$. As a result, $f$ is convex, $g(\mathbf{w})$ is smooth. As a result, Algorithm 2 is applicable.

**Hyperparameter tuning.** For ALEXR2, we tune $K_t$ in $\{5, 10\}$, tune $\alpha$ (i.e. learning rate for outer loop) and $\eta$ (i.e. learning rate for inner loop) in $\{1e\text{-}3, 1e\text{-}2, 0.1, 1\}$, and tune $\lambda$ (i.e. smoothing coefficient for outer function) and $\nu$ (i.e. smoothing coefficient for overall objective) in $\{2e\text{-}2, 2e\text{-}3, 2e\text{-}4\}$ and $\{0.1, 0.01\}$, respectively. We fix the hyperparameters in MSVR update for both ALEXR2 [7] and SONX, i.e. $\hat{\gamma} = 0.8$ and $\hat{\gamma}' = \hat{\gamma}\theta = 0.1$. For the two baseline algorithms, we tune the initial learning rate in $\{0.1, 1e\text{-}2, 1e\text{-}3, 1e\text{-}4\}$. We decay learning rate(outer lr for ALEXR2) at 50% and 75% epochs by a factor of 10. We tuned $\rho = \{4, 6, 8, 10, 20, 40\}$ for ALEXR2 and SONX and $\rho = \{10, 40, 80, 100, 200, 400, 800, 1000\}$ for SOX. We also compare a double-loop method (ICPPAC) [10, Algorithm 4], where we tune their $\eta$ in $\{0.1, 0.01\}$, $\tau$ in $\{1, 10, 100\}$, $\mu$ in $\{1e\text{-}2, 1e\text{-}3, 1e\text{-}4\}$, and fix $\theta_t$ to 0.1.

### F.3 Continual learning with non-forgetting constraints

We follow [7] and consider training a CLIP model [52] with global contrastive loss (GCL) [17] as the objective and a so-called model developmental safety (MDS) constraints on protected tasks, given by:

$$\min_{\mathbf{w}} F(\mathbf{w}, \mathcal{D}) := \frac{1}{n} \sum_{(\mathbf{x}_i, \mathbf{t}_i) \in \mathcal{D}} L_{\text{GCL}}(\mathbf{w}; (\mathbf{x}_i, \mathbf{t}_i), (\mathcal{T}_i^-, \mathcal{I}_i^-)),$$

$$\text{s.t. } h_k := \mathcal{L}_k(\mathbf{w}, D_k) - \mathcal{L}_k(\mathbf{w}_{\text{old}}, D_k) \le 0, k = 1, \cdots, m.$$

---

[7] As discuss in section 4.2, update of $\mathbf{y}$ in inner loop is equivalent to MSVR update, i.e. $y_{i,t} = \nabla f_{i,\lambda}(u_{i,t}), u_{i,t+1} = (1 - \hat{\gamma})u_{i,t} + \hat{\gamma}g_i(\mathbf{z}_{t,k}; \mathcal{B}_{i,2}^{t,k}) + \hat{\gamma}\theta(g_i(\mathbf{z}_{t,k}; \mathcal{B}_{i,2}^{t,k}) - g_i(\mathbf{z}_{t-1,k}; \mathcal{B}_{i,2}^{t,k}))$

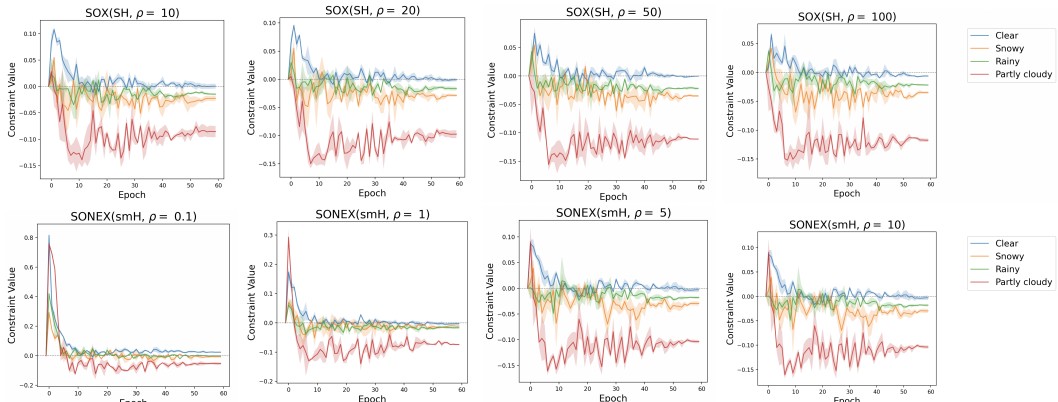

Figure 5: Training curves of 4 constraint values in zero-one loss of different methods for continual learning with non-forgetting constraints when targeting the overcast class. Top: squared-hinge penalty method with different $\rho$; Bottom: smoothed hinge penalty method with different $\rho$.

where $L_{\text{GCL}}(\mathbf{w}; (\mathbf{x}_i, \mathbf{t}_i), (\mathcal{T}_i^-, \mathcal{I}_i^-))$ is the two-way GCL for each image-text pair $(\mathbf{x}_i, \mathbf{t}_i)$,

$$L_{\text{GCL}}(\mathbf{w}; \mathbf{x}_i, t_i, \mathcal{T}_i^-, \mathcal{I}_i^-) := -\tau \log \frac{\exp(E_I(\mathbf{w}, \mathbf{x}_i)^\top E_T(t_i)/\tau)}{\sum_{t_j \in \mathcal{T}_i^-} \exp(E_I(\mathbf{x}_i)^\top E_T(t_j)/\tau)}$$
$$- \tau \log \frac{\exp(E_T(t_i)^\top E_I(\mathbf{x}_i)/\tau)}{\sum_{\mathbf{x}_j \in \mathcal{I}_i^-} \exp(E_T(t_j)^\top E_I(\mathbf{x}_i)/\tau)},$$

where $E_I(\mathbf{x})$ and $E_T(\mathbf{t})$ denote the normalized feature representation of an image $\mathbf{x}$ and a text $t$, generated by visual encoder and text encoder of CLIP model, respectively. $\mathcal{T}_i^-$ denotes the set of all texts to be contrasted with respect to (w.r.t) $\mathbf{x}_i$ (including itself) and $\mathcal{I}_i^-$ denotes the set of all images to be contrasted w.r.t $t_i$ (including itself). Here, the data $\mathcal{D}$ is a target dataset. $\mathcal{L}_k(\mathbf{w}, \mathcal{D}_k) = \frac{1}{n_k} \sum_{(\mathbf{x}_i, y_i) \sim \mathcal{D}_k} \ell_k(\mathbf{w}, \mathbf{x}_i, y_i)$ is the loss of the model $\mathbf{w}$ on the $k$-th protected task, where $\ell_k$ is a logistic loss for the $k$-th classification task.

**Hyperparameter tuning.** The training data for the target tasks are sampled from the training set of BDD100K and LAION400M [53], while the data used to construct the MDS constraints are sampled exclusively from BDD100K. For all the method we use the same learning rate of 1e-6 and the same weight decay 0.1. For each experiment we run for 60 epochs * 400 iterations per epoch and the batch size is 256. We tune $\rho$ in $\{0.01, 0.1, 1, 5, 10, 20, 40, 80\}$ for SONEX+smooth Hinge and in $\{1, 20, 50, 100, 200, 400, 800\}$ for SOX+squared hinge; we set $\gamma_1 = \gamma_2 = 0.8, \gamma_1' = \gamma_2' = 0.1, \tau = 0.05$ where $\gamma_1, \gamma_1'$ is the hyperparameter in MSVR update for objective term (i.e. $L_{\text{GCL}}$ [8]) while $\gamma_2, \gamma_2'$ is the hyperparameter in MSVR update for the penalty terms. We set $\lambda = 0.1$ which is the smoothing coefficient for smooth Hinge function in SONEX.

### F.4 Ablation Study

We conduct a series of ablation studies to examine the effects of key hyperparameters and design choices.

**Results for varying $\beta$** We investigate the impact of different $\beta$ on model performance. We evaluate $\beta \in \{0.1, 0.3, 0.5, 1.0\}$ and conduct experiments for SONEX on Amazon dataset for GDRO task and ALEXR2 on Adult dataset for AUC maximization with ROC fairness constraints. The results are shown in table 3 which indicates the importance of setting $\beta \leq 1.0$ .

**Results for varying $\theta$** We analyze the influence of different $\theta$ of ALEXR2 on model performance, testing $\theta \in \{\frac{1}{16}, \frac{1}{8}, \frac{1}{4}, \frac{1}{2}, 1\}$ and conduct experiments for ALEXR2 on Adult dataset. As shown in Table 4, the final AUC of ALEXR2 remains relatively stable across this range, suggesting that it is not highly sensitive to $\theta$.

---

[8]note that GCL is also a type of FCCO problem, we can also apply MSVR update to track the inner functions

**Benefit of Adam-type update**    Finally we compare Adam-type update with momentum-type update. We conduct experiments for SONEX on Amazon dataset and ALEXR2 on Adult dataset and the results are summarized in table 5. They clearly demonstrate the advantage of using Adam-type updates, which consistently yield superior performance.

Table 3: Final loss and AUC of SONEX and ALEXR2 on Amazon dataset and Adult dataset respectively, with varying $\beta$

| SONEX w/ varying $\beta$ | 0.1 | 0.3 | 0.5 | 1.0 |
|---|---|---|---|---|
| Final loss | 0.5657 | **0.5563** | 0.5768 | 0.578 |

| ALEXR2 w/ varying $\beta$ | 0.1 | 0.3 | 0.5 | 1.0 |
|---|---|---|---|---|
| Final AUC | 0.8975 | **0.8976** | 0.8973 | 0.8969 |

Table 4: Final AUC of ALEXR2 with varying $\theta$ on Adult dataset

| varying $\theta$ | 0.1 | 0.3 | 0.5 | 1.0 |
|---|---|---|---|---|
| Final AUC | 0.8975 | **0.8976** | 0.8973 | 0.8969 |

Table 5: Comparison between Adam-type update and Momentum-type update for SONEX and ALEXR2 on Amazon dataset and Adult dataset, respectively

| Algorithm | Adam-type | Momentum-type |
|---|---|---|
| SONEX(Final Loss) | **0.5657** | 0.9666 |
| ALEXR2(Final AUC) | **0.8975** | 0.861 |

## F.5   Verification of Assumption 4.3

We verify the assumption 4.3 by computing minimum eigenvalue $\lambda_{min}$ of $\nabla \mathbf{g}(\mathbf{w}) \nabla \mathbf{g}(\mathbf{w})^\top$ in group DRO experiment. We compute $\lambda_{min}$ for models trained on Camelyon17 and CelebA from the last epoch and report it in table 6. Our experiment results demonstrate that the minimum eigenvalue of $\nabla \mathbf{g}(\mathbf{w}) \nabla \mathbf{g}(\mathbf{w})^\top$ remains positive after training process finishes.
Besides, assumption 5.1 has been verified empirically in Appendix A.1 of [7].

## F.6   Other Details of Experiments

**Computing Resource and Running time**: The experiments of AUC Maximization with ROC Fairness Constraints and the experiments of group DRO of Camelyon17 dataset and CelebA dataset in our paper is run on an A30 24G GPU, among which the first experiment takes less than 10 minutes for each run while for the second one, Camelyon17 takes about 4 hours and CelebA takes about 5 hours, for each run. The Amazon dataset of group DRO experiment is run on one A100 40GB GPUs and takes about 12 hours each run. The experiment of continual learning with non-forgetting constraints is run on two A100 40GB GPUs and takes about 12 hours each run.
**Data Split**: We perform data split of CelebA dataset ourselves: within each group, samples are divided into training, validation, and test sets in an 8:1:1 ratio. For all the other datasets mentioned in our paper we use default split.
**Other Experiment results** We show the training curves of individual constraint values for our two experiments about constraint optimization, as shown in Figure 4, 5 (continual learning with non-forgetting constraints) and Figure 6, 7 (AUC maximization with ROC fairness constraints). Note that we use other weather conditions except foggy to construct non-forgetting constraints since there is no foggy data in BDD100k for defining such a constraint.

Table 6: Minimum eigen values $\lambda_{min}$ of $\nabla \mathbf{g}(\mathbf{w})\nabla \mathbf{g}(\mathbf{w})^\top$ at different solution.

| Camelyon17 | seed 1 | seed 2 | seed 3 |
|---|---|---|---|
| $\lambda_{min}$ | 0.0206 | 0.0349 | 0.0261 |

| CelebA | seed 10 | seed 20 |
|---|---|---|
| $\lambda_{min}$ | 0.9000 | 0.2713 |

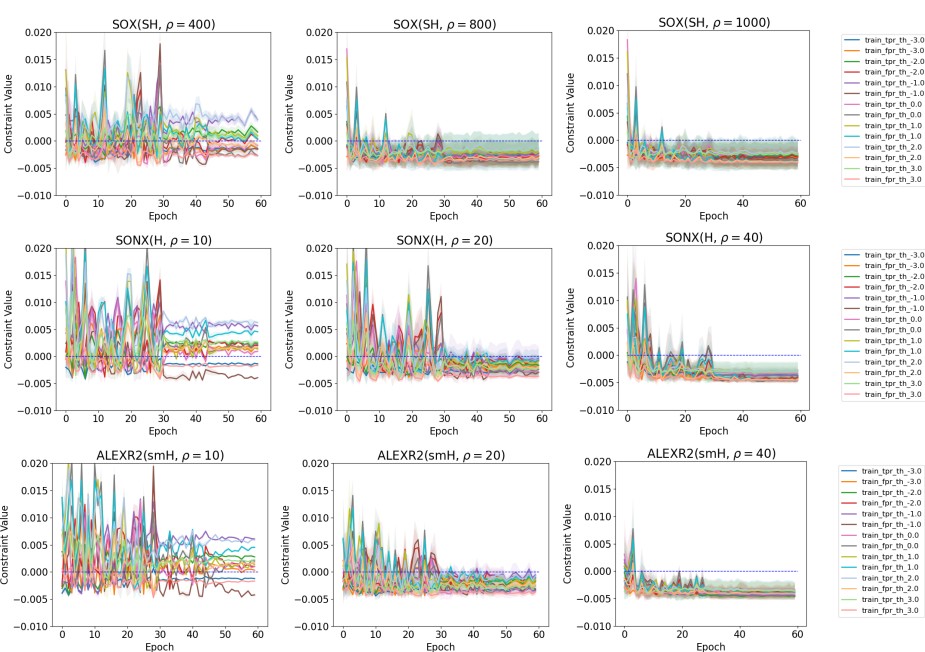

Figure 6: Training curves of 14 constraint values of different methods on adult dataset for AUC maximization with ROC fairness constraints. Top row: SOX with squared-hinge penalty method; Middle: SONX with Hinge penalty method; Bottom: ALEXR2 with smoothed hinge penalty method.

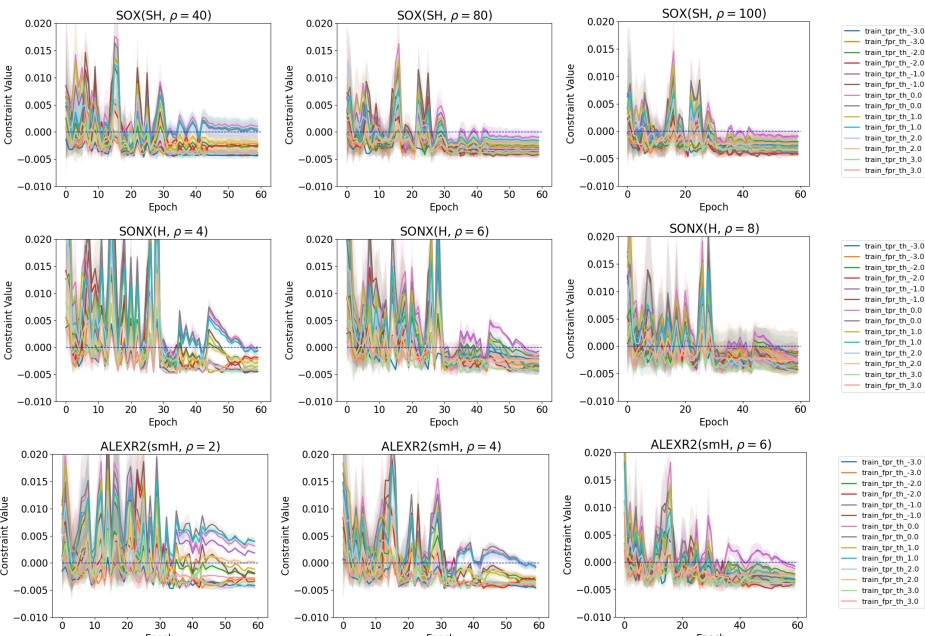

Figure 7: Training curves of 14 constraint values of different methods on COMPAS dataset for AUC maximization with ROC fairness constraints. Top row: SOX with squared-hinge penalty method; Middle: SONX with Hinge penalty method; Bottom: ALEXR2 with smoothed hinge penalty method.

