# OpenReview forum: "Stochastic Momentum Methods for Non-smooth Non-Convex Finite-Sum Coupled Compositional Optimization"
_NeurIPS.cc/2025/Conference — NeurIPS 2025 poster_

### Official Review · Reviewer_gzXU · 2025-06-10

**Clarity:** 1
**Significance:** 3
**Originality:** 3
**Rating:** 4
**Confidence:** 4

**Summary:**

This paper focuses on the finite-sum coupled compositional optimization problem in the case of non-convex and non-smooth functions. There approach combines stochastic estimation of the gradient and momentum. The new propose algorithms SONEX or ALEXR2 have an accelerate convergence rate compared to existing algorithms. Finally, the experimental validation validates the empirical benefit of the method.

**Questions:**

- In line 11 of SONEX algorithm, if you apply a ADAM update, you make a second momentum update after the line 10 computation. What is the interest to do so ?
- Can you generalize your analysis without the restrictive Lipschitz condition on $f$ and $g$ ?
- Why can you apply directly results of Lemma B.1-B.4 that are derive for different algorithms to SONEX algorithm ?

**Ethical Concerns:**

["NO or VERY MINOR ethics concerns only"]

**Final Justification:**

This paper is technically solid and provide new accelerated algorithms both in theory and in practice. I read and try to understand all the proofs/experiments. However, it is a paper for expert and I found it hard to read and follow as a non-expert of FCCO. In particular, I did not catch the motivation and the intuitions to choose these particular algorithms designs (Algorithm 1-2) for having accelerated rates.

Therefore, I choose to keep my score. I think that this work could interest the FCCO community but it is not written to be easily understandable for non-expert of the field.

**Limitations:**

Yes

**Quality:**

3

**Strengths And Weaknesses:**

Strength :
The paper is complete and detailed. The propose method has a provable accelerate rate and perform well in practice.

Weaknesses :
Majors:
- The paper is technical and hard to read for non-expert. A few intuitions are given.
- The assumptions are not commented. In which case are verifying this assumption ? What are the limitations of this assumption ? Can you give examples where these assumptions are verified. In particular, the sense and consequences of Assumption 4.3 and 4.6 are not clear for me. Moreover in Assumption 3.2, the functions are supposed to be Lipschitz. So quadratic functions are not include is this analysis.
- In line 113, $f_{\lambda}$ is said to be $\frac{2 - \lambda\rho}{\lambda - \lambda^2\rho}$ which is true. However, a tighter Lipschitz constant can be derive in $\max\left(\frac{1}{\lambda}, \frac{\rho}{1-\lambda\rho}\right)$ as demonstrated for instance in Lemma 13 of [1]. Thus, Lemma C.2 can also be improved.


Minors:
	- Line 158: Can you put references for the "existing analyses" ?
	- In Theorem 4.7/ SONEX algorithm, I think that you need the inequalities $0<\gamma\le \frac{1}{2}$ (to apply Lemma B.3), could you precise this constraint ?

Typos/minors suggestions :
- In line 473, I think there is no "-" before the subgradient.
- Between line 481 and line 482, the indice of the max is $z_i \in \mathcal{Z}$ however the $z_i$ is outside the sum on $i$, can you make the notation more precise ?
- In line 487, I suggest to precise that here $f_i$ is convex. In fact, if $f_i$ is only weakly convex, the function $f^+$ is not necessarly jointly convex and $f_{i, \lambda}$ not necessarly convex.
- In equation between line 491/492, I suggest to add $z_i \in \mathcal{Z}$ for clarity.
- Line 113, I might be $f_{\lambda}$ instead of $f_{\lambda}(g)$.
- Line 7 of SONEX algorithm : Can you precise the notation of $g_i(w_{t-1}, B_{i, 2}^t)$, the application of $g_i$ on a batch of random samples is not introduce in the paper.
- In Line 124, could you precise the definition of monotonically non-decreasing, it is not a trivial definition for a function $f_i : \mathbb{R}^{d_1} \to \mathbb{R}$.


[1] From stability of Langevin diffusion to convergence of proximal MCMC for non-log-concave sampling, Marien Renaud, Valentin De Bortoli, Arthur Leclaire, Nicolas Papadakis, arXiv 2025

---

> ### Author Rebuttal · Authors · 2025-07-31
>
> We sincerely thank the reviewer for your valuable comments.
>
> **Q1**:In line 11 of SONEX algorithm, if you apply a ADAM update, you make a second momentum update after the line 10 computation. What is the interest to do so?
>
> **A:** This choice offers practical benefits by enabling adaptive learning rates for each parameter, which is especially advantageous in training deep neural networks—particularly Transformer architectures. In our continual learning experiments with CLIP models, which are based on Transformers, this adaptive behavior proved highly beneficial. Please also see the response to Q1 of reviewer soxr.
>
>
> **Q2**: Can you generalize your analysis without the restrictive Lipschitz condition on $f$ and $g$?
>
> **A**: Unfortunately, no if we target non-smooth problems. The Lipschitz continuity of $f$ and $g$ is essential and widely assumed in all existing works on compositional optimization. Even in the standard non-smooth stochastic optimization $\mathbb E_\xi[f(\mathbf w; \xi)]$, the Lipschitz continuity is necessary for the convergence analysis [r8, r9].  In fact, our assumptions are minimal compared to prior studies of FCCO, many of which additionally require smoothness of $g$ and/or $f$.
>
>
>
>
> **Q3**: Why can you apply directly results of Lemma B.1-B.4 that are derive for different algorithms to SONEX algorithm ?
>
> **A**: Indeed, while SONEX is novel in addressing non-smooth FCCO through the use of a smoothed outer function $f_{i,\lambda}$ for gradient estimation (as in Step 9), its other components are inspired by existing algorithms such as SOX and MSVR for smooth FCCO. Thanks to its modular design and analysis, we can draw on prior results for analyzing individual components. Specifically, the $u$-update in Step 7 is identical to that of MSVR, so the recursion for its estimator error (Lemmas B.3 and B.4) is directly borrowed from MSVR [18]. The momentum update in Steps 10–11 resembles that of SOX, allowing us to use Lemma B.2 from [2] to analyze the recursion of the momentum estimator's error. Lemma B.1 is general and applies to any smooth function, regardless of the update mechanism. This modularity enables us to leverage well-established analyses for each component.
>
> **Q4**: About the assumptions.
>
> **A**: Regarding the assumptions:
> - We have verified Assumption 4.3 in our experiment in Appendix F.4.
> - Assumption 4.6 is only used for analyzing SONEX. Our second algorithm ALEXR2 does not need this assumption but still enjoys the same rate. It is necessary for SONEX to analyze the MSVR estimator $u$ following [6] as it does not leverage the convexity of $f$ as ALEXR2. Without this assumption, we can only use the moving average estimator for $u$ and establish a worse complexity of $O(1/\epsilon^7)$ for SONEX and [6] will suffer from an even worse complexity of $O(1/\epsilon^8)$.
>
> - Assumption 4.5 is standard for analyzing stochastic algorithms. Assumption 5.1 is a standard assumption for analyzing penalty-based methods of non-convex constrained optimization (see [r10, r11]). It has been empirically verified in [r10] (Table 2).
> - The Lipschitz continuity of $f$ and $g$ in Assumption 3.1 are fairly standard for FCCO problems. Since we target at **non-smooth $f$ and $g$**, Lipchitz continuity is a minimal assumption. Quadratic functions are smooth and convex, then the problem becomes easier and it is possible to extend existing algorithms for smooth min-max problems without the Lipschitz continuity, e.g., [r12]. That is out of scope of this paper.
> - In the  considered applications of GDRO and  learning with fairness constraints, $f$ is hinge and hence Lipschitz continuous.
>
>
> **Q5**: There exists a tighter lipschitz constant for $f_\lambda$ in line 113.
>
> **A**: We thank reviewer for introducing a new result improving our lemma C.2 and the subsequent theorems, and we will update them in the revision.
>
>
> **Q6**: Some typos.
>
> **A**: We thank reviewer for pointing out all these typos and we will correct them in the revision.
>
>
> [r8] Davis & Drusvyatskiy. Stochastic model-based minimization of weakly convex functions. SIAM OPT, 2019.
>
> [r9] Cutkosky et al. Optimal Stochastic Non-smooth Non-convex Optimization through Online-to-Non-convex Conversion. ICML 2023.
>
> [r10] Yang et al. Single-loop algorithms for stochastic non-convex optimization with weakly-convex constraints. arXiv 2025.
>
> [r11] Liu and  Xu. A single-loop spider-type stochastic subgradient method for expectation constrained nonconvex nonsmooth optimization. arXiv 2025.
>
> [r12] Guo et al. Unified convergence analysis for adaptive optimization with moving average estimator. ML, 2025.

---

> ### Comment · Reviewer_gzXU · 2025-08-05
>
> I thank the authors for these clear responses.
>
> Concerning my first weakness could you explain to me what was your motivation to look at these algorithms SONEX and ALEXR2  ? And what is the intuition behind the acceleration for these algorithms ?
>
> I suggest to add in the main paper the comment on assumptions that you detail in the rebuttal. This will help to understand the scope of the paper and which assumptions are easy/hard to ensure. In particular, I suggest to comment the Lipschitz continuity of $f$ and $g$ as you made in the rebuttal.
>
> I have nothing more to add.

---

> > ### Author Response · Authors · 2025-08-05
> > **Thank you!**
> >
> > We will do that in the revision.

---

> ### Author Response · Authors · 2025-08-06
> **Motivation of Alg 1& 2 and Intuition for Acceleration**
>
> - **For SONEX:** The **motivation** is to smooth the objective so that the resulting objective becomes a smooth FCCO, allowing a momentum method to reduce the error of the gradient estimator to zero, thereby inducing a faster rate. The **intuition** behind the improved rate is that by smoothing the outer function $f$ with a small smoothing parameter $\lambda = O(\epsilon)$, the resulting objective serves as a meaningful surrogate for the original one. Consequently, existing momentum techniques for smooth FCCO can be applied to achieve faster convergence. The best known rate for smooth FCCO is $O\left(\frac{nL_1^2}{B_1\sqrt{B_2}\epsilon^3}\right)$, where $L_1$ is the smoothness parameter of the outer function $f_{i,\lambda}$, which scales as $O(1/\epsilon)$ due to $\lambda = O(\epsilon)$. Therefore, the final convergence rate becomes $O\left(\frac{n}{B_1\sqrt{B_2}\epsilon^5}\right)$.
>
> - **For ALEXR2:** The **motivation** is also to smooth the outer function $f$, such that it transforms into a weakly-convex **strongly**-concave min-max problem under the **convexity** of $f$, making it amenable to inexact proximal-point methods for solving the Moreau envelope. The **intuition** for the improved rate is as follows: for weakly-convex strongly-concave min-max problems, the best rate we can achieve for a double-loop method is $O\left(\frac{1}{\mu\epsilon^4}\right)$, where $\mu$ is the strong concavity parameter. In our case, $\mu = O(\epsilon)$, since it is the inverse of the outer smoothness parameter. As a result, the total complexity is on the order of $O\left(\frac{1}{\epsilon^5}\right)$. The additional scaling factor $\frac{n}{B_1 B_2}$ accounts for the coordinate updates of dual variables and mini-batch updates.
>
> Please let us know if you need any further clarifications! Thank you!

---

> > ### Author Response · Authors · 2025-08-06
> > **Typo in the last response**
> >
> > there is a typo in the last response: $\mu = O(\epsilon)$ since it is the inverse of the smoothness parameter of $f_{i,\lambda}$. The smoothness parameter of $f_{i,\lambda}$ is $O(1/\epsilon)$ due to $\lambda=O(\epsilon)$. The response has been corrected as well.

---

### Official Review · Reviewer_soxr · 2025-06-28

**Clarity:** 3
**Significance:** 3
**Originality:** 3
**Rating:** 4
**Confidence:** 3

**Summary:**

This paper addresses Finite-sum Coupled Compositional Optimization (FCCO), which is non-smooth non-convex, where the out functions are non-smooth and inner functions are smooth or weakly convex. The currently existing state-of-the-art result face two key limitations: (1) iteration complexity O(1/epsilon^6) which is high; (2) reliance on vanilla SGD-type updates, which are not suitable for deep learning applications.

The authors propose two new stochastic momentum methods: SONEX and ALEXR2. They provide theoretical analysis that proves a faster iteration complexity of O(1/epsilon^5), improving upon previous O(1/epsilon^6) results (for finding a nearly epsilon-level KKT solution). This is achieved by applying these algorithms to multiple inequality constrained non-convex optimization problems involving smooth or weakly convex functional inequality constraints, and then optimizing a smoothed hinge penalty based formulation.

Empirical validation is conducted across GDRO, ROC-fair AUC maximization, and continual learning tasks, demonstrating improved performance over several baselines.

**Questions:**

(1) It's better to provide an ablation study that isolates the effect of momentum (β), smoothing parameter (θ), and Adam-type updates. How critical are they to the observed improvements?

(2) Maybe it is better to have more explanation regarding why single-loop method SONEX still has a worse complexity than the double-loop method ALEXR2 in terms of dependence on the inner batch size.

**Ethical Concerns:**

["NO or VERY MINOR ethics concerns only"]

**Final Justification:**

I will keep my score after discussion with authors and other reviewers.

**Limitations:**

Yes

**Quality:**

3

**Strengths And Weaknesses:**

Strengths: Provide a rigorous theoretical proof for proposed method that the convergence rate improves from O(1/epsilon^6) to O(1/epsilon^5).
Provide a lot experiments to show that the proposed methods outperform existing approaches on datasets like Camelyon17, CelebA, and BDD100K.

Weaknesses: The empirical analysis lacks detailed ablation studies isolating the effects of momentum, smoothing, and adaptive learning.

---

> ### Author Rebuttal · Authors · 2025-07-31
>
> We sincerely thank the reviewer for your constructive comments.
>
> **Q1**:It's better to provide an ablation study that isolates the effect of momentum ($\beta$), smoothing parameter ($\theta$), and Adam-type updates. How critical are they to the observed improvements?
>
> **A:** Thanks for the suggestion. This is an easy fix. We have done some ablation studies.
>
> - Using $\beta<1$ is important for SONEX. The ablation study on the Amazon dataset for GDRO shows that SONEX with $\beta=1$  converges slower, as shown in the table below.
>
> | Setting for $\beta$ |0.1 | 0.3 | 0.5 | 1.0 |
> | -------- | -------- | -------- | -------- |-------- |
> | Final Loss   | 0.5657 | **0.5563** | 0.5768 | 0.578|
>
> -  The ablation study shows that ALEXR2 is not sensitive to $\theta\in\{0, 1/16, 1/8, 1/4, 1/2, 1\}$ and $\beta\in\{0.1, 0.3, 0.5, 0.7, 1\}$.  The ablation results of ALEXR2 on the Adult dataset for fair AUC maximization are summarized in the following table.
>
> | Setting for $\beta$ | 0.1 | 0.3 | 0.5 | 0.7 | 1.0 |
> | -------- | -------- | -------- | -------- |-------- |-------- |
> | Final AUC   | 0.8975 | 0.8976 | 0.8973 | 0.8968 | 0.8969 |
>
> | Setting for $\theta$ | 0 | 1/16 | 1/8 | 1/4 | 1/2 | 1 |
> | -------- | -------- | -------- | -------- |-------- |-------- |-------- |
> | Final AUC   | 0.8976 | 0.8981 | 0.8975 | 0.8973 | 0.8972 | 0.8971 |
>
> - Using the Adam-type update is beneficial for both SONEX and ALEXR2 in the ablation studies.
>
> | Setting | Adam-type |momentum-type |
> | -------- | -------- | -------- |
> | ALEXR2 (Final AUC on Adult data)     | **0.8975**     | 0.861     |
> | SONEX (Final Loss on Amazon data)     | **0.5657**| 0.9666     |
>
>
>
> **Q2**: Why single-loop method SONEX still has a worse complexity than the double-loop method ALEXR2 in terms of dependence on the inner batch size.
>
> **A:** This reason is that SONEX does not leverage the convexity of the outer function, while ALEXR2 assumes the convexity of the outer function. Without the convexity of outer function, SONEX directly estimates the inner function values using MSVR, which inherits its dependence on the inner batch size i.e.., $O(1/\sqrt{B_2})$. Using the convexity of the outer function, ALEXR2 leverage the min-max formulation leading to a standard dependence of $O(1/B_2)$ on the inner batch size.
>
> These differences in dependence between the non-convex and convex outer function settings are consistent with prior works [18, 5].

---

> ### Comment · Reviewer_soxr · 2025-08-02
> **keep my score**
>
> Thank you for the rebuttal. I will keep my score as the authors well addressed the issues raised.

---

### Official Review · Reviewer_hTnd · 2025-07-01

**Clarity:** 3
**Significance:** 3
**Originality:** 3
**Rating:** 4
**Confidence:** 3

**Summary:**

In this paper, the authors consider a class of non-convex non-smooth FCCO, where the outer functions are non-smooth weakly convex or convex and the inner functions are smooth or weakly convex. First, they present stochastic momentum methods for non-smooth FCCO that come with provable convergence guarantees; Second, they establish a new state-of-the-art iteration complexity of O(1/epsilon^5). Moreover, they apply the algorithms to multiple inequality constrained non-convex optimization problems involving smooth or weakly convex functional inequality constraints, and achieve a new state-of-the-art complexity of O(1/epsilon^5) for finding an (nearly) epsilon-level KKT solution. They conclude the paper by presenting results on the empirical performance of their algorithms.

**Questions:**

1. What do you see as the biggest difficulty in generalizing the finite sum form to the stochastic expectation form in the problem model?

2. Prior to formally introducing the algorithm, the authors should elaborate in detail on the design logic of each step, specifying which steps are their own innovations and which are inspired by or borrowed from existing literature. What considerations led to the design of the novel steps, and what motivated the selection of the steps borrowed from prior work?

3. Some of the equations are labeled redundant and are not cited below, so please remove them.

4. In numerical experiments, what method is used to solve the subproblem in Algorithm 2?

5. The format of some references needs to be modified, e.g. [1]. x-risks -- X-risks. [30]. lagrangian -- Lagrangian. In addition, the format of proceedings is sometimes inconsistent.

6. In Assumption 4.6, there is an extra “]”.

**Ethical Concerns:**

["NO or VERY MINOR ethics concerns only"]

**Final Justification:**

Thank you for the detailed response. I will maintain my score.

**Quality:**

3

**Strengths And Weaknesses:**

Strengths:

1.The problem they address is interesting and significant for the finite-sum coupled compositional optimization.

2.This is the first work to propose stochastic momentum methods for solving non-smooth FCCO problems.

3.The assumptions are fairly mild.

Weaknesses:
In the introduction, the authors list application scenarios of FCCO, but only briefly mention them without strong motivation. It is recommended to elaborate in detail on at least one representative scenario.

---

> ### Author Rebuttal · Authors · 2025-07-31
>
> We sincerely thank the reviewer for your valuable reviews.
>
> **Q1**: What do you see as the biggest difficulty in generalizing the finite sum form to the stochastic expectation form in the problem model?
>
> **A:** The core of the algorithms for finite-sum setting is to track the inner function values $g_1(\mathbf w_t), \ldots, g_n(\mathbf w_t)$ at all iterations such that the estimation error is decreasing over iterations. The biggest difficulty in generalizing the finite-sum form to the stochastic expectation form $\mathbb E_\xi f_{\xi}(\mathbb E_{\zeta}[g_{\xi}(\mathbf w, \zeta)])$ is how to compute an estimator of the inner function $g_{\xi_t}(\mathbf w_t) = \mathbf E_{\zeta}g_{\xi_t}(\mathbf w_t, \zeta)$ for a random variable $\xi_t$ with decreasing estimation error. Since $\xi_t$ is a fresh random variable that may not be encountered before, hence the moving average or MSVR technique for the finite-sum case is not applicable.  Existing works like BSGD [r7] has to use a very large batch size to control the estimation error, which can be prohibitive in practice.
>
> [r7] Hu et al. Biased stochastic first-order methods for conditional  stochastic optimization and applications in meta learning. NeurIPS, 2020.
>
>
> **Q2**: Elaborate the design of the novel steps, and what motivated the selection of the steps borrowed from prior work?
>
> **A:** We thank the reviewer for the suggestion.  We highlight the novel design below.
>
> **Algorithm 1:** The key novel step is to compute a gradient estimator using the smoothed outer function $f_{i,\lambda}$ at Step 9. It is this step that allows us to borrow the momentum update (Step 10, 11) of an existing algorithm SOX for smooth FCCO problems [2] to tackle non-smooth FCCO problems.
>
> **Algorithm 2:** The key lies at the gradient estimator in Step 13 for the nested smooth objective $F_{\lambda, \nu}$, which allows us to build a recursion of its estimation error for the Moreau envelope defined in (5). The inner loop is borrowed from an existing algorithm ALEXR [5].
>
>
> **Q3**: In numerical experiments, what method is used to solve the subproblem in Algorithm 2?
>
> **A:** As mentioned in section 4.2, we use ALEXR [5] to solve each subproblem. Note that the $\mathbf y$-step update (line 7-9 in algorithm 2) and $\mathbf z$-step update (line 11 in algorithm 2) have closed form updates, i.e.
> $y^i_{t,k} = \nabla f_{i,\lambda}(u^i_{t,k})$, where $u^i_{t,k+1} =(1-\hat\gamma) u^i_{t,k} + \hat\gamma   g_i(\mathbf z_{t,k};B_{i,2}^{t,k}) + \hat\gamma \theta(g_i(\mathbf z_{t,k};B_{i,2}^{t,k}) - g_i(\mathbf z_{t,k-1};B_{i,2}^{t,k}))$ and $\hat\gamma=  \gamma/(1+\gamma)$,
> and
> $\mathbf z_{t,k+1} = \frac{\eta \mathbf w_t +\nu \mathbf z_{t,k}}{\eta+\nu} - \frac{\eta\nu}{\nu+\eta}G_{t,k}$
>
> **Q4**: Some typos.
>
> **A:** We thank reviewer for pointing out all these typos and we will correct them in the revision.
>
> **Q5**: Elaborate in detail on at least one representative scenario of FCCO.
>
> **A:** Thank you for the suggestion!  We have studied multiple applications of FCCO in Section 6, e.g., GDRO with CVaR divergence. The motivation of GDRO is to address the spurious correlation between features and labels, and distributional shift. We will add more details in the revision about the formulation and motivation.

---

> > ### Comment · Reviewer_hTnd · 2025-08-09
> >
> > Thank you for the detailed response. I will maintain my score.

---

### Official Review · Reviewer_3ZyT · 2025-07-03

**Clarity:** 4
**Significance:** 4
**Originality:** 3
**Rating:** 5
**Confidence:** 3

**Summary:**

This paper tackles the challenging problem of non-smooth, non-convex finite-sum coupled compositional optimization (FCCO), where the objective function is defined as $F(\mathbf{w}) = \frac{1}{n}\sum_{i=1}^n f_i(g_i(\mathbf{w}))$. Here, the outer functions $f_i$ are non-smooth (weakly convex or convex), and the inner functions $g_i$ are either smooth or weakly convex. The authors identify two major limitations in prior work (e.g., SONX): (1) a high iteration complexity of $O(\epsilon^{-6})$ under restrictive assumptions, and (2) the use of vanilla SGD updates, which are not well-suited for deep learning applications.

To address these issues, the paper proposes two novel stochastic momentum-based methods:
1. **SONEX** (single-loop): For smooth $g_i$, leverages outer smoothing (via the Moreau envelope) combined with momentum updates.
2. **ALEXR2** (double-loop): For weakly convex $g_i$, applies nested smoothing and a minimax reformulation.

Theoretically, both methods achieve a **new state-of-the-art complexity of $O(\epsilon^{-5})**, representing a notable improvement over the existing $O(\epsilon^{-6})$ rate. The authors further extend these algorithms to constrained non-convex optimization using a smoothed hinge penalty, maintaining the same complexity for obtaining approximate KKT points. Experimental results on group DRO, AUC maximization under fairness constraints, and continual learning show significant performance improvements over established baselines.

**Questions:**

1. **Generality of Assumption 4.3**:
   Theorem 4.4 depends on the condition $\lambda_{\min}(\nabla g_i \nabla g_i^\top) \geq \delta$. Is this assumption realistic for deep networks (e.g., ResNet), where Hessians can be nearly singular? If $\delta$ depends on $\epsilon$, how would that affect the overall complexity? *Empirical validation on ImageNet-scale models would substantially strengthen the claim.*

2. **Efficiency of the Double-Loop Design**:
   ALEXR2 requires inner-loop iterations with $K_t = \tilde{O}(\epsilon^{-3})$. Despite the theoretical gains, is the wall-clock time actually better than SONX in practice? Could a modified single-loop variant handle weakly convex $g_i$ effectively?

3. **Adaptive Learning Rate Assumptions**:
   Appendix E introduces Assumption E.1, which bounds the adaptive learning rates. Is this assumption satisfied in practical scenarios involving Adam? Could the theory be extended to relax this assumption?

**Ethical Concerns:**

["NO or VERY MINOR ethics concerns only"]

**Final Justification:**

I will keep my score after discussion with authors and other reviewers.

**Limitations:**

yes

**Paper Formatting Concerns:**

None.

**Quality:**

3

**Strengths And Weaknesses:**

**Strengths:**
- 1. First to incorporate momentum-based methods (Adam-compatible) into non-smooth FCCO. The complexity reduction is practically meaningful for scalable deep learning.
- 2. The proposed nested smoothing strategy for weakly convex inner functions and the formalization of "approximate $\epsilon$-stationary points" represent important technical innovations.
- 3. Rigorous theoretical analysis (17-page appendix); the convergence guarantees under weak convexity are sound. Experiments span 6 datasets across 3 diverse tasks.
- 4. Clear organization; Figures 1–3 effectively illustrate key comparisons. Detailed pseudocode enhances reproducibility.

**Weaknesses:**
- 1. The complexity results rely on the inner functions being smooth or weakly convex; fully non-smooth inner functions remain an open challenge.
- 2. While effective, the use of Moreau envelope for outer smoothing is a known approach; the novelty primarily lies in combining it with momentum methods.
- 3. Assumption 4.3, which places a lower bound on $\lambda_{\min}(\nabla g_i \nabla g_i^\top)$, lacks empirical validation beyond small-scale experiments (Table 3). Its applicability to large-scale models (e.g., Transformers) is uncertain.
- 4. The double-loop structure of ALEXR2 (Algorithm 2) is relatively complex; a clearer intuition or motivation for nested smoothing would improve accessibility.

---

> ### Author Rebuttal · Authors · 2025-07-31
>
> We sincerely thank the reviewer for your positive rating and valuable reviews.
>
> **Q1**: the novelty primarily lies in combining it with momentum methods.
>
> **A**: We'd like to point out that our second algorithm leverages **two levels** of smoothing, including the smoothing of the outer function $f$ and the Moreau envelope smoothing of the overall objective. It is this **nested smoothing** allows us to establish the state-of-the art complexity of $O(1/\epsilon^5)$ for non-smooth weakly-convex FCCO and non-smooth weakly-convex constrained optimization. If we just apply the Moreau envelope smoothing of the overall objective, the best rate is $O(1/\epsilon^6)$ [26].
>
> **Q2**:**Generality of Assumption 4.3**: Is the condition $\lambda_{min}(\nabla g_i\nabla g_i^\top)\geq \delta$ realistic for deep networks (e.g., ResNet)? If $\delta$ depends on $\epsilon$, how would that affect the overall complexity? Empirical validation on ImageNet-scale models would substantially strengthen the claim.
>
> **A:** Yes to the first question! We kindly refer the reviewer to Table 3 in Appendix F.4 about verifying this condition on ResNet50 (25.5 million paras) for GDRO on CelebA and DenseNet121 (8 million paras) for GDRO on Camelyon17. Both models are deep neural networks widely used for ImageNet-scale experiments. Also, Table 2 in [r1] has verified an equivalent condition for hinge-penalized objective of constraint optimization problem, which is also a special case of FCCO, on the CLIP model where the encoders are Transformers.
>
> For the second question, Theorem 4.4 (for finding a nearly $\epsilon$-stationary solution) can tolerate $\delta$ as small as $O(\sqrt{\epsilon})$ without sacrificing any loss in complexity. Moreover, even the condition $\lambda_{min}(\nabla g_i\nabla g_i^\top)\geq \delta$  does not hold, Theorem 4.2 still guarantees that our algorithm can find a meaningful **approximate $\epsilon$-stationary solution**.
>
>
> **Q3**:**Efficiency of the Double-Loop Design**: ALEXR2 requires inner-loop iterations with $K_t = \tilde{O}(\epsilon^{-3})$. Despite the theoretical gains, is the wall-clock time actually better than SONX in practice? Could a modified single-loop variant handle weakly convex $g_i$ effectively?
>
> **A:**   **On the wall-clock time**. The per-epoch running time of ALEXR2 is longer than SONX due to the momentum update, e.g., 1.87s of ALEXR2 vs 1.8s of SONX on COMPAS dataset.  However, since ALEXR2 converges faster than SONX (cf. Fig. 2), the overall clock time of ALEXR2 is still better for achieving the same level of training performance in our experiments. We give an example below for runtime (secs) reaching a set of given AUC level on the COMPAS dataset for AUC maximization with non-smooth ROC fairness constraints where "N/A" means that the algorithm cannot reach the given AUC.
>
>
> | Algorithm\training AUC | 0.69 | 0.70 | 0.71 | 0.72 |
> | -------- | -------- | -------- |  -------- |  -------- |
> | SONX (runtime:s)     |  10.8    |  21.6    | 37.8 | N/A |
> | ALEXR2 (runtime:s)   |  9.35    | 13.09    |22.44 | 52.36 |
>
>
>
> **On the Single-Loop Design.**
> It is indeed possible to design a single-loop algorithm for the weakly-convex strongly-concave min-max problem under consideration, similar to the approach in [r2]. However, such algorithms typically exhibit much worse dependence on the strong concavity parameter $\mu$ compared to double-loop methods. For instance, the worst-case iteration complexity in [r2] scales as $O(1/\mu^5)$. Even in the smooth setting, single-loop algorithms may suffer from an $O(1/\mu^3)$ dependence in iteration complexity [r3, r4].
>
> In our setting, $\mu$ is on the order of $\epsilon$, since the smoothed outer function has a smoothness of $O(1/\epsilon)$. Consequently, any dependence worse than $O(1/\mu)$ would lead to an iteration complexity exceeding the $O(1/\epsilon^5)$ achieved by our method.
>
>
>
> **Q4**:**Adaptive Learning Rate Assumptions**: Is Assumption E.1 satisfied in practical scenarios involving Adam? Could the theory be extended to relax this assumption?
>
> **A:** We acknowledge that this assumption may be restrictive for the extension, although our main contribution lies in the momentum updates. Our analysis in the adaptive learning rate setting including Assumption E.1 follows [r4], which empirically verified this assumption in their experiments (see Appendix K of their paper).
>
> From a theoretical standpoint, if the stochastic gradient is upper bounded and $\varepsilon$ is a constant, then the adaptive learning rate term $\frac{1}{\sqrt{\mathbf{s}_t} + \varepsilon}$ would also be both lower and upper bounded by constants.
>
> In practice, however, we typically set $\varepsilon$ to a very small value (e.g., $1\text{e}{-8}$). As a result, for coordinates with near-zero second-order moments, $\frac{1}{\sqrt{\mathbf{s}_t} + \varepsilon}$ can become extremely large.
>
> In such cases, the upper bound assumption may no longer hold. One practical solution to mitigate this issue is to apply **clipping** to the adaptive learning rate. Specifically, for coordinates with very small second-order moments, the corresponding first-order moment $\mathbf{v}_{t+1}$ is usually also very small.
>
> Therefore, using a clipped update rule of the form  $\mathbf w_{t+1} = \mathbf w_t - \eta \min(\frac{1}{\sqrt{\mathbf s_t}+\varepsilon}, c_u)\mathbf v_{t+1}$ for some constant $c_u$ may not impact performance significantly, while ensuring the assumption is satisfied.
>
> We also note that general convergence analysis of Adam is challenging. Many existing works have to assume some kind of assumptions or suffer other limitations, e.g., [r5] assumes strong growth condition of the gradient and [r6] has a dependence on $O(1/\varepsilon^2)$ in their complexity, which is non-vacuous only when $\varepsilon$ is a constant, translating to the upper bound $c_u$ in Assumption E.1.
>
> **Q5**: A clearer intuition or motivation for nested smoothing of ALEXR2 would improve accessibility.
>
> **A**: The motivation is as follows. The Moreau envelope smoothing is to ensure the overall objective $F_{\lambda, \nu}$ is smooth so that its stationary point can be found. This is a typical approach for handling non-smooth weakly convex objectives. The smoothing on the outer function $f$ is to ensure the inner min-max problem of (6) strongly-convex and **strongly-concave** so that an existing algorithm ALEXR can be used to solve the inner min-max problem, which naturally leads to a double loop algorithm.
>
>
> [r1] Yang et al. Single-loop Algorithms for Stochastic Non-convex Optimization with Weakly-Convex Constraints. arxiv 2025.
>
> [r2] Hu et al. Single-Loop Stochastic Algorithms for Difference of Max-Structured Weakly Convex Functions. NeurIPS 2024.
>
> [r3] Lin et al. On Gradient Descent Ascent for Nonconvex-Concave
> Minimax Problems.  ICML 2020.
>
> [r4] Guo et al. Unified convergence analysis for adaptive optimization with moving average estimator. ML 2025.
>
> [r5] Zhang et al. Adam Can Converge Without Any Modification On
> Update Rules.  NeurIPS 2022.
>
> [r6] Li et al. Convergence of Adam Under Relaxed Assumptions. NeurIPS 2023.

---

> > ### Comment · Reviewer_3ZyT · 2025-08-07
> >
> > Thank you so much for your comprehensive response. I think it has addressed my questions to some extent. I will keep my score.

---

### Decision · Program_Chairs · 2025-09-17

**Decision:**

Accept (poster)

**Comment:**

This paper studied non-smooth non-convex finite-sum coupled compositional optimization. The proposed stochastic momentum method achieves the best know iteration complexity of $O(1/\epsilon^5)$. It can also be applied to solve the inequality constrained non-convex problem and achieve the new state-of-the-art complexity.

All reviewers are positive to this paper and the questions have been address in rebuttal and discussion. Therefore, I recommend acceptance.